# HoRA: Cross-Head Low-Rank Adaptation with Joint Hypernetworks

## Abstract

Low-Rank Adaptation (LoRA) is a parameter-efficient fine-tuning (PEFT) technique that adapts large pre-trained models by adding low-rank matrices to their weight updates. However, in the context of fine-tuning multi-head self-attention (MHA), LoRA has been employed to adapt each attention head separately, thereby overlooking potential synergies across different heads. To mitigate this issue, we propose a novel **H**yper-shared **Lo**w-**R**ank **A**daptation (HoRA) method, which utilizes joint hypernetworks to generate low-rank matrices across attention heads. By coupling their adaptation through a shared generator, HoRA encourages cross-head information sharing, and thus directly addresses the aforementioned limitation of LoRA. By comparing LoRA and HoRA through the lens of hierarchical mixture of experts, our theoretical findings reveal that the latter achieves superior sample efficiency to the former. Furthermore, through extensive experiments across diverse language and vision benchmarks, we demonstrate that HoRA outperforms LoRA and other PEFT methods while requiring only a marginal increase in the number of trainable parameters.

## 1 Introduction

Fine-tuning large pre-trained models has become the de facto approach for adapting foundation models to downstream tasks. However, the sheer size of modern models makes full fine-tuning computationally expensive and storage-intensive, as it requires updating and storing billions of parameters for each task. To address this challenge, parameter-efficient fine-tuning (PEFT) methods have emerged as a compelling alternative. Instead of updating all parameters, PEFT techniques introduce a small number of additional task-specific parameters while keeping most of the pre-trained weights frozen. This drastically reduces the computational and memory cost of fine-tuning while retaining high task performance. Representative PEFT methods include adapter-tuning (Houlsby et al., 2019a), prefix-tuning (Li & Liang, 2021), and prompt-based approaches such as P-Tuning v2 (Liu et al., 2021) and Compacter (Mahabadi et al., 2021a). Together, these methods have been widely adopted across domains such as natural language processing (Houlsby et al., 2019a), computer vision (Jia et al., 2022), and speech recognition (Gao et al., 2022), enabling practical adaptation of large-scale models in resource-constrained settings.

Among PEFT approaches, Low-rank Adaptation (LoRA) (Hu et al., 2022) has gained particular prominence. LoRA assumes that weight updates lie in a low-rank subspace during fine-tuning, and thus approximates these updates by decomposing them into the product of two low-rank matrices. By injecting these low-rank modules into pre-trained layers (e.g., attention or feedforward layers), LoRA enables models to efficiently capture task-specific information while introducing only a negligible fraction of new parameters. Its efficiency and strong empirical performance have established LoRA as a standard baseline for PEFT in both research and practical deployment. In natural language processing, it is used for domain adaptation, instruction tuning, summarization, question answering, and generation tasks (Huan & Shun, 2025). Recent extensions, such as MTLoRA, have enhanced its applications in multi-task learning and adaptive rank allocation for more efficient transfer learning in foundation models (Agiza et al., 2024). Furthermore, LoRA-based frameworks are being applied in federated learning, speech synthesis, and reinforcement learning scenarios to enable scalable model customization in distributed or resource-constrained environments (Yang et al., 2024; Mei et al., 2024).

While LoRA has emerged as one of the most widely adopted PEFT methods due to its simplicity and efficiency, it has several limitations in the multi-head self-attention setting. First, LoRA learns independent low-rank adapters for each attention head and projection, without any mechanism for coordination or parameter sharing. This can lead to redundancy across heads, as prior work has shown that many attention heads capture overlapping or similar functions (Voita et al., 2019; Michel et al., 2019). Second, the lack of shared structure implies that each head must rely solely on its own gradient signals, which can reduce sample efficiency in low-data fine-tuning scenarios. In this work, we answer the following research question:

> **(Q)** *Can we move beyond fully independent adapters and achieve a method that is both parameter-efficient and capable of meaningful information sharing across heads?*

To address this question, we first generalize prior works that investigate the theoretical connections between Mixture of Experts and single-head attention (Le et al., 2025; Truong et al., 2025a) to the setting of multi-head self-attention. We show that it can naturally be reinterpreted as a Hierarchical Mixture-of-Experts (HMoE). Within this framework, applying LoRA to multi-head self-attention corresponds to refining both the experts and their scoring functions via low-rank matrices. Building on this insight of the connections between MHA and HMoE, we propose Hyper-shared Low-Rank Adaptation (HoRA), a new method that explicitly promotes information sharing across attention heads. Instead of directly and independently learning separate low-rank adapters for each head, HoRA employs a joint hypernetwork to generate these adapters. By implementing shared information among the attention heads, HoRA encourages the experts in the aforementioned HMoE-MHA framework to complement each other by exchanging information, either on the same branch, or across different branches. Moreover, the shared hypernetwork introduces structured coupling: heads are no longer fully independent but instead benefit from common parameterization, while still retaining flexibility through specialized transformations. This design acts as a form of regularization, mitigating redundancy across heads and enabling more coherent and data-efficient adaptation. Our theoretical analysis formally demonstrates that eliminating such redundancy *improves sample efficiency*, and our empirical results corroborate this finding across diverse domains in vision and language tasks where HoRA consistently outperforms several PEFT baselines, including LoRA.

To evaluate HoRA, we conduct both theoretical and experimental studies. Theoretically, we show that HoRA enhances the sample efficiency for the low-rank matrices from an *exponential* rate to a *polynomial* rate. Empirically, we benchmark across vision and language tasks, where HoRA consistently outperforms strong PEFT baselines, including LoRA.

**Contributions.** Our main contributions are summarized as follows:
- We establish a theoretical link between applying LoRA to multi-head self-attention and HMoE. Building on this insight, we propose HoRA, a hypernetwork-based PEFT method that encourages information sharing across attention heads.
- We theoretically demonstrate that HoRA's parameter-sharing mechanism improves sample efficiency from an *exponential* to a *polynomial* rate.
- We empirically show that HoRA substantially *improves sample efficiency* and *achieves superior performance* across diverse tasks, while remaining *parameter-efficient* compared to prior PEFT methods.

## 2 BACKGROUND

**Notation.** For two positive sequences $(a_n)_{n \geq 1}$ and $(b_n)_{n \geq 1}$, if there exists a constant $C > 0$ such that $a_n \leq C b_n$ for all $n$, we denote $a_n = \mathcal{O}(b_n)$ or $a_n \lesssim b_n$. We say that $a_n = \mathcal{O}_P(b_n)$ if their quotient $a_n / b_n$ is bounded in probability, while the notation $a_n = \widetilde{\mathcal{O}}_P(b_n)$ stands for $a_n = \mathcal{O}_P(b_n \log^c(b_n))$, for some $c > 0$. We denote Euclidean norm of $u$ by $\|u\|$, here $|S|$ represents the cardinality of a set $S$. For any positive integer $n \in \mathbb{N}$, we denote $[n] = \{1, 2, \ldots, n\}$. We write $u^\alpha = u_1^{\alpha_1} u_2^{\alpha_2} \cdots u_d^{\alpha_d}$, $|\alpha| = \alpha_1 + \alpha_2 + \cdots + \alpha_d$, and $\alpha! = \alpha_1! \alpha_2! \cdots \alpha_d!$ for a multi-index $\alpha = (\alpha_1, \alpha_2, \ldots, \alpha_d) \in \mathbb{N}^d$. Finally, given a vector $u \in \mathbb{R}^d$, both notations $u = (u^{(1)}, u^{(2)}, \ldots, u^{(d)})$ and $u = (u_1, u_2, \ldots, u_d)$ are employed interchangeably.

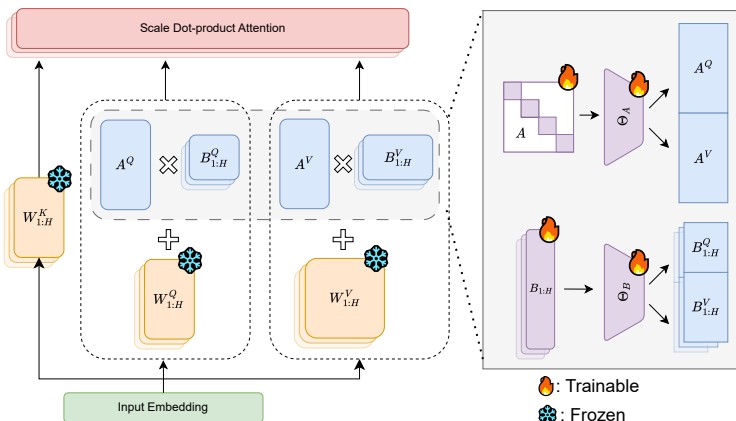

Figure 1: Illustration of HoRA in Multi-head Self-attention.

**Multi-head Self-attention (MHA).** The Transformer architecture (Vaswani et al., 2017; Dosovitskiy et al., 2021) is built upon the MHA mechanism, which enables the model to capture dependencies across different positions of a sequence in parallel. In particular, let $\boldsymbol{X} = [\boldsymbol{x}_1, \cdots, \boldsymbol{x}_N]^T \in \mathbb{R}^{N \times d}$ denote an input sequence of embeddings, where $N$ is the sequence length and $d$ is the embedding dimension. Then, an MHA layer computes its output as

$$\text{MHA}(\boldsymbol{X}_Q, \boldsymbol{X}_K, \boldsymbol{X}_V) = \text{Concat}(\boldsymbol{h}_1, \cdots, \boldsymbol{h}_H)\boldsymbol{W}_O \tag{1}$$

where $H$ is the number of attention heads, and each head $\boldsymbol{h}_i$, for $i \in \{1, 2, \ldots, H\}$, is defined as

$$\boldsymbol{h}_i = \text{Attention}(\boldsymbol{X}\boldsymbol{W}_{Q,i}, \boldsymbol{X}\boldsymbol{W}_{K,i}, \boldsymbol{X}\boldsymbol{W}_{V,i}) = \text{softmax}\left(\frac{\boldsymbol{X}\boldsymbol{W}_{Q,i}\boldsymbol{W}_{K,i}^\top\boldsymbol{X}^\top}{\sqrt{d_k}}\right)\boldsymbol{X}\boldsymbol{W}_{V,i}. \tag{2}$$

Above, the MHA layer projects the input sequence $\boldsymbol{X}$ into queries $\boldsymbol{X}_Q = (\boldsymbol{X}\boldsymbol{W}_{Q,1}, \cdots, \boldsymbol{X}\boldsymbol{W}_{Q,H})$, keys $\boldsymbol{X}_K = (\boldsymbol{X}\boldsymbol{W}_{K,1}, \cdots, \boldsymbol{X}\boldsymbol{W}_{K,H})$, and values $\boldsymbol{X}_V = (\boldsymbol{X}\boldsymbol{W}_{V,1}, \cdots, \boldsymbol{X}\boldsymbol{W}_{V,H})$ where $\boldsymbol{W}_{Q,i}, \boldsymbol{W}_{K,i} \in \mathbb{R}^{d \times d_k}$, $\boldsymbol{W}_{V,i} \in \mathbb{R}^{d \times d_v}$ are projection matrices of the $i^\text{th}$ head and $\boldsymbol{W}_O \in \mathbb{R}^{Hd_v \times d}$ is the output projection. The dimensions are typically chosen such that $d_k = d_v = \frac{d}{H}$.

**Multi-head Low-rank Adaptation (MH-LoRA)** Fine-tuning a large pre-trained transformer model for a downstream task is computationally expensive as it requires updating a massive number of parameters. LoRA (Hu et al., 2022) addresses this issue by updating the pre-trained weights with the product of two low-rank matrices. These ideas come from the observation that weight updates typically lie in a low-dimensional subspace, that is, they have low "intrinsic rank" during fine-tuning. Mathematically, given a pre-trained weight matrix $\boldsymbol{W}_0 \in \mathbb{R}^{m \times n}$, LoRA parameterizes the weight update $\Delta\boldsymbol{W}$ as the product of two low-rank matrices, that is, $\Delta\boldsymbol{W} = \boldsymbol{B}\boldsymbol{A}$ where $\boldsymbol{B} \in \mathbb{R}^{m \times r}$, $\boldsymbol{A} \in \mathbb{R}^{r \times n}$ and $r \ll \min(m, n)$. For an input sequence $\boldsymbol{X}$, the corresponding output is

$$\widehat{\boldsymbol{y}} = \boldsymbol{W}_0\boldsymbol{X} + \boldsymbol{B}\boldsymbol{A}\boldsymbol{X}.$$

During training, only matrices $\boldsymbol{A}$ and $\boldsymbol{B}$ are updated while the pre-trained weights $\boldsymbol{W}_0$ remain frozen. In practice, LoRA is applied to fine-tune projection matrices in the attention layers and we refer to it as *Multi-head LoRA*. Denote $\boldsymbol{A} = [\boldsymbol{A}_Q, \boldsymbol{A}_V]$ and $\boldsymbol{B} = [\boldsymbol{B}_Q, \boldsymbol{B}_V]$, then the output of multi-head LoRA is given by

$$f_{\text{MH−LoRA}}(\boldsymbol{X}; \boldsymbol{A}, \boldsymbol{B}) = \text{Concat}(\tilde{\boldsymbol{h}}_1, \cdots, \tilde{\boldsymbol{h}}_H)\boldsymbol{W}_O,$$

where for each head $i \in \{1, 2, \ldots, H\}$,

$$\tilde{\boldsymbol{h}}_i = \text{Attention}(\boldsymbol{X}\boldsymbol{W}_{Q,i} + \boldsymbol{X}\boldsymbol{B}_{Q,i}\boldsymbol{A}_{Q,i}, \boldsymbol{X}\boldsymbol{W}_{K,i}, \boldsymbol{X}\boldsymbol{W}_{V,i} + \boldsymbol{X}\boldsymbol{B}_{V,i}\boldsymbol{A}_{V,i}).$$

Here, $\boldsymbol{A}_{Q,i} \in \mathbb{R}^{r \times d_k}$, $\boldsymbol{B}_{Q,i} \in \mathbb{R}^{d \times r}$, $\boldsymbol{A}_{V,i} \in \mathbb{R}^{r \times d_v}$, and $\boldsymbol{B}_{V,i} \in \mathbb{R}^{d \times r}$ are the trainable low-rank matrices for the $i^{th}$ head, for $i \in \{1, 2, \ldots, H\}$.

**(Hierarchical) Mixture of Experts (HMoE)** MoE framework (Jacobs et al., 1991) is a model that decomposes a learning task into several sub-models, each specializing in a particular input region

or representation pattern. Formally, an MoE model consists of $N$ expert functions $f_i : \mathbb{R}^d \to \mathbb{R}^{d_v}$ for $i \in [N]$ and a gating function $G : \mathbb{R}^d \to \mathbb{R}^N$ that dynamically assigns input-dependent weights to the experts. The model output is given by $\widehat{y} = \sum_{i=1}^{N} G(\boldsymbol{X})_i \cdot f_i(\boldsymbol{X})$, where $G(\boldsymbol{X}) = \text{softmax}(\{s_i(\boldsymbol{X})\}_{i=1}^{N})$, and each $s_i(\boldsymbol{X})$ is a similarity score between the input and the $i^{th}$ expert.

HMoE (Jordan & Jacobs, 1994) extends the standard MoE by organizing experts in a tree-structured hierarchy. Instead of a single gating function, the HMoE model employs multiple gating nodes arranged in levels. For example, let us consider a 2-layer HMoE model. The first level employs gating functions $G_i(\boldsymbol{X})$, while the second level employs conditional gating function $G_{j|i}(\boldsymbol{X})$ associated with experts $f_{j|i}(\boldsymbol{X})$. The overall prediction of the HMoE model is given by

$$\widehat{y} = \sum_i G_1(\boldsymbol{X})_i \sum_j G_2(\boldsymbol{X})_{j|i} f_{j|i}(\boldsymbol{X}).$$

## 3 Theoretical Developments

In this section, we first establish a relation between multi-head LoRA and HMoE models in Section 3.1. From the HMoE perspective, we proceed to analyze the sample complexity of estimating low-rank matrices in the multi-head LoRA without and with the shared structure across attention heads in Sections 3.2 and 3.3, respectively. Our goal is to show that employing the shared structure yields a significant gain in the sample efficiency of estimating low-rank matrices.

### 3.1 Multi-head LoRA meets Hierarchical Mixture of Experts

In the sequel, we aim to show that multi-head LoRA can be interpreted as an HMoE model. Now, let $\tilde{\boldsymbol{x}} = \text{Vec}(\boldsymbol{X}) = (\boldsymbol{x}_1^\top, \dots, \boldsymbol{x}_N^\top)^\top \in \mathbb{R}^{Nd}$ denote the vectorization of $\boldsymbol{X}$. Denote $\boldsymbol{W}_O = ((\boldsymbol{W}_{O,1})^\top, (\boldsymbol{W}_{O,2})^\top, \dots, (\boldsymbol{W}_{O,H})^\top)^\top$, then from the definition of multi-head self-attention matrix in Eq. (2), we have

$$\text{MHA}(\boldsymbol{X}_Q, \boldsymbol{X}_K, \boldsymbol{X}_V) = \sum_{h=1}^{H} \boldsymbol{h}_h \boldsymbol{W}_{O,h} = \sum_{h=1}^{H} \text{softmax}\left(\frac{\boldsymbol{X}\boldsymbol{W}_{Q,h}(\boldsymbol{W}_{K,h})^\top \boldsymbol{X}^\top}{\sqrt{d_v}}\right) \boldsymbol{X}\boldsymbol{W}_{V,h}\boldsymbol{W}_{O,h}.$$

Let $\boldsymbol{M}_h := \frac{\boldsymbol{W}_{Q,h}(\boldsymbol{W}_{V,h})^\top}{\sqrt{d_v}}$, and $\boldsymbol{J}_i := \boldsymbol{e}_i^\top \otimes \boldsymbol{I}_d$, here $\otimes$ stands for Kronecker product

$$\boldsymbol{J}_i = e_i^\top \otimes \boldsymbol{I}_d = [\boldsymbol{0}_{d\times d} \quad \cdots \quad \boldsymbol{0}_{d\times d} \quad \boldsymbol{I}_d \quad \boldsymbol{0}_{d\times d} \quad \cdots \quad \boldsymbol{0}_{d\times d}] \in \mathbb{R}^{d\times Nd},$$

then, $\boldsymbol{J}_i$ can extract the $i^{th}$ row of a matrix $\boldsymbol{J}_i\tilde{\boldsymbol{x}} = \boldsymbol{x}_i$. Let $\boldsymbol{B}_{ij}^h = \boldsymbol{J}_i^\top \boldsymbol{M}_h \boldsymbol{J}_j$, and $\boldsymbol{E}_j^h = \boldsymbol{J}_j^\top \boldsymbol{W}_{V,h}$, then the $(i, j)$-entry can be expressed as

$$[\boldsymbol{X}\boldsymbol{M}_h\boldsymbol{X}^\top]_{i,j} = \tilde{\boldsymbol{x}}_i \boldsymbol{M}_h \tilde{\boldsymbol{x}}_j = \tilde{\boldsymbol{x}}_i \boldsymbol{M}_h \tilde{\boldsymbol{x}}_j = \tilde{\boldsymbol{x}}^\top \boldsymbol{J}_i^\top \boldsymbol{M}_h \boldsymbol{J}_j \tilde{\boldsymbol{x}} = \tilde{\boldsymbol{x}}^\top \boldsymbol{B}_{ij}^h \tilde{\boldsymbol{x}}.$$

As a result, the value at the $i^{th}$ row is given by

$$[\text{MHA}(\boldsymbol{X}_Q, \boldsymbol{X}_K, \boldsymbol{X}_V)]_i = \sum_{h=1}^{H} \sum_{j=1}^{N} \frac{\exp(\tilde{\boldsymbol{x}}^\top \boldsymbol{B}_{ij}^h \tilde{\boldsymbol{x}})}{\sum_{l=1}^{N} \exp(\tilde{\boldsymbol{x}}^\top \boldsymbol{B}_{il}^h \tilde{\boldsymbol{x}})} \cdot \tilde{\boldsymbol{x}}^\top \boldsymbol{E}_j^h \boldsymbol{W}_{O,h}.$$

Denote $s_{i,j}^h : \mathbb{R}^{Nd} \to \mathbb{R}$ be the score functions and $f_j^h : \mathbb{R}^{Nd} \to \mathbb{R}^{d_v}$ be the expert functions

$$s_{i,j}^h(\tilde{\boldsymbol{x}}) = \tilde{\boldsymbol{x}}^\top \boldsymbol{B}_{ij}^h \tilde{\boldsymbol{x}} = \frac{\boldsymbol{x}_i^\top \boldsymbol{W}_{Q,k}(\boldsymbol{W}_{K,k})^\top \boldsymbol{x}_i}{\sqrt{d_v}}, \ f_j^h(\tilde{\boldsymbol{x}}) = \tilde{\boldsymbol{x}}^\top \boldsymbol{E}_j^h.$$

Then, the output of the $i^{th}$ row in the MHA can be formulated as a HMoE:

$$[\text{MHA}(\boldsymbol{X}_Q, \boldsymbol{X}_K, \boldsymbol{X}_V)]_i = \sum_{h=1}^{H} \sum_{j=1}^{N} \frac{\exp(s_{ij}(\tilde{\boldsymbol{x}}))}{\sum_{k=1}^{N} \exp(s_{ik}(\tilde{\boldsymbol{x}}))} f_j^h(\tilde{\boldsymbol{x}})\boldsymbol{W}_{O,h}.$$

Applying LoRA allows the experts and the score function to be refined by the low-rank updates:

$$\tilde{f}_j^h(\tilde{\boldsymbol{x}}) = \tilde{\boldsymbol{x}}^\top \boldsymbol{J}_j^\top (\boldsymbol{W}_{V,h} + \underline{\boldsymbol{B}_{V,h}\boldsymbol{A}_{V,h}}), \tag{3}$$

$$\tilde{s}_{i,j}^h(\boldsymbol{x}) = \frac{\tilde{\boldsymbol{x}}^\top \boldsymbol{P}_i^\top (\boldsymbol{W}_{Q,h} + \underline{\boldsymbol{B}_{Q,h}\boldsymbol{A}_{Q,h}})(\boldsymbol{W}_{K,h})^\top \boldsymbol{P}_i\tilde{\boldsymbol{x}}}{\sqrt{d_v}} = \frac{\boldsymbol{x}_i^\top (\boldsymbol{W}_{Q,h} + \underline{\boldsymbol{B}_{Q,h}\boldsymbol{A}_{Q,h}})(\boldsymbol{W}_{K,h})^\top \boldsymbol{x}_i}{\sqrt{d_v}}, \tag{4}$$

where $h \in [H]$ and $j \in [N]$. In this case, the $i^{th}$ row of multi-head LoRA can be written as

$$[f_{\mathrm{MH-LoRA}}(\boldsymbol{X}, \boldsymbol{A}, \boldsymbol{B})]_i = \sum_{h=1}^H \sum_{j=1}^N \frac{\exp(\tilde{s}_{ij}(\boldsymbol{x}))}{\sum_{k=1}^N \exp(\tilde{s}_{ik}(\boldsymbol{x}))} \tilde{f}_j^h(\tilde{\boldsymbol{x}}) \boldsymbol{W}_{O,h}.$$

This equation formalizes the relationship between the multi-head LoRA framework and the HMoE model, a connection that plays a central role in our subsequent theoretical analysis.

## 3.2 WITHOUT SHARED STRUCTURE

From the HMoE perspective, we will determine the sample complexity of estimating low-rank matrices in multi-head LoRA without the shared structure across attention heads in this section. For that purpose, let us present a regression framework that has been adopted by several MoE-based PEFT works for studying the asymptotic properties of their models, including prefix tuning (Le et al., 2025) and LLaMA-adapter (Diep et al., 2025).

**Problem setup.** Let $(\boldsymbol{X}_1, \boldsymbol{Y}_1), (\boldsymbol{X}_2, \boldsymbol{Y}_2), \ldots, (\boldsymbol{X}_n, \boldsymbol{Y}_n) \in \mathbb{R}^{\bar{d}} \times \mathbb{R}^{\bar{d}}$ be i.i.d. samples of size $n$ generated from the following regression model:

$$\boldsymbol{Y}_i = g_{G_*}(\boldsymbol{X}_i) + \varepsilon_i, \quad i = 1, 2, \ldots, n, \tag{5}$$

where we assume that the inputs $\boldsymbol{X}_1, \boldsymbol{X}_2, \ldots, \boldsymbol{X}_n$ are i.i.d. samples from some probability distribution $\mu$ with bounded support $\mathcal{X}$. Meanwhile, $\varepsilon_1, \varepsilon_2, \ldots, \varepsilon_n$ are independent Gaussian noise variables such that $\mathbb{E}[\varepsilon_i | \boldsymbol{X}_i] = 0$ and $\mathrm{Var}[\varepsilon_i | \boldsymbol{X}_i] = \sigma^2 I_{\bar{d}}$, for all $i \in [n]$. Meanwhile, the HMoE-based regression function $g_{G_*}$ consists of $H$ expert groups, each of which has $L$ experts:

$$g_{G_*}(\boldsymbol{X}) := \sum_{h=1}^H \pi_h^* \sum_{j=1}^L \frac{\exp(\boldsymbol{X}^\top(\boldsymbol{P}_{Q,h}^0 + \boldsymbol{B}_{Q,h,j}^* \boldsymbol{A}_{Q,h,j}^*)\boldsymbol{P}_{K,h}^0 \boldsymbol{X} + c_j^*)}{D_{g,*}^h(\boldsymbol{X})} \cdot (\boldsymbol{P}_{V,h}^0 + \boldsymbol{B}_{V,h,j}^* \boldsymbol{A}_{V,h,j}^*)\boldsymbol{X}, \tag{6}$$

where we denote $D_{g,*}^h(\boldsymbol{X}) := \sum_{j'=1}^L \exp(\boldsymbol{X}^\top(\boldsymbol{P}_{Q,h}^0 + \boldsymbol{B}_{Q,h,j'}^* \boldsymbol{A}_{Q,h,j'}^*)\boldsymbol{P}_{K,h}^0 \boldsymbol{X} + c_{j'}^*)$, while $G_* := \sum_{h=1}^H \pi_h^* \sum_{j'=1}^L \exp(c_{j'}^*)\delta_{(\boldsymbol{B}_{Q,h,j'}^*, \boldsymbol{A}_{Q,h,j'}^*, \boldsymbol{B}_{V,h,j'}^*, \boldsymbol{A}_{V,h,j'}^*)}$ represents a *mixing measure*, that is, a combination of Dirac measures $\delta$, associated with unknown parameters $(\pi_h^*, c_{j'}^*, \boldsymbol{B}_{Q,h,j'}^*, \boldsymbol{A}_{Q,h,j'}^*, \boldsymbol{B}_{V,h,j'}^*, \boldsymbol{A}_{V,h,j'}^*)_{j' \in [L], h \in [H]}$ in the compact parameter space $\widehat{\Theta} \subset [0,1] \times \mathbb{R} \times \mathbb{R}^{\bar{d} \times r} \times \mathbb{R}^{r \times \bar{d}} \times \mathbb{R}^{\bar{d} \times r} \times \mathbb{R}^{r \times \bar{d}}$. In addition, the matrices $\boldsymbol{P}_{Q,h}^0 \in \mathbb{R}^{\bar{d} \times \bar{d}}$, $\boldsymbol{P}_{K,h}^0 \in \mathbb{R}^{\bar{d} \times \bar{d}}$, and $\boldsymbol{P}_{V,h}^0 \in \mathbb{R}^{\bar{d} \times \bar{d}}$ are frozen so as to align with the formulations in Eq. (3) and Eq. (4).

**Least squares estimator.** We can estimate low-rank matrices $(\boldsymbol{B}_{Q,h,j'}^*, \boldsymbol{A}_{Q,h,j'}^*, \boldsymbol{B}_{V,h,j'}^*, \boldsymbol{A}_{V,h,j'}^*)$ through estimating the ground-truth mixing measure $G_*$. To this end, we employ the least-squares method (van de Geer, 2000), which yields the following estimator:

$$\widehat{G}_n := \arg \min_{G \in \mathcal{G}_{H,L'}(\widehat{\Theta})} \sum_{i=1}^n (\boldsymbol{Y}_i - g_G(\boldsymbol{X}))^2,$$

where $\mathcal{G}_{H,L'}(\widehat{\Theta}) = \{G = \sum_{h=1}^H \pi_h \sum_{j'=1}^\ell \exp(c_{j'})\delta_{(\boldsymbol{B}_{Q,h,j'}, \boldsymbol{A}_{Q,h,j'}, \boldsymbol{B}_{V,h,j'}, \boldsymbol{A}_{V,h,j'})}, 1 \le \ell \le L', (\pi_h, c_{j'}, \boldsymbol{B}_{Q,h,j'}, \boldsymbol{A}_{Q,h,j'}, \boldsymbol{B}_{V,h,j'}, \boldsymbol{A}_{V,h,j'}) \in \widehat{\Theta}\}$ denotes the set of all mixing measures whose expert group has at most $L'$ experts. As the true number of experts $L$ is usually unknown in practice, it is natural to fit each expert group by $L'$ experts, where $L'$ is sufficiently large such that $L' > L$.

**Voronoi loss.** Consider a mixing measure $G \in \mathcal{G}_{H,L'}(\widehat{\Theta})$. For $h \in [H]$, denote $\tau(h) \in [H]$ be the value such that $|\pi_{\tau(h)} - \pi_h^*| \leq |\pi_{\tau(h)} - \pi_{h'}^*|$ for each $h' \in [H]$. To quantify the discrepancy between two mixing measures, we consider a loss function built upon the concepts of Voronoi cells $\{\mathcal{W}_{j|h} \equiv \mathcal{W}_{j|h}(G) : j \in [L'], h \in [H]\}$ generated by the atoms of $G_*$ (Manole & Ho, 2022): $\mathcal{W}_{j|h} = \{i \in [L'] : \|\boldsymbol{H}_{\tau(h),i} - \boldsymbol{H}_{h,j}^*\| \leq \|\boldsymbol{H}_{\tau(h),i} - \boldsymbol{H}_{h,l}^*\|, \forall l \neq j\}$, where we denote $\boldsymbol{H} := (\boldsymbol{B}_Q, \boldsymbol{A}_Q, \boldsymbol{B}_V, \boldsymbol{A}_V)$. Then, the Voronoi loss of interest is defined as

$$
\mathcal{D}_{1,r}(G, G_*) := \sum_{h=1}^{H} |\pi_{\tau(h)} - \pi_h^*| + \sum_{h=1}^{H} \pi_{\tau(h)} \sum_{l=1}^{L} \left| \sum_{i \in \mathcal{W}_{l|h}} \exp(c_i) - \exp(c_h^*) \right|
$$

$$
+ \sum_{h=1}^{H} \pi_{\tau(h)} \sum_{l=1}^{L} \sum_{i \in \mathcal{W}_{l|h}} \exp(c_i)(\|\Delta \boldsymbol{B}_{Qh,il}\|^r + \|\Delta \boldsymbol{A}_{Qh,il}\|^r + \|\Delta \boldsymbol{B}_{Vh,il}\|^r + \|\Delta \boldsymbol{A}_{Vh,il}\|^r) \quad (7)
$$

where $\Delta \boldsymbol{B}_{Qh,il} := \boldsymbol{B}_{Q,\tau(h),i} - \boldsymbol{B}_{Q,h,l}^*$, and $\Delta \boldsymbol{A}_{Qh,il}$, $\Delta \boldsymbol{B}_{Vh,il}$, $\Delta \boldsymbol{A}_{Vh,il}$ are defined similarly. With these components in place, we are now prepared to analyze the sample complexity of estimating low-rank matrices in multi-head LoRA under the non-shared setting.

**Theorem 1.** *Under the non-shared structure setting in Eq.(6), the following minimax lower bound of estimating $G_*$ satisfies for any $r \geq 1$:*

$$
\sup_{G \in \mathcal{G}_{H,L'}(\widehat{\Theta}) \setminus \mathcal{G}_{H,L-1}(\widehat{\Theta})} \mathbb{E}_{g_G}[\mathcal{D}_{1,r}(\widehat{G}_n, G)] \gtrsim n^{-1/2}, \quad (8)
$$

*where $\mathbb{E}_{g_G}$ stands for the expectation taken with respect to the product measure $g_G^n$.*

The proof of Theorem 1 is in Appendix B.1. The result of Theorem 1 implies that the rates for estimating low-rank matrices $\boldsymbol{B}_{Q,h,j'}^*, \boldsymbol{A}_{Q,h,j'}^*, \boldsymbol{B}_{V,h,j'}^*, \boldsymbol{A}_{V,h,j'}^*$ are slower than any polynomial rates of order $\mathcal{O}_P(n^{-1/2r})$, for $r \geq 1$. Therefore, these rates may be as slow as $\mathcal{O}_P(\log^{-\tau}(n))$ for some constant $\tau > 0$ (according to the inequality $\log(n) < n$).

**Sample complexity of estimating low-rank matrices.** Consequently, we may need exponentially data points of the order $\mathcal{O}(\exp(\epsilon^{-1/\tau}))$ to achieve estimators of the low-rank matrices $\boldsymbol{B}_{Q,h,j'}^*, \boldsymbol{A}_{Q,h,j'}^*, \boldsymbol{B}_{V,h,j'}^*, \boldsymbol{A}_{V,h,j'}^*$ with a given error $\epsilon > 0$. Thus, the sample complexity of estimating low-rank matrices in multi-head LoRA without the shared structure across attention heads is suboptimal. This issue occurs due to the separate structures of the low-rank matrices, which yields a negative interaction among low-rank matrices expressed through the partial differential equation (PDE) $\frac{\partial^2 F}{\partial B_V^{(u_1 v_1)} \partial B_V^{(u_2 v_2)}} = \frac{\partial^2 F}{\partial A_V^{(u_1 v_1)} \partial A_V^{(u_2 v_2)}} = 0$, where we define $F(\boldsymbol{X}, \boldsymbol{A}, \boldsymbol{B}) := \exp(\boldsymbol{X}^\top (\boldsymbol{P}_Q + \boldsymbol{B}_Q \boldsymbol{A}_Q) \boldsymbol{P}_K \boldsymbol{X})(\boldsymbol{P}_V + \boldsymbol{B}_V \boldsymbol{A}_V)\boldsymbol{X}$. As shown in a previous work on MoE theories (Nguyen et al., 2024c), this PDE-based interaction decelerates the convergence rate of parameter estimation. The simple linear form of experts in Eq. (6) also accounts for the slow parameter estimation rates, which has been justified in (Nguyen et al., 2024c).

### 3.3 With Shared Structure

**Shared structure across attention heads.** To address the issue of suboptimal sample complexity of estimating low-rank matrices in the non-shared setting, we impose a shared structure across attention heads in multi-head LoRA. In particular, we reformulate the low-rank matrices as

$$
\boldsymbol{A}_{Q,h,j} = \sigma_1(\boldsymbol{W}_{Q,\boldsymbol{A},j} \boldsymbol{A}_j), \quad \boldsymbol{A}_{V,h,j} = \sigma_1(\boldsymbol{W}_{V,\boldsymbol{A},j} \boldsymbol{A}_j)
$$
$$
\boldsymbol{B}_{Q,h,j} = \sigma_2(\boldsymbol{W}_{Q,\boldsymbol{B},j} \boldsymbol{B}_{h,j}), \quad \boldsymbol{B}_{V,h,j} = \sigma_2(\boldsymbol{W}_{V,\boldsymbol{B},j} \boldsymbol{B}_{h,j}),
$$

for all $j \in [N]$ and $h \in [H]$, where $\sigma_1$ and $\sigma_2$ are some activation functions, while $\boldsymbol{W}_{Q,\boldsymbol{A},j}, \boldsymbol{W}_{V,\boldsymbol{A},j}, \boldsymbol{W}_{Q,\boldsymbol{B},j}$, and $\boldsymbol{W}_{V,\boldsymbol{B},j}$ are weight matrices. Above, $\boldsymbol{A}_{Q,h,j}$ and $\boldsymbol{A}_{V,h,j}$ share the matrix $\boldsymbol{A}_j$, while $\boldsymbol{B}_{Q,h,j}$ and $\boldsymbol{B}_{V,h,j}$ share the matrix $\boldsymbol{B}_{h,j}$. Given this shared structure, it can be checked that the PDE-based interaction among low-rank matrices at the end of Section 3.2 no longer occurs. For simplicity, we will set $\boldsymbol{W}_{Q,\boldsymbol{A},j} = \boldsymbol{W}_{V,\boldsymbol{A},j} = \boldsymbol{W}_{1,j}$ and $\boldsymbol{W}_{Q,\boldsymbol{B},j} = \boldsymbol{W}_{V,\boldsymbol{B},j} = \boldsymbol{W}_{2,j}$ with a note that the original shared setting can be analyzed in a similar fashion. For the sake of theory, we assume that the activation functions $\sigma_1$ and $\sigma_2$ satisfy conditions specified in Appendix B.2.

**Problem setup.** In this setting, we still assume that $(\boldsymbol{X}_1, \boldsymbol{Y}_1), (\boldsymbol{X}_2, \boldsymbol{Y}_2), \ldots, (\boldsymbol{X}_n, \boldsymbol{Y}_n) \in \mathbb{R}^{\bar{d}} \times \mathbb{R}^{\bar{d}}$ are i.i.d. samples drawn from a regression framework but with the following regression function:

$$g_{\widetilde{G}_*}(\boldsymbol{X}) := \sum_{h=1}^{H} \pi_h^* \sum_{j=1}^{L} \frac{\exp(\boldsymbol{X}^\top (\boldsymbol{P}_{Q,h}^0 + \sigma_2(\boldsymbol{W}_{2,j}^* \boldsymbol{B}_{h,j}^*)) \sigma_1(\boldsymbol{W}_{1,j}^* \boldsymbol{A}_j^*) \boldsymbol{P}_{K,h}^0 \boldsymbol{X} + c_j^*)}{D_{g,*}^h(\boldsymbol{X})}$$
$$\cdot (\boldsymbol{P}_{V,h}^0 + \sigma_2(\boldsymbol{W}_{2,j}^* \boldsymbol{B}_{h,j}^*)) \sigma_1(\boldsymbol{W}_{1,j}^* \boldsymbol{A}_j^*)) \boldsymbol{X}, \tag{9}$$

where we denote $D_{g,*}^h(\boldsymbol{X}) := \sum_{j=1}^{L} \exp(\boldsymbol{X}^\top (\boldsymbol{P}_{Q,h}^0 + \sigma_2(\boldsymbol{W}_{2,j}^* \boldsymbol{B}_{h,j}^*)) \sigma_1(\boldsymbol{W}_{1,j}^* \boldsymbol{A}_j^*) \boldsymbol{P}_{K,h}^0 \boldsymbol{X} + c_j^*)$, and $\widetilde{G}_* := \sum_{h=1}^{H} \pi_h^* \sum_{j=1}^{L} \exp(c_{j'}^*) \delta_{(\boldsymbol{B}_{h,j}^*, \boldsymbol{A}_j^*)}$, $\boldsymbol{W}_{2,j}^* \in \mathbb{R}^{\bar{d} \times d}$, and $\boldsymbol{W}_{1,j}^* \in \mathbb{R}^{r \times r}$. Due to the change of regression function, the least squares estimator is tailored to this setting as

$$\widetilde{G}_n := \arg \min_{\widetilde{G} \in \mathcal{G}_{H,L'}(\widetilde{\Theta})} \sum_{i=1}^{n} (\boldsymbol{Y}_i - g_{\widetilde{G}}(\boldsymbol{X}))^2,$$

where we define $\mathcal{G}_{H,L'}(\widetilde{\Theta}) = \{\widetilde{G} = \sum_{h=1}^{H} \pi_h \sum_{j=1}^{\ell} \exp(c_j) \delta_{(\boldsymbol{W}_{2,j} \boldsymbol{B}_{h,j}, \boldsymbol{W}_{1,j} \boldsymbol{A}_j)}, \ell \in [L'], (\pi_h, c_j', \boldsymbol{W}_{2,j}, \boldsymbol{B}_{h,j}, \boldsymbol{W}_{1,j}, \boldsymbol{A}_j) \in \widetilde{\Theta}\}$, where $\widetilde{\Theta} \subset [0,1] \times \mathbb{R} \times \mathbb{R}^{\bar{d} \times d} \times \mathbb{R}^{\bar{d} \times r} \times \mathbb{R}^{r \times r} \times \mathbb{R}^{r \times d}$. Furthermore, the Voronoi loss of interest in this setting is given by

$$\mathcal{D}_2(\widetilde{G}, \widetilde{G}_*) := \sum_{h=1}^{H} |\pi_{\tau(h)} - \pi_h^*| + \sum_{h=1}^{H} \pi_{\tau(h)} \sum_{l=1}^{L} \left| \sum_{i \in \mathcal{W}_{l|h}} \exp(c_i) - \exp(c_h^*) \right|$$

$$+ \sum_{h=1}^{H} \pi_{\tau(h)} \sum_{l=1}^{L} \Bigg[ \sum_{i \in \mathcal{W}_{l|h}, |\mathcal{W}_{l|h}|=1} \exp(c_i^*)(\|\Delta(\boldsymbol{W}_2 \boldsymbol{B})_{h,il}\| + \|\Delta(\boldsymbol{W}_1 \boldsymbol{A})_{h,il}\|)$$

$$+ \sum_{i \in \mathcal{W}_{l|h}, |\mathcal{W}_{l|h}|>1} \exp(c_i^*)(\|\Delta(\boldsymbol{W}_2 \boldsymbol{B})_{h,il}\|^2 + \|\Delta(\boldsymbol{W}_1 \boldsymbol{A})_{h,il}\|^2) \Bigg],$$

where $\Delta(\boldsymbol{W}_2 \boldsymbol{B})_{h,il} := \boldsymbol{W}_{2,i} \boldsymbol{B}_{\tau(h),i} - \boldsymbol{W}_{2,l} \boldsymbol{B}_{h,l}^*$ and $\Delta(\boldsymbol{W}_1 \boldsymbol{A})_{h,il} := \boldsymbol{W}_{1,i} \boldsymbol{A}_i - \boldsymbol{W}_{1,l} \boldsymbol{A}_l^*$. Above, the Voronoi cells are defined as $\mathcal{W}_{j|h} = \{i \in [L'] : \|\boldsymbol{H}_{\tau(h),i} - \boldsymbol{H}_{h,j}^*\| \le \|\boldsymbol{H}_{\tau(h),i} - \boldsymbol{H}_{h,l}^*\|, \forall l \ne j\}$, where $\boldsymbol{H} := (\boldsymbol{W}_2 \boldsymbol{B}, \boldsymbol{W}_1 \boldsymbol{A})$. With these ingredients in place, we are now ready to establish the sample complexity of estimating low-rank matrices under the shared structure in Theorem 2.

**Theorem 2.** *Under the shared structure setting in Eq. (9), assume that the activation functions $\sigma_1$ and $\sigma_2$ satisfy the condition in Appendix B.2, then we obtain*

$$\mathcal{D}_2(\widetilde{G}_n, \widetilde{G}_*) = \mathcal{O}_P([\log(n)/n]^{1/2}). \tag{10}$$

The proof of Theorem 2 is in Appendix B.2. The bound in Eq. (10) indicates that the rates for estimating low-rank matrices $\boldsymbol{W}_{1,l} \boldsymbol{A}_l^*$ and $\boldsymbol{W}_{2,l} \boldsymbol{B}_{h,l}^*$ are at the order of $\widetilde{\mathcal{O}}_P(n^{-1/2})$ or $\widetilde{\mathcal{O}}_P(n^{-1/4})$, depending on the cardinality of the corresponding Voronoi cells.

**Sample complexity of estimating low-rank matrices.** As a consequence, the above results imply that achieving estimators of the low-rank matrices with a given error $\epsilon$ requires only a polynomial number of data points of order $\mathcal{O}(\epsilon^{-2})$ or $\mathcal{O}(\epsilon^{-4})$. In contrast to the exponential sample complexity observed in the non-sharing structure, the sharing structure thus attains superior performance in terms of estimating low-rank matrices in the multi-head LoRA.

## 4 HYPER-SHARED LOW-RANK ADAPTATION (HORA)

Motivated by the theoretical developments in Section 3 where the shared structure across attention heads improves the sample complexity of estimating low-rank matrices in multi-head LoRA, this section introduces our practical method known as Hyper-shared Low-rank Adaptation (HoRA).

**Vanilla LoRA.** In the following formulations, we use underscore to highlight the learnable components. Recall that for the $i^{th}$ head in an attention layer, vanilla LoRA fine-tunes the query projection matrices $\boldsymbol{W}_{Q,i} \in \mathbb{R}^{k \times d}$ and the value projection matrices $\boldsymbol{W}_{V,i} \in \mathbb{R}^{k \times d}$ as follows:

$$\boldsymbol{W}_{Q,i}' = \boldsymbol{W}_{Q,i} + \underline{\boldsymbol{B}_{Q,i} \boldsymbol{A}_Q}, \quad \boldsymbol{W}_{V,i}' = \boldsymbol{W}_{V,i} + \underline{\boldsymbol{B}_{V,i} \boldsymbol{A}_V}.$$

**HoRA.** In vanilla LoRA, $\boldsymbol{A}_Q, \boldsymbol{A}_V \in \mathbb{R}^{k \times r}$ are shared among attention heads, while the $\boldsymbol{B}-$adapters $\boldsymbol{B}_{Q,i}, \boldsymbol{B}_{V,i} \in R^{r \times d}$ are separated. In this work, we will encourage shared information among these matrices across different heads. To this end, instead of optimizing these low-rank matrices independently and directly, we propose to generate these matrices with the hypernetworks:

$$\boldsymbol{A}_Q = \sigma_1(\underline{\boldsymbol{W}_{Q,A}\boldsymbol{A}}), \boldsymbol{A}_V = \sigma_1(\underline{\boldsymbol{W}_{V,A}\boldsymbol{A}});$$

$$\boldsymbol{B}_{Q,i} = \underline{\boldsymbol{W}_{Q,B,2}}\sigma_2(\underline{\boldsymbol{W}_{B,1}}\mathrm{LN}(\underline{\boldsymbol{B}_i})), \boldsymbol{B}_{V,i} = \underline{\boldsymbol{W}_{V,B,2}}\sigma_2(\underline{\boldsymbol{W}_{B,1}}\mathrm{LN}(\underline{\boldsymbol{B}_i})).$$

In this formulation, $\boldsymbol{W}_{V,B,1} \in R^{d_{hid} \times d_e}$, while $\boldsymbol{W}_{Q,B,2} \in \mathbb{R}^{(r \times d) \times d_{hid}}$ is the concatenation of $d_{hid}$ low-rank matrices. $\boldsymbol{B}_i \in \mathbb{R}^{d_e}$ is a learnable vector corresponding to the $i^{th}$ head where inputs $\boldsymbol{A}$ are matrices while inputs $\boldsymbol{B}_i$ are embedding vectors. For parameter efficiency, we implemented $\boldsymbol{A}$ as diagonal matrices. The learnable matrices $\boldsymbol{W}_{Q,A,1}$ and $\boldsymbol{W}_{Q,B,2}$ are initialized with Kaiming uniform initialization, while matrices $\boldsymbol{W}_{V,A,1}$ and $\boldsymbol{W}_{V,B,2}$ are initialized with zero initialization. $\sigma_1$ and $\sigma_2$ are the activation functions, which we used the sigmoid functions in our experiments. We also initialize matrices $\boldsymbol{W}_{B,1}$ with Kaiming uniform initialization. Inspired by the phenomenon in Ortiz et al. (2023), instead of directly using as the input $\boldsymbol{B}_i$, we apply a non-learnable normalized layer $\mathrm{LN}(\boldsymbol{x}) = \{\boldsymbol{x} - \mathbb{E}(\boldsymbol{x})\}/\mathrm{Std}(\boldsymbol{x})$ to vector embeddings $\boldsymbol{B}_i = \mathrm{LN}(\boldsymbol{B}_i')$. For a demonstration of HoRA, we refer to Figure 1.

**Benefits of shared structure across attention heads.** We emphasize that the shared statistics are not limited to attention heads; they also extend across the key and value projection matrices. By sharing part of the hypernetwork's structure across heads and across key/value projections, the model captures common adaptation patterns, reducing redundancy and encouraging information sharing. At the same time, the head-specific second layers preserve the flexibility needed for specialization. This structured coupling introduces an implicit regularization effect, which both improves sample efficiency—since gradients from different heads contribute to shaping a shared representation—and reduces the risk of overfitting in low-data settings. Moreover, this parameterization is scalable: as model size and number of heads grow, the shared structure amortizes parameter costs, yielding an efficient and expressive adaptation mechanism.

## 5 EXPERIMENTS

**Experimental Settings.** To evaluate the effectiveness of our method, our experiments span two tasks, including image classification and commonsense reasoning. We compare our method with *Prefix Tuning* (Li & Liang, 2021), *LoRA* (Hu et al., 2022), *DoRA* (Liu et al., 2024), and Adapter (Houlsby et al., 2019a). We also conduct a sample efficiency experiment in Section 5.1 to clarify the efficiency of our design. The experiments were conducted on 1 A100-GPUs. To ensure consistency with the theoretical setting, we conduct experiments by applying low-rank matrices to the query and value matrices at each layer. In addition, we also provide an extended version where these matrices are applied to the proj_up and proj_down matrices under the LLaMA-13B setting in Ablation C.4. More details of hyperparameters are shown in Appendix C.1.

**Image Classification.** We first evaluate our method on image classification using the Vision Transformer (ViT) (Dosovitskiy, 2021) pretrained on ImageNet-21K (Deng et al., 2009). Experiments are conducted on two widely adopted benchmarks: VTAB-1K (Zhai et al., 2019) and FGVC.

The VTAB-1K benchmark contains 19 classification tasks grouped into three categories—Natural, Specialized, and Structured—each with only 1,000 labeled examples for training. As shown in Table 1, HoRA achieves the strongest performance overall, with an average accuracy of 74.4%. Moreover, compared to LoRA, HoRA delivers consistent gains across all domains: $+2.2\%$ on Natural, $+2.1\%$ on Specialized, and $+2.2\%$ on Structured tasks. These results demonstrate the effectiveness of stabilizing training while sharing information among attention heads. Detailed per-dataset results are reported in Appendix C.3.

We next assess performance on the FGVC benchmark, which covers five fine-grained datasets: CUB-200-2011, NABirds, Oxford Flowers, Stanford Dogs, and Stanford Cars. As shown in Table 2, HoRA achieves the highest overall accuracy of 89.96%, outperforming all PEFT baselines as well as full fine-tuning. In particular, HoRA sets new best results on four out of five datasets: CUB-200-2011 (88.6%), NABirds (85.9%), Oxford Flowers (99.2%), and Stanford Dogs (91.0%). On

Stanford Cars, HoRA performs competitively (85.0%), while maintaining the best overall average. Compared to LoRA and DoRA, our method improves the average accuracy by notable margins of +5.2% and +2.8%, respectively.

Together, these results highlight the dual strengths of our approach. On VTAB-1K, HoRA demonstrates superior generalization under data scarcity. On FGVC, it achieves strong fine-grained recognition. Across both settings, HoRA consistently advances the state of the art in PEFT, while introducing only an additional 0.09% learnable parameters relative to the total parameters.

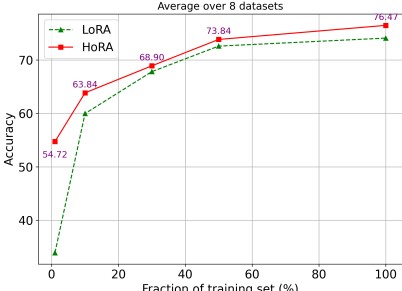

Figure 2: Sample efficiency on the commonsense reasoning datasets.

Table 1: Image Classification results on VTAB-1K.

| Method | #Params. (%) | Natural | Specialized | Structured | AVG |
|--------|-------------|---------|-------------|------------|-----|
| FFT | - | 75.89 | 83.38 | 47.64 | 65.6 |
| LoRA | 0.39 | 79.4 | 84.55 | 59.78 | 72.2 |
| DoRA | 0.40 | 80.33 | 85.15 | 60.11 | 72.8 |
| Adapter | 0.18 | 79.01 | 84.08 | 58.49 | 71.4 |
| Prefix | 0.16 | 77.06 | 82.3 | 52.0 | 67.6 |
| **HoRA** | 0.47 | **81.67** | **86.68** | **61.96** | **74.4** |

Table 2: Image Classification Results on the FGVC Datasets

| Method | #Params (%) | CUB-200 -2011 | NABirds | Oxford Flowers | Stanford Dogs | Stanford Cars | AVG |
|--------|-------------|---------------|---------|----------------|---------------|---------------|-----|
| FFT | - | 87.3 | 82.7 | 98.8 | 89.4 | 84.5 | 88.54 |
| LoRA | 0.55 | 84.6 | 78.2 | 98.9 | 85.1 | 77.1 | 84.78 |
| DoRA | 0.57 | 87.3 | 80.0 | 99.1 | 87.6 | 81.9 | 87.18 |
| Adapter | 0.47 | 87.1 | 84.3 | 98.5 | 89.8 | 68.6 | 85.66 |
| Prefix | 0.42 | 87.5 | 82.0 | 98 | 74.2 | **90.2** | 86.38 |
| **HoRA** | 0.64 | **88.6** | **85.9** | **99.2** | **91.0** | 85.0 | **89.96** |

**Commonsense Reasoning.** We next evaluate its performance in the language domain on commonsense reasoning. This benchmark consists of eight tasks (BoolQ, PIQA, SIQA, HellaSwag, WinoGrande, ARC-e, ARC-c, and OBQA) with predefined training and test splits. All these tasks evaluate the model through multiple-choice questions. Following the protocol of Hu et al. (2023), we combine all tasks into a unified training dataset of approximately 150k examples. Experiments are conducted on LLaMA-7B and LLaMA-13B (Touvron et al., 2023). To ensure fairness, we adopt the same rank of 32 for LoRA, DoRA, and HoRA. As shown in Table 3, HoRA achieves the strongest performance across all tasks and model sizes. On LLaMA-7B, it improves over LoRA and DoRA by +1.7% and +1.0%, respectively, reaching 76.64%. On LLaMA-13B, HoRA attains 80.82% average accuracy, outperforming LoRA by +2.6% and DoRA by +0.6%.

Table 3: Results on the commonsense reasoning task

| Model | Method | #Params. (%) | BoolQ | PIQA | SIQA | HellaSwag | WinoGrande | ARC-e | ARC-c | OBQA | AVG |
|-------|--------|-------------|-------|------|------|-----------|------------|-------|-------|------|-----|
| LLaMA-7B | **Prefix** | 0.11 | 64.3 | 76.8 | **79.3** | 42.1 | 72.1 | 72.9 | 54 | 60.6 | 65.26 |
| | **LoRA** | 0.25 | 67.2 | 79.4 | 76.6 | 78.3 | 78.4 | 77.1 | 61.5 | 74.2 | 74.09 |
| | **DoRA** | 0.25 | 67.22 | 79.98 | 76.66 | 80.66 | 79.72 | 79.5 | 61.01 | 74.8 | 74.94 |
| | **Adapter** | 0.99 | 63 | 79.2 | 76.3 | 67.9 | 75.7 | 74.5 | 57.1 | 72.4 | 70.76 |
| | **HoRA** | 0.28 | **68.59** | **81.5** | 79.07 | **81.42** | **80.51** | **80.01** | **63.82** | **78.2** | **76.64** |
| LLaMA-13B | **Prefix** | 0.03 | 65.3 | 75.4 | 72.1 | 55.2 | 68.6 | 79.5 | 62.9 | 68 | 68.38 |
| | **LoRA** | 0.2 | 71.7 | 82.4 | 79.6 | 90.4 | **83.6** | 83.1 | 68.5 | 82.1 | 80.18 |
| | **DoRA** | 0.2 | 72.2 | 83.19 | **80.81** | 88.92 | 81.93 | 82.95 | **69.37** | 81 | 80.05 |
| | **Adapter** | 0.8 | 71.8 | 83 | 79.2 | 88.1 | 82.4 | 82.5 | 67.3 | 81.8 | 79.51 |
| | **HoRA** | 0.21 | **72.42** | **84.17** | 80.25 | **91.43** | 82.95 | **83.21** | 69.11 | **83** | **80.82** |

## 5.1 SAMPLE EFFICIENCY

In Section 3, we have presented the theoretical benefits of implementing shared statistics among different attention heads to enhance the sample efficiency. In this section, we empirically evaluate this claim by comparing the sample efficiency of HoRA with LoRA on the commonsense reasoning task on the LLaMA-7B setting. Following the approach of d'Ascoli et al. (2021), we subsample

each class at fractions $f = \{1\%, 10\%, 30\%, 50\%, 100\%\}$ and scale the number of training epochs by $1/f$, ensuring the total number of data seen by the model remains constant. The results were presented in Figure 2 and Appendix C.2, where HoRA outperforms LoRA in average. Moreover, this gap is significant in a low-data regime, with the gap of more than 20% when subsampling 1% of the dataset, suggesting an improved sample efficiency of HoRA compared to vanilla LoRA.

## 6 CONCLUSION

We introduce **HoRA**, a parameter-efficient fine-tuning method that addresses the limitations of LoRA. Viewing Multi-head LoRA through the lens of HMoE, HoRA enables parameter sharing across layers. By coupling low-rank matrices via shared structures, HoRA reduces redundancy while preserving flexibility. Our theory establishes stronger generalization guarantees, and our experiments show competitive performance with substantial parameter savings. However, the extent of parameter sharing needs to be chosen carefully as over-sharing can reduce expressiveness and lower performance. Additionally, our current evaluations are limited to transformer-based architectures and do not yet explore other types of models. Future work includes exploring adaptive sharing mechanisms that can dynamically balance efficiency and expressiveness, extending the method to different architectures, and conducting large-scale benchmarks on diverse downstream tasks.

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

# Supplement to "HoRA: Cross-Head Low-Rank Adaptation with Joint Hypernetworks"

In this supplementary material, we review important related work in Appendix A, provide detailed theoretical verification in Appendix B, and present additional experiments in Appendix C to support our proposed mechanisms HoRA. Finally, we discuss the use of large language models in this paper in Appendix D.

## A    RELATED WORKS

**Attention**    The attention mechanism was first introduced to improve the sequence-to-sequence model in machine translation (Bahdanau et al., 2016) by allowing models to dynamically focus on relevant parts of the input. Vaswani et al. (2017) generalized this idea with the Transformer architecture, where scaled dot-product attention becomes the foundation of modern large language models. Since then, attention has been widely adopted across domains, including NLP (Devlin, 2018), computer vision (Dosovitskiy et al., 2021), and multi-model learning (Radford et al., 2019). However, the quadratic complexity of attention in sequence length has motivated research to improve efficiency, such as sparse and low-rank approximation (Kitaev et al., 2020; Wang et al., 2020). These approaches aim to keep the expressiveness of attention while reducing its computational and space complexity, making it more practical for large-scale applications.

**Parameter-efficient Fine-tuning (PEFT) and Low-rank Adaptation (LoRA)**    With the growing size of models, full fine-tuning has become increasingly impractical. Parameter-efficient Fine-Tuning (PEFT) addresses this challenge by adapting models by training a relatively small number of parameters while keeping most pre-trained weights frozen (Houlsby et al., 2019b; Lester et al., 2021). Existing approaches include *Adapter-based*, which insert lightweight modules into the Transformer layer (He et al., 2022), and *Prompt-based*, which add a learnable token to the input (Wei et al., 2023). While effective, these approaches introduce inference latency compared to the original models.

Among the PEFT methods, Low-rank Adaptation (LoRA) (Hu et al., 2022) has emerged as a simple but powerful PEFT method that does not add extra inference burden. LoRA assumes that weight updates during fine-tuning occur in a low-rank subspace and reparameterized weight updates are represented as the product of two low-rank matrices. Since the low-rank components can be merged into the pre-trained weights after training, LoRA doesn't add extra inference cost compared to the original model. Due to its efficiency and strong empirical results, LoRA has become a widely adopted baseline for PEFT in both academic research and real-world applications. In natural language processing, it is employed for tasks such as domain adaptation, instruction tuning, summarization, question answering, and text generation (Huan & Shun, 2025). Moreover, LoRA-based approaches have been extended to areas like federated learning, speech synthesis, and reinforcement learning, supporting scalable model adaptation in distributed and resource-limited settings (Yang et al., 2024; Mei et al., 2024). Recent extensions, such as MTLoRA, have enhanced its applications in multi-task learning for more efficient transfer learning in foundation models Agiza et al. (2024), DoRA (Liu et al., 2024) decomposes weight updates into magnitude and directional components, AdaLoRA (Zhang et al., 2023) dynamically adjusts rank allocation, ReLoRA (Lialin et al., 2023) notably increases the expressivenes of LoRA by proposing a strategy to train high-rank models with low-rank updates, and VeRA (Kopiczko et al., 2024) shares a pair of frozen random matrices across layers with a learnable scaling factor. Recently, Truong et al. (2025b) and Pham et al. (2025) integrated Bayesian inference into LoRA fine-tuning, improving robustness and generalization through uncertainty-aware and distributionally robust adaptation mechanisms. These developments highlight LoRA's role as a foundation for modern PEFT research.

**(Hierachical) Mixture of Experts (HMoE)**    The Mixture of Experts framework (Jacobs et al., 1991) combines multiple experts with a gating function that adaptively assigns inputs to experts. From that foundation, early work (Shazeer et al., 2017) showed that sparsely-gated MoE layers can effectively scale models' capacity by activating only a subset of experts per input. This design has since been applied to large language models (Du et al., 2022), computer vision (Puigcerver et al., 2024), and multi-modal learning (Han et al., 2024), showing strong gains in scalability and efficiency

(Nguyen et al., 2024b;a; 2023). More recently, theoretical work has highlighted the connection between MoE and Multi-head Self-attention (Le et al., 2024; Truong et al., 2025a), motivating new parameter-efficient fine-tuning methods. Hierarchical Mixture of Experts (Jordan & Jacobs, 1994) arranges experts into a tree-like structure with gating occurring at multiple levels. Instead of routing an input directly to an expert, the gating functions make sequential decisions at each level of the hierarchy, which improves training efficiency by narrowing down the relevant subset of experts. As an advanced variant of MoE,it has been shown to handle complex data structures more effectively as well as enhance both generalization and computational efficiency (Peralta & Soto, 2014; Zhao et al., 2019; İrsoy & Alpaydın, 2018) by allowing different branches of the hierarchy to specialize in different regions of the input space.

**HyperNetwork** The HyperNetwork framework (Ha et al., 2016) introduces an approach where the parameters of a target model are not learned directly but are instead generated by an auxiliary neural network, referred to as *HyperNetwork*. Earlier work focused on recurrent neural networks, where HyperNetwork improved generalization and adaptability by producing context-dependent updates (Ha et al., 2016). Later work explores this idea to continual learning, where task-specific weights generated by a HyperNetwork mitigated catastrophic forgetting (von Oswald et al., 2022). In the context of parameter-efficient fine-tuning (PEFT), HyperNetworks have been used to share adaptation across tasks and reduce redundancy. For example, Mahabadi et al. (2021b) proposed using HyperNetwork to generate task-specific adapter weights for multi-task fine-tuning, significantly reducing the number of parameters while maintaining strong performance, or (Li & Liang, 2021) uses HyperNetworks to extend *Prompt tuning* by using Hypernetworks to generate the additional parameters, instead of optimizing those parameters directly. Recently, Truong et al. (2025a) and Nguyen et al. (2024b) have investigated the theoretical benefits of those hypernetworks in different PEFT methods, and have shown that the usage of HyperNetworks is beneficial in enhancing the sample efficiency. Additionally, HyperPEFT (Ding et al., 2024) examines the effectiveness of hypernetworks in parameter-efficient fine-tuning, demonstrating that using a shared hypernetwork promotes knowledge transfer across tasks in pre-trained models, which helps reduce catastrophic forgetting and reveals underlying relationships between tasks. These works have highlighted HyperNetwork as a powerful tool for PEFT that reduces the number of parameters by learning through a lightweight Hypernetwork, improving both efficiency and flexibility.

# B PROOFS OF THEORETICAL RESULTS

## B.1 PROOF OF THEOREM 1

In this section, we present a detailed analysis of Theorem 1.

*Proof of Theorem 1.* The proof of this theorem includes two steps:

**Step 1. The $L^2$ density distance may be small compared to the Voronoi loss.**

In this step, we show that the following limit satisfies for all $r \geq 1$:

$$\lim_{\epsilon \to 0} \inf_{G \in \mathcal{G}_{H,L'}(\widehat{\Theta}):\mathcal{D}_{1,r}(G,G_*) \leq \epsilon} \frac{\|g_G - g_{G_*}\|_{L^2(\mu)}}{\mathcal{D}_{1,r}(G,G_*)} = 0. \tag{11}$$

To demonstrate this inequality, we can construct a sequence of mixing measure such that

$$\lim_{n \to \infty} \mathcal{D}_{1,r}(G_n, G_*) = 0 \text{ and } \lim_{n \to \infty} \|g_G - g_{G_*}\|_{L^2(\mu)}/\mathcal{D}_{1,r}(G_n, G_*) = 0.$$

To prove this, we can consider the sequence

$$G_n = \sum_{h=1}^{H} \pi_h^n \sum_{i=1}^{L+1} \exp(c_i^n) \delta_{\boldsymbol{B}_{Q,h,j}^n, \boldsymbol{A}_{Q,h,j}^n, \boldsymbol{B}_{V,h,j}^n, \boldsymbol{A}_{V,h,j}^n}$$

such that

- $\pi_h^n = \pi_h^*$ for any $1 \leq h \leq H$.

- $\exp(c_1^n) = \exp(c_2^n) = \frac{1}{2}\exp(c_1^*) + \frac{1}{2n^{r+1}}$ and $\exp(c_i^n) = \exp(c_{i-1}^*)$ for any $3 \leq i \leq L+1$.

- $\boldsymbol{B}_{Q,h,1}^n = \boldsymbol{B}_{Q,h,2}^n = \boldsymbol{B}_{Q,h,1}^*$ and $\boldsymbol{B}_{Q,h,i}^n = \boldsymbol{B}_{Q,h,i-1}^*$ for any $3 \leq i \leq L+1$.

- $\boldsymbol{A}_{Q,h,1}^n = \boldsymbol{A}_{Q,h,2}^n = \boldsymbol{A}_{Q,h,1}^*$ and $\boldsymbol{A}_{Q,h,i}^n = \boldsymbol{A}_{Q,h,i-1}^*$ for any $3 \leq i \leq L+1$.

- $\boldsymbol{B}_{V,h,1}^n = \boldsymbol{B}_{V,h,1}^* + n^{-1}e_{11}(\boldsymbol{A}_{V,h,1}^*)^{-1}$, $\boldsymbol{B}_{V,h,2}^n = \boldsymbol{B}_{V,h,1}^* - n^{-1}e_{11}(\boldsymbol{A}_{V,h,1}^*)^{-1}$ and $\boldsymbol{B}_{V,h,i+1}^n = \boldsymbol{B}_{V,h,i}^*$ for any $3 \leq i \leq L+1$.

- $\boldsymbol{A}_{V,h,1}^n = \boldsymbol{A}_{V,h,2}^n = \boldsymbol{A}_{V,h,1}^*$ and $\boldsymbol{A}_{V,h,i}^n = \boldsymbol{A}_{V,h,i-1}^*$ for any $3 \leq i \leq L+1$,

here we denote $e_{11}$ be the matrix that all of its coefficients are equal to 0, except (1,1)-coefficient, which is equal to 1, and, without loss of generality, we assume that $\det(\boldsymbol{A}_{V,1}^*) \neq 0$ (which implies that $\boldsymbol{A}_{V,1}^*$ is invertible). Then, it is evident that $\pi(h) = h$ for all $h \in [H]$. Accordingly, the loss function takes the form

$$\mathcal{D}_{1,r}(G_n, G_*) = \frac{1}{n^{r+1}}\sum_{h=1}^{H}\pi_h + \sum_{h=1}^{H}\pi_h\left[\exp(c_1^*) + \frac{1}{n^{r+1}}\right]\cdot\frac{1}{n^r}\|\boldsymbol{A}_{V,h,1}^*\| = \mathcal{O}(n^{-r}),$$

which implies $\mathcal{D}_{1,r}(G_n, G_*) \to 0$.

Next, we show that $\lim_{n\to\infty}\|g_{G_n} - g_{G_*}\|_{L^2(\mu)}/\mathcal{D}_{1,r}(G_n, G_*) = 0$. Let

$$D_{g,*}^h(x) = \sum_{l=1}^{L}\exp(\boldsymbol{X}^\top(\boldsymbol{P}_{Q,h}^0 + \boldsymbol{B}_{Q,h}^*\boldsymbol{A}_{Q,h}^*)\boldsymbol{P}_{K,h}^0 + c_j^*),$$

$$D_{g,n}^h(x) = \sum_{l=1}^{L}\exp(\boldsymbol{X}^\top(\boldsymbol{P}_{Q,h}^0 + \boldsymbol{B}_{Q,h}^n\boldsymbol{A}_{Q,h}^n)\boldsymbol{P}_{K,h}^0 + c_j^n),$$

we take into account the discrepancy

$$\mathcal{L}_n(\boldsymbol{X}) := g_{G_n}(\boldsymbol{X}) - g_{G_*}(\boldsymbol{X})$$

$$= \sum_{h=1}^{H}\pi_h\Big(\sum_{j=1}^{L}\big(\frac{\exp(\boldsymbol{X}^\top(\boldsymbol{P}_{Q,h}^0 + \boldsymbol{B}_{Q,h,j}^n\boldsymbol{A}_{Q,h,j}^n)\boldsymbol{P}_{K,h}^0\boldsymbol{X} + c_j^n)}{D_{g,n}^h(\boldsymbol{X})}\cdot(\boldsymbol{P}_{V,h}^0 + \boldsymbol{B}_{V,h,j}^n\boldsymbol{A}_{V,h,j}^n)\boldsymbol{X}$$

$$- \frac{\exp(\boldsymbol{X}^\top(\boldsymbol{P}_{Q,h}^0 + \boldsymbol{B}_{Q,h,j}^*\boldsymbol{A}_{Q,h,j}^*)\boldsymbol{P}_{K,h}^0\boldsymbol{X} + c_j^*)}{D_{g,*}^h(\boldsymbol{X})}\cdot(\boldsymbol{P}_{V,h}^0 + \boldsymbol{B}_{V,h,j}^*\boldsymbol{A}_{V,h,j}^*)\boldsymbol{X})\Big)$$

$$:= \sum_{h=1}^{H}\pi_h\tilde{\mathcal{L}}_n^h(\boldsymbol{X}).$$

We examine the decomposition of $\mathcal{L}_n^h(\boldsymbol{X}) = D_{g,*}^h(\boldsymbol{X})\tilde{\mathcal{L}}_n^h(\boldsymbol{X})$:

$$\mathcal{L}_n^h(\boldsymbol{X}) = \sum_{l=1}^{L}\sum_{i\in\mathcal{V}_{l|h}}\exp(c_i^n)\Big[\exp(\boldsymbol{X}^\top(\boldsymbol{P}_{Q,h}^0 + \boldsymbol{B}_{Q,h,j}^n\boldsymbol{A}_{Q,h,j}^n)\boldsymbol{P}_{K,h}^0\boldsymbol{X})(\boldsymbol{P}_{V,h}^0 + \boldsymbol{B}_{V,h,j}^n\boldsymbol{A}_{V,h,j}^n)\boldsymbol{X}$$

$$- \exp(\boldsymbol{X}^\top(\boldsymbol{P}_{Q,h}^0 + \boldsymbol{B}_{Q,h,j}^*\boldsymbol{A}_{Q,h,j}^*)\boldsymbol{P}_{K,h}^0\boldsymbol{X})(\boldsymbol{P}_{V,h}^0 + \boldsymbol{B}_{V,h,j}^*\boldsymbol{A}_{V,h,j}^*)\boldsymbol{X}\Big]$$

$$- \sum_{l=1}^{L}\sum_{i\in\mathcal{V}_{l|h}}\Big[\exp(\boldsymbol{X}^\top(\boldsymbol{P}_{Q,h}^0 + \boldsymbol{B}_{Q,h,j}^n\boldsymbol{A}_{Q,h,j}^n)\boldsymbol{P}_{K,h}^0\boldsymbol{X})) - \exp(\boldsymbol{X}^\top(\boldsymbol{P}_{Q,h}^0 + \boldsymbol{B}_{Q,h,j}^*\boldsymbol{A}_{Q,h,j}^*)\boldsymbol{P}_{K,h}^0\boldsymbol{X}))\Big]g_{G_n}(\boldsymbol{X})$$

$$+ \sum_{l=1}^{L}\left(\sum_{i\in\mathcal{V}_{l|h}}\exp(c_i^n) - \exp(c_l^*)\right)\exp(\boldsymbol{P}_{Q,h}^0 + \boldsymbol{B}_{Q,h,j}^*\boldsymbol{A}_{Q,h,j}^*)\big[(\boldsymbol{P}_{V,h}^0 + \boldsymbol{B}_{V,h,j}^*\boldsymbol{A}_{V,h,j}^*) - g_{G_n}(x)\big]$$

$$:= \mathcal{A}_n^h(\boldsymbol{X}) - \mathcal{B}_n^h(\boldsymbol{X}) + \mathcal{C}_n^h(\boldsymbol{X}).$$

Based on the definition of $\boldsymbol{B}_{Q,h,i}^n, \boldsymbol{A}_{Q,h,i}^n, \boldsymbol{B}_{V,h,i}^n, \boldsymbol{A}_{V,h,i}^n$, we obtain

$$\mathcal{A}_n^h(\boldsymbol{X}) = \sum_{i=1}^2 \frac{1}{2}\left[\exp(c_1^*) + \frac{1}{n^{r+1}}\right]\exp\left(\boldsymbol{X}^\top(\boldsymbol{P}_{Q,h}^0 + \boldsymbol{B}_{Q,h,1}^*\boldsymbol{A}_{Q,h,1}^*)\boldsymbol{P}_{K,h}^0\boldsymbol{X})\right)(\boldsymbol{B}_{V,h,i}^n\boldsymbol{A}_{V,h,i}^n - \boldsymbol{B}_{V,h,1}^*\boldsymbol{A}_{V,h,1}^*)\boldsymbol{X}$$

$$= \frac{1}{2}\left[\exp(b_{*,1}) + \frac{1}{n^{r+1}}\right]\exp\left(\boldsymbol{X}^\top(\boldsymbol{P}_{Q,h}^0 + \boldsymbol{B}_{Q,h,1}^*\boldsymbol{A}_{Q,h,1}^*)\boldsymbol{P}_{K,h}^0\boldsymbol{X})\right)[(\boldsymbol{B}_{V,h,1}^n\boldsymbol{A}_{V,h,1}^n - \boldsymbol{B}_{V,h,1}^*\boldsymbol{A}_{V,h,1}^*)$$

$$+ (\boldsymbol{B}_{V,h,2}^n\boldsymbol{A}_{V,h,2}^n - \boldsymbol{B}_{V,h,2}^*\boldsymbol{A}_{V,h,2}^*)]\boldsymbol{X}$$

$$= 0.$$

The last equality can be justified by $\boldsymbol{B}_{V,h,1}^n\boldsymbol{A}_{V,h,1}^n - \boldsymbol{B}_{V,h,1}^*\boldsymbol{A}_{V,h,1}^* = \frac{1}{n}e_{11}$ and $\boldsymbol{B}_{V,h,2}^n\boldsymbol{A}_{V,h,2}^n - \boldsymbol{B}_{V,h,2}^*\boldsymbol{A}_{V,h,2}^* = -\frac{1}{n}e_{11}$. Also, from the choice that $\boldsymbol{B}_{Q,h,1}^n = \boldsymbol{B}_{Q,h,1}^*$ and $\boldsymbol{A}_{Q,h,1}^n = \boldsymbol{A}_{Q,h,1}^*$, we have $\mathcal{B}_n^h(\boldsymbol{X}) = 0$. In addition, from the value of $c_i^n$ and $c_l^*$, it is straightforward to deduce that $\mathcal{C}_n^h(\boldsymbol{X}) = \mathcal{O}(n^{-(r+1)})$. Combining these results gives us $\mathcal{L}_n^h(\boldsymbol{X})/\mathcal{D}_{1,r}(G_n, G_*) \to 0$. Also noting that the term $D_{g,*}^h(\boldsymbol{X})$ is bounded given that the parameter space $\widehat{\Theta}$ and input space $\mathcal{X}$ are compact, we have $\tilde{\mathcal{L}}_n^h(\boldsymbol{X})/\mathcal{D}_{1,r}(G_n, G_*) \to 0$ for almost every $\boldsymbol{X}$. By summing up these results for $h$, we have $\mathcal{L}_n(\boldsymbol{X})/\mathcal{D}_{1,r}(G_n, G_*) \to 0$ for almost every $\boldsymbol{X}$. This result implies that

$$\|g_{G_n} - g_{G_*}\|_{L^2(\mu)}/\mathcal{D}_{1,r}(G_n, G_*) \to 0,$$

which illustrates Eq. (11).

**Step 2: Apply Le Cam's two-point argument.**

We conclude the proof by showing the minimax property of the estimator

$$\inf_{\bar{G}_n \in \mathcal{G}_{H,L'}(\widehat{\Theta})} \sup_{G \in \mathcal{G}_{H,L'}(\widehat{\Theta})\backslash\mathcal{G}_{H,L-1}(\widehat{\Theta})} \mathbb{E}_{g_G}[\mathcal{D}_{1,r}(\bar{G}_n, G)] \gtrsim n^{-1/2}.$$

Now, Eq. (11) implies that for $\epsilon > 0$ and a fixed constant $c > 0$ determined later, there exists a mixing measures $G'_* \in \mathcal{G}_k(\Theta)$ satisfying $\mathcal{D}_{1,r}(G'_*, G_*) = 2\epsilon$ and $\|g_{G'_*} - g_{G_*}\|_{L^2(\mu)} \le C_1\epsilon$. Using Le Cam's two points argument in Yu (1997) with weak triangle inequality property for the Voronoi loss function $\mathcal{D}_{1,r}$, we have

$$\inf_{\bar{G}_n \in \mathcal{G}_{H,L'}(\widehat{\Theta})} \sup_{G \in \mathcal{G}_{H,L'}(\widehat{\Theta})\backslash\mathcal{G}_{H,L-1}(\widehat{\Theta})} \mathbb{E}_{g_G}[\mathcal{D}_{1,r}(\bar{G}_n, G)]$$

$$\gtrsim \frac{\mathcal{D}_{1,r}(G'_*, G_*)}{8}\exp\left(-n\mathbb{E}_{\boldsymbol{X}\sim\mu}[\mathrm{KL}(\mathcal{N}(g_{G'_*}(\boldsymbol{X}), \sigma^2 I_{\bar{d}}), \mathcal{N}(g_{G'_*}(\boldsymbol{X}), \sigma^2 I_{\bar{d}}))]\right).$$

Bearing in mind that the KL divergence between two Gaussian distributions can be calculated as

$$\mathrm{KL}(\mathcal{N}(g_{G'_*}(\boldsymbol{X}), \sigma^2 I_{\bar{d}}), \mathcal{N}(g_{G_*}(\boldsymbol{X}), \sigma^2 I_{\bar{d}})) = \frac{\|g_{G'_*}(\boldsymbol{X}) - g_{G_*}(\boldsymbol{X})\|^2}{2\sigma^2}.$$

As a result, we have

$$\inf_{\bar{G}_n \in \mathcal{G}_{H,L'}(\widehat{\Theta})} \sup_{G \in \mathcal{G}_{H,L'}(\widehat{\Theta})\backslash\mathcal{G}_{H,L-1}(\widehat{\Theta})} \mathbb{E}_{g_G}[\mathcal{D}_{1,r}(\bar{G}_n, G)]$$

$$\gtrsim \epsilon \cdot \exp(-n\|g_{G'_*} - g_{G_*}\|_{L^2(\mu)}^2)$$

$$\gtrsim \epsilon \cdot \exp(-C_1 n\epsilon^2), \tag{12}$$

Here, we choose $\epsilon = n^{-1/2}$, it follows that $\epsilon \cdot \exp(-C_1 n\epsilon^2) = n^{-1/2}\exp(-C_1)$. Consequently, the minimax lower bound in equation Eq. (11) is attained, thereby completing the proof. $\square$

### B.2 PROOF OF THEOREM 2

Before delving into the details of the proof, it is important to note that the analysis can be reduced to the case where both $\boldsymbol{W}_{1,j}$ and $\boldsymbol{W}_{2,j}$ are identity matrices for each $j$. Consequently, we may assume without loss of generality that $\sigma_2(\boldsymbol{W}_{2,j}^*\boldsymbol{B}_{h,j}^*) = \sigma_2(\boldsymbol{B}_{h,j}^*)$ and $\sigma_1(\boldsymbol{W}_{1,j}^*\boldsymbol{A}_j^*) = \sigma_1(\boldsymbol{A}_j^*)$. The central ingredient of the proof is the model convergence property, namely that the estimator $g_{\widetilde{G}_n}$ converges to $g_{G_*}$ at a rate of order $\mathcal{O}([\log(n)/n]^{1/2})$.

**Proposition 1** (Model convergence). *Given the least square estimator $\widetilde{G}_n$, the convergence rate of the regression function estimation $g_{\widetilde{G}_n}$ to the true regression function $g_{\widetilde{G}_*}$ under the $L^2(\mu)$ is parameteric on the sample size, i.e.*

$$\|g_{\widetilde{G}_n} - g_{\widetilde{G}_*}\|_{L^2(\mu)} = \mathcal{O}(\sqrt{\log(n)/n}). \tag{13}$$

Although the proof of this result is presented later, it is worth noting that, as Eq. (13) is established, we leverage the model convergence result to derive parameter convergence, employing a Taylor expansion for the local analysis and applying Fatou's lemma for the global analysis.

**Assumption.** We impose the following distinguishability assumptions on the two functions.

*(A.1) (Algebraic Independence)* If there exists two couples of parameter matrices $(\boldsymbol{B}, \boldsymbol{A})$ and $(\tilde{\boldsymbol{B}}, \tilde{\boldsymbol{A}})$ such that

$$\sigma_2(\boldsymbol{B})\sigma_1(\boldsymbol{A}) = \sigma_2(\tilde{\boldsymbol{B}})\sigma_1(\tilde{\boldsymbol{A}}),$$

then it follows that $\boldsymbol{B} = \tilde{\boldsymbol{B}}$ and $\boldsymbol{A} = \tilde{\boldsymbol{A}}$.

*(A.2) (Uniform Lipschitz)* Consider

$$\boldsymbol{F}(\boldsymbol{X}, \boldsymbol{A}, \boldsymbol{B}) := \exp(\boldsymbol{X}^\top(\boldsymbol{P}_Q^0 + \sigma_2(\boldsymbol{B})\sigma_1(\boldsymbol{A}))\boldsymbol{X})(\boldsymbol{P}_V^0 + \sigma_2(\boldsymbol{B})\sigma_1(\boldsymbol{A}))\boldsymbol{X},$$

then for any $\eta \in \{1, 2\}$ and index $\beta = (\beta_1, \beta_2) \in \mathbb{N}^{r \times \bar{d}} \times \mathbb{N}^{\bar{d} \times r}$

$$\sum_{|\alpha|=\eta} \left\| \left( \frac{\partial^{|\beta|}\boldsymbol{F}}{\partial \boldsymbol{A}^{\beta_1}\partial \boldsymbol{B}^{\beta_2}}(\boldsymbol{X}, \boldsymbol{A}, \boldsymbol{B}) - \frac{\partial^{|\beta|}\boldsymbol{F}}{\partial \boldsymbol{A}^{\beta_1}\partial \boldsymbol{B}^{\beta_2}}(\boldsymbol{X}, \boldsymbol{A}', \boldsymbol{B}') \right) \gamma^\beta \right\| \leq C\|(\boldsymbol{A}, \boldsymbol{B}) - (\boldsymbol{A}', \boldsymbol{B}')\|^\xi \|\gamma\|^\eta,$$

for any vector $\gamma \in \mathbb{R}^{2\bar{d}r}$, and for some positive constants $\xi$ and $C$ that are independent of the input $\boldsymbol{X}$ and the parameters $\boldsymbol{A}, \boldsymbol{B}$.

*(A.3) (Strong identifiability)* For any non-negative integer $\ell \geq 0$ and any collection of distinct parameter matrices $\{(\boldsymbol{B}_j, \boldsymbol{A}_j)\}_{j \in [\ell]}$, the functions in the set below are almost surely independent in $\boldsymbol{X}$:

$$\begin{aligned}
\Big\{ \ &\boldsymbol{X}^{(u)}, \ \boldsymbol{X}^{(u)}\boldsymbol{X}^\top\sigma_2(\mathbf{B}_j), \ \boldsymbol{X}^{(u)}\sigma_1(\mathbf{A}_j)\boldsymbol{X}, \ \boldsymbol{X}^\top\sigma_2(\mathbf{B}_j), \ \sigma_1(\mathbf{A}_j)\boldsymbol{X}, \\
&\boldsymbol{X}^{(u)}\boldsymbol{X}^{(v)}, \ \boldsymbol{X}^{(u)}\boldsymbol{X}^{(v)}[\boldsymbol{X}^\top\sigma_2(\mathbf{B}_j)]^2, \ \boldsymbol{X}^{(u)}\boldsymbol{X}^{(v)}[\sigma_1(\mathbf{A}_j)\boldsymbol{X}]^2, \\
&\boldsymbol{X}^{(u)}\boldsymbol{X}^{(v)}\boldsymbol{X}^\top\sigma_2(\mathbf{B}_j)\sigma_1(\mathbf{A}_j)\boldsymbol{X} \ : \ j \in [\ell], \ u, v \in [d] \Big\}
\end{aligned}$$

*Return to the proof of Theorem 2.* Through a permutation, without loss of generality, we can suppose that $\tau(h) = h$ for all $h \in [H]$. The focus of this argument is to establish the following inequality:

$$\inf_{\widetilde{G} \in \mathcal{G}_{H,L'}(\widetilde{\Theta})} \|g_{\widetilde{G}} - g_{\widetilde{G}^*}\|_{L^2(\mu)}/\mathcal{D}_2(\widetilde{G}, G^*) > 0. \tag{14}$$

We can divide our demonstration into two parts. The first part, namely *local part*, is to establish Eq. (14) when $\mathcal{D}_2(\widetilde{G}, \widetilde{G}^*)$ is small enough

$$\lim_{\epsilon \to 0} \inf_{G \in \mathcal{G}_{H,L'}(\widetilde{\Theta}):\mathcal{D}_2(\widetilde{G},\widetilde{G}^*)\leq \epsilon} \|g_{\widetilde{G}} - g_{\widetilde{G}^*}\|_{L^2(\mu)}/\mathcal{D}_2(\widetilde{G}, \widetilde{G}^*) > 0. \tag{15}$$

The Taylor expansion is the main tool used to resolve this problem in the local regime. The *global part* of the proof concerns the behavior of this property when $\mathcal{D}_2(\widetilde{G}, \widetilde{G}^*)$ is sufficiently large.

$$\inf_{\widetilde{G} \in \mathcal{G}_{H,L'}(\widetilde{\Theta}):\mathcal{D}_2(\widetilde{G},\widetilde{G}^*)>\epsilon} \|g_{\widetilde{G}} - g_{\widetilde{G}^*}\|_{L^2(\mu)}/\mathcal{D}_2(\widetilde{G}, \widetilde{G}^*) > 0.$$

**Proof of local part Eq. (15)**

Suppose that Eq. (15) does not hold, i.e.

$$\lim_{\epsilon \to 0} \inf_{\widetilde{G} \in \mathcal{G}_{H,L'}(\widetilde{\Theta}): \mathcal{D}_2(\widetilde{G}, \widetilde{G}^*) \leq \epsilon} \|g_{\widetilde{G}} - g_{\widetilde{G}^*}\|_{L^2(\mu)} / \mathcal{D}_2(\widetilde{G}, \widetilde{G}^*) = 0.$$

Denote

$$g^h_{\widetilde{G}_n}(\boldsymbol{X}) = \sum_{j=1}^{L} \frac{\exp(\boldsymbol{X}^\top (\boldsymbol{P}^0_{Q,h} + \boldsymbol{B}^n_{h,j} \boldsymbol{A}^n_{h,j}) \boldsymbol{P}^0_{K,h} \boldsymbol{X} + c^n_j)}{D^h_{g,n}(\boldsymbol{X})} \cdot (\boldsymbol{P}^0_{V,h} + \boldsymbol{B}^n_{h,j} \boldsymbol{A}^n_{h,j}) \boldsymbol{X},$$

$$g^h_{\widetilde{G}_*}(\boldsymbol{X}) = \sum_{j=1}^{L} \frac{\exp(\boldsymbol{X}^\top (\boldsymbol{P}^0_{Q,h} + \boldsymbol{B}^*_{h,j} \boldsymbol{A}^*_{h,j}) \boldsymbol{P}^0_{K,h} \boldsymbol{X} + c_j)}{D^h_{g,*}(\boldsymbol{X})} \cdot (\boldsymbol{P}^0_{V,h} + \boldsymbol{B}^*_{h,j} \boldsymbol{A}^*_{h,j}) \boldsymbol{X},$$

where

$$D^h_{g,*}(\boldsymbol{X}) = \sum_{l=1}^{L} \exp(\boldsymbol{X}^\top (\boldsymbol{P}^0_{Q,h} + \boldsymbol{B}^*_{h,j} \boldsymbol{A}^*_{h,j}) \boldsymbol{P}^0_{K,h} + c^*_j),$$

$$D^h_{g,n}(\boldsymbol{X}) = \sum_{l=1}^{L} \exp(\boldsymbol{X}^\top (\boldsymbol{P}^0_{Q,h} + \boldsymbol{B}^n_{h,j} \boldsymbol{A}^n_{h,j}) \boldsymbol{P}^0_{K,h} + c^n_j).$$

Then, we have

$$g_{\widetilde{G}_n}(\boldsymbol{X}) = \sum_{h=1}^{H} \pi^n_h g^h_{\widetilde{G}_n}(\boldsymbol{X}), \quad g_{\widetilde{G}_*}(\boldsymbol{X}) = \sum_{h=1}^{H} \pi^*_h g^h_{\widetilde{G}_*}(\boldsymbol{X}).$$

**Step 1 - Decomposition the discrepancy between regression functions.**

The first step of this proof includes decompose the quantity $g_{\widetilde{G}_n}(\boldsymbol{X}) - g_{\widetilde{G}^*}(\boldsymbol{X})$ using Taylor expansion. Recall that

$$\begin{aligned}
\mathcal{L}_n(\boldsymbol{X}) &:= g_{\widetilde{G}_n}(\boldsymbol{X}) - g_{\widetilde{G}_*}(\boldsymbol{X}) \\
&= \sum_{h=1}^{H} \pi^n_h g^h_{\widetilde{G}_n}(\boldsymbol{X}) - \sum_{h=1}^{H} \pi^*_h g^h_{\widetilde{G}_*}(\boldsymbol{X}) \\
&= \sum_{h=1}^{H} \pi^n_h (g^h_{\widetilde{G}_n}(\boldsymbol{X}) - g^h_{\widetilde{G}_*}(\boldsymbol{X})) + \sum_{h=1}^{H} (\pi^n_h - \pi^*_h) g^h_{\widetilde{G}_*}(\boldsymbol{X}) \\
&:= \sum_{h=1}^{H} \pi^n_h \tilde{\mathcal{L}}^h_n(\boldsymbol{X}) + \sum_{h=1}^{H} (\pi^n_h - \pi^*_h) g^h_{\widetilde{G}_*}(\boldsymbol{X}),
\end{aligned}$$

where $\tilde{\mathcal{L}}^h_n(\boldsymbol{X}) = g^h_{\widetilde{G}_n}(\boldsymbol{X}) - g^h_{\widetilde{G}_*}(\boldsymbol{X})$.

Each term $\mathcal{L}^h_n(\boldsymbol{X}) = D^h_{g,*}(\boldsymbol{X}) \tilde{\mathcal{L}}^h_n(\boldsymbol{X})$ can be decomposed as

$$\begin{aligned}
\mathcal{L}^h_n(\boldsymbol{X}) = {}& \sum_{j=1}^{L} \sum_{i \in \mathcal{W}_{j|h}} \exp(c_{n,i}) \Big[ \exp(\boldsymbol{X}^\top (\boldsymbol{P}^0_{Q,h} + \sigma_2(\boldsymbol{B}^n_{h,i}) \sigma_1(\boldsymbol{A}^n_i)) \boldsymbol{X})(\boldsymbol{P}^0_V + \sigma_2(\boldsymbol{B}^n_{h,i}) \sigma_1(\boldsymbol{A}^n_i)) \\
& - \exp(\boldsymbol{X}^\top (\boldsymbol{P}^0_{Q,h} + \sigma_2(\boldsymbol{B}^*_{h,j}) \sigma_1(\boldsymbol{A}^*_j)) \boldsymbol{X})(\boldsymbol{P}^0_V + \sigma_2(\boldsymbol{B}^*_{h,j}) \sigma_1(\boldsymbol{A}^*_j)) \Big] \\
& - \sum_{j=1}^{L} \sum_{i \in \mathcal{W}_{j|h}} \exp(c_{n,i}) \Big[ \exp(\boldsymbol{X}^\top (\boldsymbol{P}^0_{Q,h} + \sigma_2(\boldsymbol{B}^n_{h,i}) \sigma_1(\boldsymbol{A}^n_s i)) \boldsymbol{X}) \hspace{1cm} (16) \\
& - \exp(\boldsymbol{X}^\top (\boldsymbol{P}^0_{Q,h} + \sigma_2(\boldsymbol{B}^*_{h,i}) \sigma_1(\boldsymbol{A}^*_i)) \boldsymbol{X}) \Big] g^h_{G_n}(\boldsymbol{X}) \\
& + \sum_{j=1}^{L} \Big( \sum_{i \in \mathcal{W}_{j|h}} \exp(c_{n,i}) - \exp(c^*_j) \Big) \exp(\boldsymbol{X}^\top (\boldsymbol{P}^0_{Q,h} + \sigma_2(\boldsymbol{B}^*_{h,j}) \sigma_1(\boldsymbol{A}^*_j)) \boldsymbol{X}) \\
& := \mathcal{A}^h_n(\boldsymbol{X}) - \mathcal{B}^h_n(\boldsymbol{X}) + \mathcal{C}^h_n(\boldsymbol{X}). \hspace{2cm} (17)
\end{aligned}$$

**Decomposition for the function $\mathcal{A}_n^h(\boldsymbol{X})$.** Let

$$\boldsymbol{R}(\boldsymbol{X}; \boldsymbol{B}, \boldsymbol{A}) = \exp(\boldsymbol{X}^\top(\boldsymbol{P}_Q^0 + \sigma_2(\boldsymbol{B})\sigma_1(\boldsymbol{A}))\boldsymbol{X}),$$

$$\boldsymbol{S}(\boldsymbol{X}; \boldsymbol{B}, \boldsymbol{A}) = (\boldsymbol{P}_V^0 + \sigma_2(\boldsymbol{B})\sigma_1(\boldsymbol{A}))\boldsymbol{X},$$

$$\boldsymbol{G}(\boldsymbol{X}; \boldsymbol{B}, \boldsymbol{A}) = \boldsymbol{R}(\boldsymbol{X}; \boldsymbol{B}, \boldsymbol{A})\boldsymbol{S}(\boldsymbol{X}; \boldsymbol{B}, \boldsymbol{A}).$$

Our term $\mathcal{A}_n^h$ can be decomposed based on the number of element in each Voronoi cells

$$\mathcal{A}_n^h = \sum_{j:|\mathcal{W}_{j|h}|=1} \sum_{i \in \mathcal{A}_{j|u,h}} \exp(c_{n,i})[\boldsymbol{G}(\boldsymbol{X}; \boldsymbol{B}_{h,i}^n, \boldsymbol{A}_i^n) - \boldsymbol{G}(\boldsymbol{X}; \boldsymbol{B}_{h,j}^*, \boldsymbol{A}_j^*)]$$

$$+ \sum_{j:|\mathcal{W}_{j|h}|>1} \sum_{i \in \mathcal{W}_{j|h}} \exp(c_{n,i})[\boldsymbol{G}(\boldsymbol{X}; \boldsymbol{B}_{h,i}^n, \boldsymbol{A}_i^n) - \boldsymbol{G}(\boldsymbol{X}; \boldsymbol{B}_{h,j}^*, \boldsymbol{A}_j^*)]$$

$$:= \mathcal{A}_{n,1}^h + \mathcal{A}_{n,2}^h.$$

Using the first-order Taylor expansion, we have

$$\boldsymbol{R}(\boldsymbol{X}; \boldsymbol{B}_{h,i}^n, \boldsymbol{A}_i^n) = \boldsymbol{R}(\boldsymbol{X}; \boldsymbol{B}_{h,j}^*, \boldsymbol{A}_j^*) + \sum_{|\alpha|=1} (\Delta \boldsymbol{A}_{n,ij})^{\alpha_1}(\Delta \boldsymbol{B}_{n,ij}^h)^{\alpha_2} \frac{\partial^{|\alpha|}\boldsymbol{R}}{\partial \boldsymbol{A}^{\alpha_1}\partial \boldsymbol{B}^{\alpha_2}}(\boldsymbol{X}; \boldsymbol{B}_{h,j}^*, \boldsymbol{A}_j^*) + \mathcal{R}_{ij,1}(\boldsymbol{X}),$$

$$\boldsymbol{S}(\boldsymbol{X}; \boldsymbol{B}_{n,i}^n, \boldsymbol{A}_{n,i}) = \boldsymbol{S}(\boldsymbol{X}; \boldsymbol{B}_{h,j}^*, \boldsymbol{A}_j^*) + \sum_{|\alpha|=1} (\Delta \boldsymbol{A}_{n,ij})^{\alpha_1}(\Delta \boldsymbol{B}_{n,ij}^h)^{\alpha_2} \frac{\partial^{|\alpha|}\boldsymbol{S}}{\partial \boldsymbol{A}^{\alpha_1}\partial \boldsymbol{B}^{\alpha_2}}(\boldsymbol{X}; \boldsymbol{B}_{h,j}^*, \boldsymbol{A}_j^*) + \mathcal{R}_{ij,2}(\boldsymbol{X}),$$

for any $i$ and $j$ satisfying $i \in \mathcal{W}_{j|h}$ and $\mathcal{W}_{j|h} = 1$. In the formulas above, $\mathcal{R}_{ij,1}(\boldsymbol{X})$ and $\mathcal{R}_{ij,2}(\boldsymbol{X})$ denote the Taylor expansion remainder. The results above gives us

$$\mathcal{A}_{n,1}^h(\boldsymbol{X}) = \sum_{j:|\mathcal{W}_{j|h}|=1} \sum_{i \in \mathcal{W}_{j|h}} \frac{\exp(c_{n,i})}{\alpha!} \sum_{|\alpha|=1} \Big\{ (\Delta \boldsymbol{A}_{n,ij})^{\alpha_1}(\Delta \boldsymbol{B}_{n,ij}^h)^{\alpha_1} \frac{\partial^\alpha \boldsymbol{R}}{\partial \boldsymbol{A}^{\alpha_1}\boldsymbol{B}^{\alpha_2}}(\boldsymbol{X}; \boldsymbol{B}_{h,j}^*, \boldsymbol{A}_j^*)\boldsymbol{S}(\boldsymbol{X}; \boldsymbol{B}_{h,j}^*\boldsymbol{A}_j^*)$$

$$+ (\Delta \boldsymbol{A}_{n,ij})^{\alpha_1}(\Delta \boldsymbol{B}_{n,ij}^h)^{\alpha_1} \boldsymbol{R}(\boldsymbol{X}; \boldsymbol{B}_{h,j}^*\boldsymbol{A}_j^*)\frac{\partial^\alpha \boldsymbol{S}}{\partial \boldsymbol{A}^{\alpha_1}\boldsymbol{B}^{\alpha_2}}(\boldsymbol{X}; \boldsymbol{B}_{h,j}^*\boldsymbol{A}_j^*) \Big\} + \mathcal{R}_{n,1}^h(\boldsymbol{X})$$

$$= \sum_{j:|\mathcal{W}_{j|h}|=1} \sum_{|\alpha|=1} \Big\{ \bar{U}_{n,j,\alpha}^h \frac{\partial^{|\alpha|}\boldsymbol{R}}{\partial \boldsymbol{A}^{\alpha_1}\boldsymbol{B}^{\alpha_2}}(\boldsymbol{X}; \boldsymbol{B}_{h,j}^*\boldsymbol{A}_j^*)\boldsymbol{S}(\boldsymbol{X}; \boldsymbol{B}_{h,j}^*\boldsymbol{A}_j^*)$$

$$+ \bar{U}_{n,j,\alpha}^h \boldsymbol{R}(\boldsymbol{X}; \boldsymbol{B}_{h,j}^*\boldsymbol{A}_j^*)\frac{\partial^\alpha \boldsymbol{S}}{\partial \boldsymbol{A}^{\alpha_1}\boldsymbol{B}^{\alpha_2}}(\boldsymbol{X}; \boldsymbol{B}_{h,j}^*\boldsymbol{A}_j^*) \Big\} + \mathcal{R}_{n,1}^h,$$

where the reminder is small compared with the loss function $\mathcal{R}_{n,1}^h/\mathcal{D}_2(G^n, G_*)$, which is due to the uniform Lipschitz property of function $G$. Here, the coefficients $\bar{U}_{n,j,\alpha}^h$ are defined as

$$\bar{U}_{n,j,\alpha_1,\alpha_2}^h = \sum_{i \in \mathcal{W}_{j|h}} \frac{\exp(c_{n,i})}{\alpha!}(\Delta \boldsymbol{A}_{n,ij})^{\alpha_1}(\Delta \boldsymbol{B}_{n,ij}^h)^{\alpha_2}, \forall \alpha : |\alpha| = 1.$$

For $\mathcal{A}_{n,2}^h$, using the Taylor expansion up to second order, we have

$$\mathcal{A}_{n,2}^h = \sum_{j:|\mathcal{A}_j|>1} \sum_{1 \le |\alpha| \le 2} \Big\{ \bar{U}_{n,j,\alpha_1,\alpha_2}^h \frac{\partial^\alpha \boldsymbol{R}}{\partial \boldsymbol{A}^{\alpha_1}\boldsymbol{B}^{\alpha_2}}(\boldsymbol{X}; \boldsymbol{B}_{h,j}^*, \boldsymbol{A}_j^*)\boldsymbol{S}(\boldsymbol{X}; \boldsymbol{B}_{h,j}^*\boldsymbol{A}_j^*)$$

$$+ \Big\{ \bar{U}_{n,j,\alpha_1,\alpha_2}^h \boldsymbol{R}(\boldsymbol{X}; \boldsymbol{B}_{h,j}^*\boldsymbol{A}_j^*)\frac{\partial^\alpha \boldsymbol{S}}{\partial \boldsymbol{A}^{\alpha_1}\boldsymbol{B}^{\alpha_2}}(\boldsymbol{X}; \boldsymbol{B}_{h,j}^*, \boldsymbol{A}_j^*) \Big\}$$

$$+ \sum_{|\alpha|=1,|\beta|=1} \bar{U}_{n,j,\alpha_1,\beta_1,\alpha_2,\beta_2}^h \frac{\partial^{|\alpha|}\boldsymbol{R}}{\partial \boldsymbol{A}^{\alpha_1}\partial \boldsymbol{B}^{\alpha_2}}(\boldsymbol{X}; \boldsymbol{B}_{h,j}^*, \boldsymbol{A}_j^*)\frac{\partial^{|\beta|}\boldsymbol{S}}{\partial \boldsymbol{A}^{\beta_1}\partial \boldsymbol{B}^{\beta_2}}(\boldsymbol{X}; \boldsymbol{B}_{h,j}^*, \boldsymbol{A}_j^*) + \mathcal{R}_{n,2}(\boldsymbol{X}),$$

where the remainder $\mathcal{R}_{n,2}(\boldsymbol{X})$ is small compared with $\mathcal{D}_2(G^n, G_*)$: $\mathcal{R}_{n,2}(\boldsymbol{X})/\mathcal{D}_2(G^n, G_*) \to 0$. Here, the coefficients take the following forms:

$$\bar{U}_{n,j,\alpha_1,\alpha_2}^h = \sum_{i \in \mathcal{W}_{j|h}} \frac{\exp(c_{n,i})}{\alpha!}(\Delta \boldsymbol{A}_{n,ij})^{\alpha_1}(\Delta \boldsymbol{B}_{n,ij}^h)^{\alpha_2}, \forall |\alpha| = 2$$

$$\bar{U}_{n,j,\alpha_1,\beta_1,\alpha_2,\beta_2}^h = \sum_{i \in \mathcal{W}_{j|h}} \frac{\exp(c_{n,i})}{\alpha!\beta!}(\Delta \boldsymbol{A}_{n,ij})^{\alpha_1+\beta_1}(\Delta \boldsymbol{B}_{n,ij}^h)^{\alpha_2+\beta_2}, \forall |\alpha| = |\beta| = 1.$$

Simple calculation gives us the following formulation of the partial derivative of $\boldsymbol{R}(\boldsymbol{X}; \boldsymbol{B}, \boldsymbol{A})$ and $\boldsymbol{S}(\boldsymbol{X}; \boldsymbol{B}, \boldsymbol{A})$:

$$\frac{\partial \boldsymbol{R}}{\partial \boldsymbol{A}^{(u)}}(\boldsymbol{X}; \boldsymbol{B}, \boldsymbol{A}) = \boldsymbol{X}^{(u)}\sigma_1'(\boldsymbol{A}^{(u)})\boldsymbol{X}^\top\sigma_2(\boldsymbol{B})\exp(\boldsymbol{X}^\top(\boldsymbol{P}_Q^0 + \sigma_2(\boldsymbol{B})\sigma_1(\boldsymbol{A}))),$$

$$\frac{\partial \boldsymbol{R}}{\partial \boldsymbol{B}(u)}(\boldsymbol{X}; \boldsymbol{B}, \boldsymbol{A}) = \boldsymbol{X}^{(u)}\sigma_1(\boldsymbol{A}^{(u)})\boldsymbol{X}^\top\sigma_2'(\boldsymbol{B})\exp(\boldsymbol{X}^\top(\boldsymbol{P}_Q^0 + \sigma_2(\boldsymbol{B})\sigma_1(\boldsymbol{A})))$$

$$\frac{\partial^2 \boldsymbol{R}}{\partial \boldsymbol{A}^{(u)}\partial \boldsymbol{A}^{(v)}}(\boldsymbol{X}; \boldsymbol{B}, \boldsymbol{A}) = \Big[\boldsymbol{X}^{(u)}\boldsymbol{X}^{(v)}\sigma_1'(\boldsymbol{A}^{(u)})\sigma_1'(\boldsymbol{A}^{(v)})(\boldsymbol{X}^\top\sigma_2(\boldsymbol{B}))^2 + \mathbf{1}_{u=v}\boldsymbol{X}^{(u)}\sigma_1''(\boldsymbol{A}^{(u)})\boldsymbol{X}^\top\sigma_2(\boldsymbol{B})\Big]$$
$$\times \exp(\boldsymbol{X}^\top(\boldsymbol{P}_Q^0 + \sigma_2(\boldsymbol{B})\sigma_1(\boldsymbol{A})))$$

$$\frac{\partial^2 \boldsymbol{R}}{\partial \boldsymbol{B}^{(u)}\partial \boldsymbol{B}^{(v)}}(\boldsymbol{X}; \boldsymbol{B}, \boldsymbol{A}) = \Big[\boldsymbol{X}^{(u)}\boldsymbol{X}^{(v)}\sigma_2'(\boldsymbol{B}^{(u)})\sigma_2'(\boldsymbol{B}^{(v)})(\boldsymbol{X}^\top\sigma_2(\boldsymbol{A}))^2 + \mathbf{1}_{u=v}\boldsymbol{X}^{(u)}\sigma_2''(\boldsymbol{B}^{(u)})\boldsymbol{X}^\top\sigma_2(\boldsymbol{B})\Big]$$
$$\times \exp(\boldsymbol{X}^\top(\boldsymbol{P}_Q^0 + \sigma_2(\boldsymbol{B})\sigma_1(\boldsymbol{A})))$$

$$\frac{\partial^2 \boldsymbol{R}}{\partial \boldsymbol{A}^{(u)}\partial \boldsymbol{B}^{(v)}}(\boldsymbol{X}; \boldsymbol{B}, \boldsymbol{A}) = \Big[\boldsymbol{X}^{(u)}\boldsymbol{X}^{(v)}\sigma_1'(\boldsymbol{A}^{(u)})\sigma_2'(\boldsymbol{B}^{(v)}) + \boldsymbol{X}^{(u)}\sigma_1'(\boldsymbol{B}^{(u)})\boldsymbol{X}^\top\sigma_2(\boldsymbol{B})\Big]$$
$$\times \exp(\boldsymbol{X}^\top(\boldsymbol{P}_Q^0 + \sigma_2(\boldsymbol{B})\sigma_1(\boldsymbol{A}))\boldsymbol{X}^{(v)})$$

$$\frac{\partial \boldsymbol{S}}{\partial \boldsymbol{A}^{(u)}}(\boldsymbol{X}; \boldsymbol{B}, \boldsymbol{A}) = \boldsymbol{X}^{(u)}\sigma_1'(\boldsymbol{A})\sigma_2(\boldsymbol{B})$$

$$\frac{\partial \boldsymbol{S}}{\partial \boldsymbol{B}^{(u)}}(\boldsymbol{X}; \boldsymbol{B}, \boldsymbol{A}) = \boldsymbol{X}^{(u)}\sigma_1(\boldsymbol{A})\sigma_2'(\boldsymbol{B})$$

$$\frac{\partial^2 \boldsymbol{S}}{\partial \boldsymbol{A}^{(u)}\partial \boldsymbol{A}^{(v)}}(\boldsymbol{X}; \boldsymbol{B}, \boldsymbol{A}) = \mathbf{1}_{u=v}\boldsymbol{X}^{(u)}\sigma_1''(\boldsymbol{A})\sigma_2(\boldsymbol{B})$$

$$\frac{\partial^2 \boldsymbol{S}}{\partial \boldsymbol{B}^{(u)}\partial \boldsymbol{B}^{(v)}}(\boldsymbol{X}; \boldsymbol{B}, \boldsymbol{A}) = \mathbf{1}_{u=v}\boldsymbol{X}^{(u)}\sigma_1(\boldsymbol{A})\sigma_2''(\boldsymbol{B})$$

$$\frac{\partial^2 \boldsymbol{S}}{\partial \boldsymbol{A}^{(u)}\partial \boldsymbol{B}^{(v)}}(\boldsymbol{X}; \boldsymbol{B}, \boldsymbol{A}) = \mathbf{1}_{u=v}\boldsymbol{X}^{(u)}\sigma_1'(\boldsymbol{A})\sigma_2'(\boldsymbol{B})$$

Plugging these formulations into the functions $\mathcal{A}_{n,1}^h(\boldsymbol{X})$ and $\mathcal{A}_{n,2}^h(\boldsymbol{X})$, we achieve that

$$\mathcal{A}_{n,1}^h(\boldsymbol{X}) = \sum_{j:|\mathcal{W}_{j|h}|=1} \exp(\boldsymbol{X}^\top(\boldsymbol{P}_{Q,h}^0 + \boldsymbol{B}_{h,j}^*\boldsymbol{A}_{h,j}^*)\boldsymbol{P}_{K,h}^0\boldsymbol{X})\Big[(\bar{V}_{h,n,1,j}^\top\boldsymbol{X}\boldsymbol{X}^\top\sigma_2(\boldsymbol{B}_{h,j}^*)$$
$$+ \bar{V}_{h,n,1,j}^\top\boldsymbol{X}\sigma_1(\boldsymbol{A}_j^*)\boldsymbol{X})(\boldsymbol{P}_V^0 + \sigma_2(\boldsymbol{B}_{j,h}^*)\sigma_2(\boldsymbol{A}_j^*))\boldsymbol{X} + \bar{V}_{h,n,1,j}^\top\boldsymbol{X}\sigma_2(\boldsymbol{B}_{h,j}^*) + \sigma_1(\boldsymbol{A}_j^*)\boldsymbol{X}\bar{V}_{h,n,2,j}\Big] + \mathcal{R}_{n,1}(\boldsymbol{X})$$

$$\mathcal{A}_{n,2}^h(\boldsymbol{X}) = \sum_{j:|\mathcal{W}_{j|h}|>1} \exp(\boldsymbol{X}^\top(\boldsymbol{P}_{Q,h}^0 + \sigma_2(\boldsymbol{B}_{h,j}^*)\sigma_1(\boldsymbol{A}_j^*))\boldsymbol{X})\Big[(\bar{V}_{h,n,1,j}^\top\boldsymbol{X}\boldsymbol{X}^\top\sigma_2(\boldsymbol{B}_{h,j}^*) + \bar{V}_{h,n,2,j}^\top\boldsymbol{X}\sigma_1(\boldsymbol{A}_j^*)\boldsymbol{X}),$$
$$+ \boldsymbol{X}^\top\bar{V}_{h,n,1,j}\boldsymbol{X}(\boldsymbol{X}^\top\sigma_2(\boldsymbol{B}_{h,j}^*) + \bar{V}_{h,n,4,j}^\top\boldsymbol{X}\boldsymbol{X}^\top\sigma_2(\boldsymbol{B}_{h,j}^*) + \boldsymbol{X}^\top\bar{V}_{h,n,5,j}\boldsymbol{X}(\sigma_1(\boldsymbol{A}_j^*)\boldsymbol{X})^2 + \bar{V}_{h,n,6,j}^\top\boldsymbol{X}\sigma_1(\boldsymbol{A}_j^*)\boldsymbol{X}$$
$$+ \boldsymbol{X}^\top\bar{V}_{h,n,7,j}\boldsymbol{X} + \boldsymbol{X}^\top\bar{V}_{h,n,7,j}\boldsymbol{X}\boldsymbol{X}^\top\sigma_2(\boldsymbol{B}_{h,j}^*\sigma_1(\boldsymbol{A}_j^*)\boldsymbol{X}) \times (\boldsymbol{P}_V^0 + \sigma_2(\boldsymbol{B}_j^*)\sigma_1(\boldsymbol{A}_j^*))\boldsymbol{X} + \bar{V}_{h,n,1,j}^\top\boldsymbol{X}\sigma_2(\boldsymbol{B}_j^*)$$
$$+ \sigma_1(\boldsymbol{A}_j^*) + \bar{V}_{h,n,4,j}^\top\boldsymbol{X}\sigma_2(\boldsymbol{B}_j^*) + \sigma_1(\boldsymbol{A}_j^*)\boldsymbol{X}\bar{V}_{h,n,6,j} + \bar{V}_{h,n,7,j}^\top\boldsymbol{X}\Big] + \mathcal{R}_{h,n,2}(\boldsymbol{X}),$$

where the values of $\bar{V}_{h,n,1,j}, \ldots, \bar{V}_{h,n,7,j}$ are given by

$$\bar{V}_{h,n,1,j} := (\bar{U}_{h,n,j,e_u,0_d}\sigma_1'(\boldsymbol{A}^{(u)}))_{u=1}^d$$
$$\bar{V}_{h,n,2,j} := (\bar{U}_{h,n,j,0_d,e_u}\sigma_2'(\boldsymbol{A}^{(u)}))_{u=1}^d$$
$$\bar{V}_{h,n,3,j} := (\bar{U}_{h,n,j,e_u+e_v,0_d}\sigma_1'(\boldsymbol{A}^{(u)})\sigma_1'(\boldsymbol{A}^{(v)}))_{u,v=1}^d$$
$$\bar{V}_{h,n,4,j} := (\bar{U}_{h,n,j,2e_u,0_d}\sigma_1''(\boldsymbol{A}^{(u)}))_{u=1}^d$$
$$\bar{V}_{h,n,5,j} := (\bar{U}_{h,n,j,e_u+e_v,0_d}\sigma_2'(\boldsymbol{B}^{(u)})\sigma_2'(\boldsymbol{B}^{(v)}))_{u,v=1}^d$$
$$\bar{V}_{h,n,6,j} := (\bar{U}_{h,n,j,0_d,2e_u}\sigma_2''(\boldsymbol{B}^{(u)}))_{u=1}^d$$
$$\bar{V}_{h,n,7,j} := (\bar{U}_{n,j,e_u,e_v}\sigma_1'(\boldsymbol{B}^{(u)})\sigma_2'(\boldsymbol{B}^{(v)}))_{u,v=1}^d$$

Here, $e_u$ denotes the $u$-th canonical basis vector in $\mathbb{R}^d$, that is, the vector whose $u$-th component equals 1 and all other components equal 0. Similarly, $e_{uv}$ denotes the canonical basis matrix in $\mathbb{R}^{d \times d}$, with a 1 in the $(u, v)$-th entry and 0 elsewhere.

**Decomposition of the function $\mathcal{B}_n(X)$.** Consider the function $\mathcal{B}_n^h(X)$, we decompose it as

$$\mathcal{B}_n^h(X) = \sum_{j:|\mathcal{W}_{j|h}|=1} \sum_{i \in \mathcal{W}_{j|h}} \exp(c_{n,i}) \left[ R(X; B_{n,i}^h, A_{n,i}) - R(X; B_{h,j}^*, A_j^*) \right] g_{G_n}^h(X)$$

$$+ \sum_{j:|\mathcal{W}_{j|h}|>1} \sum_{i \in \mathcal{W}_{j|h}} \exp(c_{n,i}) \left[ R(X; B_{n,i}^h, A_{n,i}^h) - R(X; B_{h,j}^*, A_j^*) \right] g_{G_n}^h(X)$$

$$:= \mathcal{B}_{n,1}^h(X) + \mathcal{B}_{n,2}^h(X).$$

Using Taylor's expansions up to the first order for $\mathcal{B}_{n,1}^h$ and the second order for $\mathcal{B}_{n,2}^h$, we have

$$\mathcal{B}_{n,1}^h = \sum_{j:|\mathcal{W}_{j|h}|=1} \sum_{|\alpha|=1} \bar{U}_{n,j,\alpha_1,\alpha_2}^h \frac{\partial^\alpha R}{\partial A^{\alpha_1} B^{\alpha_2}}(X; B_{h,j}^*, A_j^*) g_{G_n}^h(X) + \mathcal{R}_{n,3}^h(X)$$

$$\mathcal{B}_{n,2}^h = \sum_{j:|\mathcal{W}_{j|h}|=1} \sum_{1 \le |\alpha| \le 2} \bar{U}_{n,j,\alpha_1,\alpha_2}^h \frac{\partial^\alpha R}{\partial A^{\alpha_1} B^{\alpha_2}}(X; B_{h,j}^*, A_j^*) g_{G_n}^h(X) + \mathcal{R}_{n,4}^h(X)$$

where the Taylor remainders $\mathcal{R}_{n,3}^h(X)$ and $\mathcal{R}_{n,4}^h(X)$ are small compared with $\mathcal{D}_2(G_n, G_*)$: $\mathcal{R}_{n,3}^h(X)/\mathcal{D}_2(G_n, G_*) \to 0$, and $\mathcal{R}_{n,4}^h(X)/\mathcal{D}_2(G_n, G_*) \to 0$. This leads to

$$\mathcal{B}_{n,1}^h(X) = \sum_{j:|\mathcal{W}_{j|h}|=1} \exp(X^\top (P_{Q,h}^h + \sigma_2(B_{h,j}^*)\sigma_1(A_j^*))X) \left[ \bar{V}_{h,n,1,j}^\top XX^\top \sigma_2(B_{h,j}^*) + \bar{V}_{h,n,2,j}^\top X\sigma_1(A_j^*)X \right] g_{G_n}^h(X)$$

$$+ \mathcal{R}_{n,3}^h X$$

$$\mathcal{B}_{n,2}^h(X) = \sum_{j:|\mathcal{W}_{j|h}|>1} \exp(X^\top (P_{Q,h}^h + \sigma_2(B_{h,j}^*)\sigma_1(A_j^*))X) \left[ \bar{V}_{h,n,1,j}^\top XX^\top \sigma_2(B_{h,j}^*) + \bar{V}_{h,n,2,j}^\top X\sigma_1(A_j^*)X \right.$$

$$\left. + X^\top \bar{V}_{h,n,3,j} X(X^\top \sigma_2(B_j^*))^2 + \bar{V}_{h,n,4,j}^\top XX^\top \sigma_2(B_{j,n}^*)\sigma_1(A_j^*)X \right] g_{G_n}^h(X) + \mathcal{R}_{n,4}^h X.$$

Putting all the above results together, the function $\mathcal{L}_n^h(\boldsymbol{X})$ can be represented as

$$
\begin{aligned}
\mathcal{L}_n^h(\boldsymbol{X}) =& \sum_{j:|\mathcal{W}_{j|h}|=1} \exp(\boldsymbol{X}^\top(\boldsymbol{P}_{Q,h}^0 + \boldsymbol{B}_{h,j}^*\boldsymbol{A}_{h,j}^*)\boldsymbol{P}_{K,h}^0\boldsymbol{X})\Big[(\bar{V}_{h,n,1,j}^\top\boldsymbol{X}\boldsymbol{X}^\top\sigma_2(\boldsymbol{B}_{h,j}^*) + \bar{V}_{h,n,2,j}^\top\boldsymbol{X}\sigma_1(\boldsymbol{A}_j^*)\boldsymbol{X}) \\
&\times (\boldsymbol{P}_V^0 + \sigma_2(\boldsymbol{B}_{j,h})^*\sigma_1(\boldsymbol{A}_j^*))\boldsymbol{X} + \bar{V}_{h,n,1,j}^\top\boldsymbol{X}\sigma_2(\boldsymbol{B}_{h,j}^*) + \sigma_1(\boldsymbol{A}_j^*)\boldsymbol{X}\bar{V}_{h,n,2,j}\Big] \\
&+ \sum_{j:|\mathcal{W}_{j|h}|>1} \exp(\boldsymbol{X}^\top(\boldsymbol{P}_{Q,h}^0 + \boldsymbol{B}_{h,j}^*\boldsymbol{A}_{h,j}^*)\boldsymbol{P}_{K,h}^0\boldsymbol{X})\Big[(\bar{V}_{h,n,1,j}^\top\boldsymbol{X}\boldsymbol{X}^\top\sigma_2(\boldsymbol{B}_{h,j}^*) + \bar{V}_{h,n,2,j}^\top\boldsymbol{X}\sigma_1(\boldsymbol{A}_j^*)\boldsymbol{X} \\
&+ \boldsymbol{X}^\top\bar{V}_{h,n,3,j}\boldsymbol{X}(\boldsymbol{X}^\top\sigma_2(\boldsymbol{B}_{h,j}^*))^2 + \bar{V}_{h,n,4,j}^\top\boldsymbol{X}\boldsymbol{X}^\top\sigma_2(\boldsymbol{B}_{h,j}^*) + \boldsymbol{X}^\top\bar{V}_{h,n,5,j}\boldsymbol{X}(\sigma_1(\boldsymbol{A}_j^*)\boldsymbol{X})^2 \\
&+ \bar{V}_{h,n,6,j}^\top\boldsymbol{X}\sigma_1(\boldsymbol{A}_j^*)\boldsymbol{X} + \boldsymbol{X}^\top\bar{V}_{h,n,7,j}\boldsymbol{X} + \boldsymbol{X}^\top\bar{V}_{h,n,7,j}\boldsymbol{X}\boldsymbol{X}^\top\sigma_2(\boldsymbol{B}_{h,j}^*)\sigma_1(\boldsymbol{A}_j^*)\boldsymbol{X}) \\
&\times (\boldsymbol{P}_{V,h}^0 + \sigma_2(\boldsymbol{B}_{j,h}^*)\sigma_1(\boldsymbol{A}_j^*)\boldsymbol{X}) + \bar{V}_{h,n,1,j}^\top\boldsymbol{X}\sigma_2(\boldsymbol{B}_{h,j}^\top) + \sigma_1(\boldsymbol{A}_j^*)\boldsymbol{X}\bar{V}_{h,n,2,j} + \bar{V}_{h,n,4,j}^\top\boldsymbol{X}\sigma_2(\boldsymbol{B}_j^*) \\
&+ \sigma_1(\boldsymbol{A}_j^*)\boldsymbol{X}\bar{V}_{h,n,6,j} + \bar{V}_{h,n,7,j}^\top\boldsymbol{X}\Big] \\
&- \sum_{j:|\mathcal{W}_{j|h}|=1} \exp(\boldsymbol{X}^\top(\boldsymbol{P}_Q^0 + \sigma_2(\boldsymbol{B}_{h,j})\sigma_1(\boldsymbol{A}_j^*))\boldsymbol{X})\Big[\bar{V}_{h,n,1,j}^\top\boldsymbol{X}\boldsymbol{X}^\top\sigma_2(\boldsymbol{B}_{h,j}^*) + \bar{V}_{h,n,2,j}^\top\boldsymbol{X}\sigma_1(\boldsymbol{A}_j^*)\boldsymbol{X}\Big]g_{\widetilde{G}_n}(\boldsymbol{X}) \\
&- \sum_{j:|\mathcal{W}_{j|h}|>1} \exp(\boldsymbol{X}^\top(\boldsymbol{P}_Q^0 + \sigma_2(\boldsymbol{B}_{h,j})\sigma_1(\boldsymbol{A}_j^*))\boldsymbol{X})\Big[\bar{V}_{h,n,1,j}^\top\boldsymbol{X}\boldsymbol{X}^\top\sigma_2(\boldsymbol{B}_{h,j}^*) + \bar{V}_{h,n,2,j}^\top\boldsymbol{X}\sigma_1(\boldsymbol{A}_j^*)\boldsymbol{X}\boldsymbol{X}^\top \\
&+ \boldsymbol{X}^\top\bar{V}_{h,n,3,j}\boldsymbol{X}(\boldsymbol{X}^\top\sigma_2(\boldsymbol{B}_{h,j}^*))^2 + \bar{V}_{h,n,4,j}^\top\boldsymbol{X}\boldsymbol{X}^\top\sigma_2(\boldsymbol{B}_{h,j}^*) + \boldsymbol{X}^\top\bar{V}_{h,n,5,j}\boldsymbol{X}(\boldsymbol{X}^\top\sigma_1(\boldsymbol{A}_j^*))^2 \\
&+ \bar{V}_{h,n,6,j}^\top\boldsymbol{X}\sigma_1(\boldsymbol{A}_j^*)\boldsymbol{X} + \boldsymbol{X}^\top\bar{V}_{h,n,7,j}\boldsymbol{X} + \boldsymbol{X}^\top\bar{V}_{h,n,7,j}\boldsymbol{X}\boldsymbol{X}^\top\sigma_2(\boldsymbol{B}_{h,j}^*)\sigma_1(\boldsymbol{A}_j^*)\boldsymbol{X}\Big]g_{\widetilde{G}_n}(\boldsymbol{X}) \\
&+ \sum_{j=1}^L \bar{T}_{n,j}\exp(\boldsymbol{X}^\top(\boldsymbol{P}_{Q,h}^0 + \boldsymbol{B}_{h,j}^*\boldsymbol{A}_{h,j}^*)\boldsymbol{P}_{K,h}^0\boldsymbol{X})\big[(\boldsymbol{P}_V^0 + \boldsymbol{B}_{h,j}^*\boldsymbol{A}_j^*)\boldsymbol{X} - g_{\widetilde{G}_n}(\boldsymbol{X})\big] \\
&+ \mathcal{R}_{n,1}^h(\boldsymbol{X}) + \mathcal{R}_{n,2}^h(\boldsymbol{X}) - \mathcal{R}_{n,3}^h(\boldsymbol{X}) - \mathcal{R}_{n,4}^h(\boldsymbol{X}),
\end{aligned}
\tag{18}
$$

where $\bar{T}_{n,j}^h := \sum_{i\in\mathcal{W}_{j|h}}\exp(c_{n,i}) - \exp(c_j^*)$ for any $j \in [L]$.

**Step 2 - Non-vanishing coefficients.** The Eq. (18) shows that the ratio $\mathcal{L}_n(\boldsymbol{X})/\mathcal{D}_{2n}$ can be decomposed as a linear combination of the following independent function

$$g^h_{\widetilde{G}_*}(x),$$

$$\frac{1}{D^h_{g,*}(\boldsymbol{X})}\boldsymbol{R}(\boldsymbol{X};\boldsymbol{B}^*_{h,j},\boldsymbol{A}^*_j)\boldsymbol{X}^{(u)}\boldsymbol{X}^\top\sigma_2(\boldsymbol{B}^*_j)\boldsymbol{S}(\boldsymbol{X};\boldsymbol{B}^*_j,\boldsymbol{A}^*_j),$$

$$\frac{1}{D^h_{g,*}(\boldsymbol{X})}\boldsymbol{R}(\boldsymbol{X};\boldsymbol{B}^*_{h,j},\boldsymbol{A}^*_j)\boldsymbol{X}^{(u)}\sigma_1(\boldsymbol{A}^*_j)\boldsymbol{X}\boldsymbol{S}(\boldsymbol{X};\boldsymbol{B}^*_j,\boldsymbol{A}^*_j),$$

$$\frac{1}{D^h_{g,*}(\boldsymbol{X})}\boldsymbol{X}^{(u)}\sigma_2(\boldsymbol{B}^*_j),\ \frac{1}{D^h_{g,*}(\boldsymbol{X})}\boldsymbol{R}(\boldsymbol{X};\boldsymbol{B}^*_{h,j},\boldsymbol{A}_j)\sigma_1(\boldsymbol{A}^*_j)\boldsymbol{X}e_u,$$

$$\frac{1}{D^h_{g,*}(\boldsymbol{X})}\boldsymbol{R}(\boldsymbol{X};\boldsymbol{B}^*_{h,j},\boldsymbol{A}^*_j)\boldsymbol{X}^{(u)}\boldsymbol{X}^{(v)}(\boldsymbol{X}^\top\sigma_2(\boldsymbol{B}^*_{h,j}))^2\boldsymbol{S}(\boldsymbol{X};\boldsymbol{B}^*_{h,j},\boldsymbol{A}^*_j),$$

$$\frac{1}{D^h_{g,*}(\boldsymbol{X})}\boldsymbol{R}(\boldsymbol{X};\boldsymbol{B}^*_{h,j},\boldsymbol{A}^*_j)\boldsymbol{X}^{(u)}\boldsymbol{X}^{(v)}(\sigma_1(\boldsymbol{A}^*_j)\boldsymbol{X})^2\boldsymbol{S}(\boldsymbol{X};\boldsymbol{B}^*_{h,j},\boldsymbol{A}^*_j),$$

$$\frac{1}{D^h_{g,*}(\boldsymbol{X})}\boldsymbol{R}(\boldsymbol{X};\boldsymbol{B}^*_{h,j},\boldsymbol{A}^*_j)\boldsymbol{X}^{(u)}\sigma_1(\boldsymbol{A}^*_j)\boldsymbol{X}\boldsymbol{S}(\boldsymbol{X};\boldsymbol{B}^*_{h,j},\boldsymbol{A}^*_j),$$

$$\frac{1}{D^h_{g,*}(\boldsymbol{X})}\boldsymbol{R}(\boldsymbol{X};\boldsymbol{B}^*_{h,j},\boldsymbol{A}^*_j)\boldsymbol{X}^{(u)}\boldsymbol{X}^{(v)}\boldsymbol{S}(\boldsymbol{X};\boldsymbol{B}^*_{h,j},\boldsymbol{A}^*_j),$$

$$\frac{1}{D^h_{g,*}(\boldsymbol{X})}\boldsymbol{R}(\boldsymbol{X};\boldsymbol{B}^*_{h,j},\boldsymbol{A}^*_j)\boldsymbol{X}^{(u)}\boldsymbol{X}^{(v)}\boldsymbol{X}^\top\sigma_2(\boldsymbol{B}^*_{h,j})\boldsymbol{X}\boldsymbol{S}(\boldsymbol{X};\boldsymbol{B}^*_{h,j},\boldsymbol{A}^*_j),$$

$$\frac{1}{D^h_{g,*}(\boldsymbol{X})}\boldsymbol{R}(\boldsymbol{X};\boldsymbol{B}^*_{h,j},\boldsymbol{A}^*_j)\boldsymbol{X}^{(u)}\boldsymbol{X}^\top\sigma_2(\boldsymbol{B}_{j,h})g^\top_{\widetilde{G}_n},\ \frac{1}{D^h_{g,*}(\boldsymbol{X})}\boldsymbol{R}(\boldsymbol{X};\boldsymbol{B}^*_{h,j},\boldsymbol{A}^*_j)\boldsymbol{X}^{(u)}\boldsymbol{X}^\top\sigma_1(\boldsymbol{A}_j)g^\top_{\widetilde{G}_n},$$

$$\frac{1}{D^h_{g,*}(\boldsymbol{X})}\boldsymbol{R}(\boldsymbol{X};\boldsymbol{B}^*_{h,j},\boldsymbol{A}^*_j)\boldsymbol{X}^{(u)}\boldsymbol{X}^{(v)}(\boldsymbol{X}^\top\sigma_2(\boldsymbol{B}_{j,h}))^2g^h_{\widetilde{G}_n},\ \frac{1}{D^h_{g,*}(\boldsymbol{X})}\boldsymbol{R}(\boldsymbol{X};\boldsymbol{B}^*_{h,j},\boldsymbol{A}^*_j)\boldsymbol{X}^{(u)}\boldsymbol{X}^\top\sigma_2(\boldsymbol{B}_{j,h})g^h_{\widetilde{G}_n},$$

$$\frac{1}{D^h_{g,*}(\boldsymbol{X})}\boldsymbol{R}(\boldsymbol{X};\boldsymbol{B}^*_{h,j},\boldsymbol{A}^*_j)\boldsymbol{X}^{(u)}\boldsymbol{X}^{(v)}(\sigma_1(\boldsymbol{A}_j)\boldsymbol{X})^2g^h_{\widetilde{G}_n},\ \frac{1}{D^h_{g,*}(\boldsymbol{X})}\boldsymbol{R}(\boldsymbol{X};\boldsymbol{B}^*_{h,j},\boldsymbol{A}^*_j)\boldsymbol{X}^{(u)}\sigma_1(\boldsymbol{B}_{j,h})\boldsymbol{X}g^h_{\widetilde{G}_n},$$

$$\frac{1}{D^h_{g,*}(\boldsymbol{X})}\boldsymbol{R}(\boldsymbol{X};\boldsymbol{B}^*_{h,j},\boldsymbol{A}^*_j)\boldsymbol{X}^{(u)}\boldsymbol{X}^{(v)}g^h_{\widetilde{G}_n},\ \frac{1}{D^h_{g,*}(\boldsymbol{X})}\boldsymbol{R}(\boldsymbol{X};\boldsymbol{B}^*_{h,j},\boldsymbol{A}^*_j)\boldsymbol{X}^{(u)}\boldsymbol{X}^{(v)}\boldsymbol{X}^\top\sigma_2(\boldsymbol{B}_{j,h})\sigma_1(\boldsymbol{A}_j)\boldsymbol{X}g^h_{\widetilde{G}_n},$$

$$\frac{1}{D^h_{g,*}(\boldsymbol{X})}\boldsymbol{R}(\boldsymbol{X};\boldsymbol{B}^*_{h,j},\boldsymbol{A}^*_j)\boldsymbol{S}(\boldsymbol{X};\boldsymbol{B}^*_{h,j},\boldsymbol{A}^*_j),\ \frac{1}{D^h_{g,*}(\boldsymbol{X})}\boldsymbol{R}(\boldsymbol{X};\boldsymbol{B}^*_{h,j},\boldsymbol{A}^*_j)g^h_{\widetilde{G}_n},$$

for any indices $1 \le h \le H$, $1 \le j \le L$, and $1 \le u_1,v_1,u_2,v_2 \le d$.

We establish that in the limit $n \to \infty$, there exists at least one coefficient of these functions that does not disappear. Assume by contrary that all these coefficients of these linear independent functions go to 0. From Eq. (18), we obtain that $\bar{U}_{h,n,j,\alpha_1,\alpha_2}/\mathcal{D}_{2n}$, $\bar{U}_{h,n,j,\alpha_1,\beta_1,\alpha_2,\beta_2}/\mathcal{D}_{2n}$, and $\bar{T}_{h,n,j}/\mathcal{D}_{2n}$ go to 0 for all the coefficient $\alpha_1,\beta_1,\alpha_2,\beta_2 \in \mathbb{R}^{d\times d}$ satisfying that $1 \le |\alpha_1|+|\beta_1|+|\alpha_2|+|\beta_2| \le 2$.

Consider the coefficient of $g^h_{\widetilde{G}_*}(x)$, we have

$$\frac{1}{\mathcal{D}_{2n}}|\pi^n_h - \pi^*_h| \to 0. \tag{19}$$

Since $\bar{T}^h_{n,j}/\mathcal{D}_{2n} \to 0$, we have for any $j \in [L]$

$$\frac{1}{\mathcal{D}_{2n}}\pi^n_h\left|\sum_{i\in\mathcal{W}_{j|h}}\exp(c_{n,i}) - \exp(c^*_j)\right| = \frac{|\bar{T}^h_{n,j}|}{\mathcal{D}_{2n}} \to 0.$$

Taking the summation with respect to $j \in [L]$ and $h \in [H]$, we have

$$\frac{1}{\mathcal{D}_{2n}}\sum_{h=1}^H\pi^n_h\sum_{l=1}^L\left|\sum_{i\in\mathcal{W}_{l|h}}\exp(c_i) - \exp(c^*_h)\right| \to 0. \tag{20}$$

For index $j \in [L]$ such that $|\mathcal{W}_{j|h}| = 1$, the limits $\bar{U}_{h,n,j,e_u,0_d}/\mathcal{D}_{2n} \to 0$ implies that

$$\frac{1}{\mathcal{D}_{2n}} \pi_h^n \sum_{j:|\mathcal{W}_{j|h}|=1} \sum_{i \in \mathcal{W}_{j|h}} \exp(c_{n,i}) \|\Delta \boldsymbol{A}_{n,ij}\|_1 \to 0.$$

Noting that in Euclidean finite-dimensional space, all the norms are equivalent, we can express the equation above using $\ell_2$ norm, before summing up with respect to $l$ and $h$:

$$\frac{1}{\mathcal{D}_{2n}} \sum_{h=1}^{H} \pi_h^n \sum_{j:|\mathcal{W}_{j|h}|=1} \sum_{i \in \mathcal{W}_{j|h}} \exp(c_{n,i}) \|\Delta \boldsymbol{A}_{n,ij}\| \to 0.$$

Analogously, since $\bar{U}_{h,n,j,0_d,e_u}/\mathcal{D}_{2n} \to 0$, it also follows that

$$\frac{1}{\mathcal{D}_{2n}} \pi_h^n \sum_{j:|\mathcal{W}_{j|h}|=1} \sum_{i \in \mathcal{W}_{j|h}} \exp(c_{n,i}) \|\Delta \boldsymbol{B}_{n,ij}\|_1 \to 0,$$

which implies that

$$\frac{1}{\mathcal{D}_{2n}} \sum_{h=1}^{H} \pi_h^n \sum_{j:|\mathcal{W}_{j|h}|=1} \sum_{i \in \mathcal{W}_{j|h}} \exp(c_{n,i})(\|\Delta \boldsymbol{A}_{n,ij}\| + \|\Delta \boldsymbol{B}_{n,ij}^h\|) \to 0. \tag{21}$$

The similar argument also demonstrates that for $\mathcal{W}_{j|h} > 1$, the limits $\bar{U}_{h,n,j,2e_u,0_d}/\mathcal{D}_{2n} \to 0$ and $\bar{U}_{h,n,j,0_d,2e_u}/\mathcal{D}_{2n} \to 0$ imply

$$\frac{1}{\mathcal{D}_{2n}} \sum_{h=1}^{H} \pi_h^n \sum_{j:|\mathcal{W}_{j|h}|>1} \sum_{i \in \mathcal{W}_{j|h}} \exp(c_{n,i})(\|\Delta \boldsymbol{A}_{n,ij}\| + \|\Delta \boldsymbol{B}_{n,ij}^h\|) \to 0. \tag{22}$$

By putting all the results in Eq. (19), Eq. (20), Eq. (22), and Eq. (22) together, we achieve that $1 = \frac{\mathcal{D}_{2n}}{\mathcal{D}_{2n}} \to 0$, which is a contradiction. As a result, at least one of the coefficients of the linear independent functions in $\mathcal{L}_n(\boldsymbol{X})/\mathcal{D}_{2n}$ does not vanish as $n \to \infty$.

**Step 3 - Application of the Fatou's lemma.** Denote $\bar{m}_n$ as the maximum of the absolute values of the coefficients of the linear independent functions in $\mathcal{L}_n(\boldsymbol{X})/\mathcal{D}_{2n}$. Given that at least one of these coefficients does not vanish, we have $1/\bar{m}_n \not\to 0$ as $n \to \infty$. Since $\|h_{\widetilde{G}_n} - h_{\widetilde{G}_*}\|_{L^2(\mu)}/\mathcal{D}_{2n} \to 0$ as $n \to \infty$, we also have $\|h_{\widetilde{G}_n} - h_{\widetilde{G}_*}\|_{L^2(\mu)}/\bar{m}_n \mathcal{D}_{2n} \to 0$. Using Fatou's lemma, we have

$$0 = \lim_{n\to\infty} \frac{\|g_{\widetilde{G}_n} - g_{\widetilde{G}_*}\|_{L^2(\mu)}}{\bar{m}_n \mathcal{D}_{2n}} \geq \int \liminf_{n\to\infty} \frac{|g_{\widetilde{G}_n}(\boldsymbol{X}) - g_{\widetilde{G}_*}(\boldsymbol{X})|}{\bar{m}_n \mathcal{D}_{2n}} d\mu(\boldsymbol{X}) \geq 0.$$

As a consequence, we achieve that

$$\liminf_{n\to\infty} \frac{|g_{\widetilde{G}_n}(\boldsymbol{X}) - g_{\widetilde{G}_*}(\boldsymbol{X})|}{\bar{m}_n \mathcal{D}_{2n}} = 0, \quad a.s.\boldsymbol{X}.$$

When $n \to \infty$, we denote

$$\frac{\bar{T}_{h,n,j}}{\bar{m}_n \mathcal{D}_{2n}} \to \bar{\lambda}_{0,j}, \quad \frac{\bar{V}_{h,n,\tau,j}}{\bar{m}_n \mathcal{D}_{2n}} \to \bar{\lambda}_{h,\tau,j}$$

for any indices $h \in [H]$, $j \in [L]$, $\tau \in [7]$, bearing in mind that at least one element of the set $\{\lambda_{h,0,j}, \bar{\lambda}_{h,\tau,j} : j \in [L], \tau \in [7]\}$ is not equal to 0. Given the notation above, the limit

$\liminf_{n\to\infty} \frac{|g_{\widetilde{G}_n}(\boldsymbol{X}) - g_{\widetilde{G}_*}(\boldsymbol{X})|}{m_n \mathcal{D}_{2n}}$ can be expressed as

$$\sum_{h=1}^{H} \sum_{j:|\mathcal{W}_{j|h}|=1} \exp(\boldsymbol{X}^\top (\boldsymbol{P}_{Q,h}^0 + \boldsymbol{B}_{h,j}^* \boldsymbol{A}_{h,j}^*) \boldsymbol{P}_{K,h}^0 \boldsymbol{X}) \Big[ (\bar{\lambda}_{h,1,j}^\top \boldsymbol{X} \boldsymbol{X}^\top \sigma_2(\boldsymbol{B}_{h,j}^*) + \bar{\lambda}_{h,2,j}^\top \boldsymbol{X} \sigma_1(\boldsymbol{A}_j^*) \boldsymbol{X})$$

$$\times (\boldsymbol{P}_V^0 + \sigma_2(\boldsymbol{B}_{h,j}^* \sigma_1(\boldsymbol{A}_j^*))\boldsymbol{X} + \bar{\lambda}_{h,1,j}^\top \boldsymbol{X} \sigma_2(\boldsymbol{B}_{h,j}^*) + \sigma_1(\boldsymbol{A}_j^*) \boldsymbol{X} \bar{\lambda}_{h,2,j} \Big]$$

$$+ \sum_{h=1}^{H} \sum_{j:|\mathcal{W}_{j|h}|>1} \exp(\boldsymbol{X}^\top (\boldsymbol{P}_{Q,h}^0 + \boldsymbol{B}_{h,j}^* \boldsymbol{A}_{h,j}^*) \boldsymbol{P}_{K,h}^0 \boldsymbol{X}) \Big[ (\bar{\lambda}_{h,1,j}^\top \boldsymbol{X} \boldsymbol{X}^\top \sigma_2(\boldsymbol{B}_{h,j}^*)$$

$$+ \bar{\lambda}_{h,2,j}^\top \boldsymbol{X} \sigma_1(\boldsymbol{A}_j^*) \boldsymbol{X} + \boldsymbol{X}^\top \bar{\lambda}_{h,3,j} \boldsymbol{X} (\boldsymbol{X}^\top \sigma_2(\boldsymbol{B}_{h,j}^*))^2 + \lambda_{h,4,j}^\top \boldsymbol{X} \boldsymbol{X}^\top \sigma_2(\boldsymbol{B}_{h,j}^*) + \boldsymbol{X}^\top \bar{\lambda}_{h,5,j} \boldsymbol{X} (\sigma_1(\boldsymbol{A}_{h,j}^*) \boldsymbol{X})^2$$

$$+ \lambda_{h,6,j}^\top \boldsymbol{X} \sigma_1(\boldsymbol{A}_j^*) \boldsymbol{X} + \boldsymbol{X}^\top \bar{\lambda}_{h,7,j} \boldsymbol{X} + \boldsymbol{X}^\top \bar{\lambda}_{h,7,j} \boldsymbol{X} \boldsymbol{X}^\top \sigma_2(\boldsymbol{B}_{h,j}^*) \sigma_1(\boldsymbol{A}_j^*) \boldsymbol{X})$$

$$\times (\boldsymbol{P}_V^0 + \sigma_2(\boldsymbol{B}_{j,h})^* \sigma_1(\boldsymbol{A}_j^*)) \boldsymbol{X} + \bar{\lambda}_{h,1,j}^\top \boldsymbol{X} \sigma_2(\boldsymbol{B}_{h,j}^*) + \sigma_1(\boldsymbol{A}_j^*) \boldsymbol{X} \bar{\lambda}_{h,2,j} + \bar{\lambda}_{h,4,j}^\top \boldsymbol{X} \sigma_2(\boldsymbol{B}_{h,j}^*) + \sigma_1(\boldsymbol{A}_j^*) \boldsymbol{X} \bar{\lambda}_{h,6,j} + \lambda_{h,7,j}^\top \boldsymbol{X} \Big]$$

$$- \sum_{h=1}^{H} \sum_{j:|\mathcal{W}_{j|h}|=1} \exp(\boldsymbol{X}^\top (\boldsymbol{P}_{Q,h}^0 + \boldsymbol{B}_{h,j}^* \boldsymbol{A}_{h,j}^*) \boldsymbol{P}_{K,h}^0 \boldsymbol{X}) \Big[ \bar{\lambda}_{h,1,j}^\top \boldsymbol{X} \boldsymbol{X}^\top \sigma_2(\boldsymbol{B}_{h,j}^*) + \bar{\lambda}_{h,2,j}^\top \boldsymbol{X} \sigma_1(\boldsymbol{A}_j^*) \boldsymbol{X} \Big] g_{\widetilde{G}_n}(\boldsymbol{X})$$

$$- \sum_{h=1}^{H} \sum_{j:|\mathcal{W}_{j|h}|>1} \exp(\boldsymbol{X}^\top (\boldsymbol{P}_{Q,h}^0 + \boldsymbol{B}_{h,j}^* \boldsymbol{A}_{h,j}^*) \boldsymbol{P}_{K,h}^0 \boldsymbol{X}) \Big[ \bar{\lambda}_{h,1,j}^\top \boldsymbol{X} \boldsymbol{X}^\top \sigma_2(\boldsymbol{B}_{h,j}^*) + \bar{\lambda}_{h,2,j}^\top \boldsymbol{X} \sigma_1(\boldsymbol{A}_j^*) \boldsymbol{X}$$

$$+ \boldsymbol{X}^\top \bar{\lambda}_{h,3,j} \boldsymbol{X} (\boldsymbol{X}^\top \sigma_2(\boldsymbol{B}_{h,j}^*))^2 + \bar{\lambda}_{h,4,j}^\top \boldsymbol{X} \boldsymbol{X}^\top \sigma_2(\boldsymbol{B}_{h,j}^*) + \boldsymbol{X}^\top \bar{\lambda}_{h,5,j} \boldsymbol{X} (\sigma_1(\boldsymbol{A}_j^*) \boldsymbol{X})^2$$

$$+ \bar{\lambda}_{h,6,j}^\top \boldsymbol{X} \sigma_1(\boldsymbol{A}_j^*) \boldsymbol{X} + \boldsymbol{X}^\top \bar{\lambda}_{h,7,j} \boldsymbol{X} + \boldsymbol{X}^\top \bar{\lambda}_{h,7,j} \boldsymbol{X} \boldsymbol{X}^\top \sigma_2(\boldsymbol{B}_{h,j}^*) \sigma_1(\boldsymbol{A}_j^*) \boldsymbol{X} \Big] g_{\widetilde{G}_n}(\boldsymbol{X})$$

$$+ \sum_{h=1}^{H} \sum_{j=1}^{L} \bar{\lambda}_{0,j} \exp(\boldsymbol{X}^\top (\boldsymbol{P}_{Q,h}^0 + \boldsymbol{B}_{h,j}^* \boldsymbol{A}_{h,j}^*) \boldsymbol{P}_{K,h}^0 \boldsymbol{X}) \Big[ (\boldsymbol{P}_V^0 + \boldsymbol{B}_{h,j}^* \boldsymbol{A}_j^*) - g_{\widetilde{G}_*}(\boldsymbol{X}) \Big] = 0. \qquad (23)$$

for almost surely $\boldsymbol{X}$. Nevertheless, this equation implies that all the coefficients $\{\bar{\lambda}_{h,0,j}, \bar{\lambda}_{\tau,j} : j \in [L], \tau \in [7]\}$ are 0's, which is a contradiction. As a consequence, we achieve that

$$\lim_{\epsilon \to 0} \inf_{\widetilde{G} \in \mathcal{G}_{H,L'}(\widetilde{\Theta}): \mathcal{D}_2(\widetilde{G}, \widetilde{G}^*) \le \epsilon} \|g_{\widetilde{G}} - g_{\widetilde{G}^*}\|_{L^2(\mu)} / \mathcal{D}_2(\widetilde{G}, \widetilde{G}^*) > 0$$

**Proof of global part (Eq. (15))**

The proof in local part shows that there exists a constant $\epsilon'$ such that

$$\inf_{\widetilde{G} \in \mathcal{G}_{H,L'}(\widetilde{\Theta}): \mathcal{D}_2(\widetilde{G}, \widetilde{G}_*) \le \epsilon'} \|g_{\widetilde{G}} - g_{\widetilde{G}_*}\|_{L^2(\mu)} / \mathcal{D}_2(\widetilde{G}, \widetilde{G}_*) > 0.$$

To complete the proof of this result, we show the global part that

$$\inf_{\widetilde{G} \in \mathcal{G}_{H,L'}(\widetilde{\Theta}): \mathcal{D}_2(\widetilde{G}, \widetilde{G}_*) > \epsilon'} \|g_{\widetilde{G}} - g_{\widetilde{G}_*}\|_{L^2(\mu)} / \mathcal{D}_2(\widetilde{G}, \widetilde{G}_*) > 0.$$

The proof of the above equation relies mostly on the identifiability of mixing measure in $\mathcal{G}_L(\Theta)$. Assume by contradiction that this claim does not hold, then there exists a sequence of measure $\widetilde{G}_n = \sum_{j=1}^{L} \exp(c_{n,j}) \delta_{(\boldsymbol{W}_{2,j}^n, \boldsymbol{B}_{h,j}^n, \boldsymbol{W}_{1,j}^n, \boldsymbol{A}_{h,j}^n)} \in \mathcal{G}_{H,L'}(\widetilde{\Theta})$ such that

$$\begin{cases} \mathcal{D}_2(\widetilde{G}_n, \widetilde{G}_*) > \epsilon' \\ \|g_{\widetilde{G}_n} - g_{\widetilde{G}_*}\|_{L^2(\mu)} / \mathcal{D}_2(\widetilde{G}_n, G_*) \to 0, \end{cases}$$

as $n \to \infty$. Without loss of generality, we can suppose that both $\boldsymbol{W}_{2,j}^n$ and $\boldsymbol{W}_{1,j}^n$ are identity matrices. As a result, we have $\|g_{\widetilde{G}_n} - g_{\widetilde{G}_*}\|_{L^2(\mu)} \to 0$ as $n \to \infty$. From the hypothesis that the parameter space $\Theta$ is a compact set, there exists a mixing measure $\widetilde{G} \in \mathcal{G}_{H,L'}(\widetilde{\Theta})$ such that one of the $\widetilde{G}_n$'s subsequence converges to $\widetilde{G}$. By extracting this sequence, without loss of generality, we

can suppose that $\widetilde{G}_n \to \widetilde{G}'$. Since $\mathcal{D}_2(\widetilde{G}_n, \widetilde{G}_*) > \epsilon'$ for all $n \geq 1$, we obtain that $\mathcal{D}_2(\widetilde{G}', \widetilde{G}_*) \geq \epsilon'$. Using the Fatou's lemma, we have

$$0 = \lim_{n\to\infty} \|g_{\widetilde{G}_n} - g_{\widetilde{G}_*}\|_{L^2(\mu)} = \lim_{n\to\infty} \int \|g_{\widetilde{G}_n}(\boldsymbol{X}) - g_{G_*}(\boldsymbol{X})\|^2 d\mu(\boldsymbol{X})$$

$$= \int \liminf_{n\to\infty} \|g_{\widetilde{G}_n}(\boldsymbol{X}) - g_{\widetilde{G}_*}(\boldsymbol{X})\|^2 d\mu(\boldsymbol{X}) \geq 0.$$

As a result, $g_{\widetilde{G}'}(\boldsymbol{X}) = g_{\widetilde{G}_*}(\boldsymbol{X})$ for almost surely $\boldsymbol{X}$, which implies from identifiability in $\mathcal{G}_{H,L'}(\widetilde{\Theta})$ that $\widetilde{G}' \equiv \widetilde{G}_*$. Thus, $\mathcal{D}_2(\widetilde{G}', \widetilde{G}_*) = 0$, which is a contradiction with the fact that $\mathcal{D}_2(\widetilde{G}', \widetilde{G}_*) \geq \epsilon'$. This completes our proof.

**Proof for identifiability property.** In this part, we prove that the equality $g_{\widetilde{G}}(\boldsymbol{X}) = g_{\widetilde{G}_*}(\boldsymbol{X})$ for almost sure every $\boldsymbol{X}$ implies the identity $\widetilde{G} = \widetilde{G}_*$. For the convenience of presentation, we simplify the softmax notation that, for any mixing measure $\widetilde{G} = \sum_{h=1}^{H} \sum_{j=1}^{L} \exp(c_j) \delta_{(\boldsymbol{B}_{h,j}^*, \boldsymbol{A}_j^*)}$, we denote

$$\mathrm{softmax}_{\widetilde{G}}(u) = \frac{\exp(u)}{\sum_{j=1}^{L} \exp(\boldsymbol{X}^\top (\boldsymbol{P}_Q^0 + \sigma_2(\boldsymbol{B}_j^h)) \sigma_1(\boldsymbol{A}_j)) \boldsymbol{X} + c_j)},$$

where $u \in \{\boldsymbol{X}^\top(\boldsymbol{P}_Q^0 + \sigma_2(\boldsymbol{B}_{h,j})\sigma_2(\boldsymbol{A}_j))\boldsymbol{X} + c_j : j \in [L]\}$. The equation $g_{\widetilde{G}}(\boldsymbol{X}) = g_{\widetilde{G}_*}(\boldsymbol{X})$ implies that

$$\sum_{h=1}^{H} \pi_h \sum_{j=1}^{L} \mathrm{softmax}(\boldsymbol{X}^\top(\boldsymbol{P}_Q^0 + \sigma_2(\boldsymbol{B}_{h,j})\sigma_2(\boldsymbol{A}_j))\boldsymbol{X} + c_j)(\boldsymbol{P}_{V,h}^0 + \sigma_2(\boldsymbol{B}_{h,j}^*)\sigma_1(\boldsymbol{A}_j^*))\boldsymbol{X}$$

$$= \sum_{h=1}^{H} \pi_h \sum_{j=1}^{L'} \mathrm{softmax}(\boldsymbol{X}^\top(\boldsymbol{P}_Q^0 + \sigma_2(\bar{\boldsymbol{B}}_{h,j})\sigma_2(\bar{\boldsymbol{A}}_j))\boldsymbol{X} + c_j^*)(\boldsymbol{P}_{V,h}^0 + \sigma_2(\bar{\boldsymbol{B}}_{h,j}^*)\sigma_1(\bar{\boldsymbol{A}}_j^*))\boldsymbol{X}.$$

From this equation, we can deduce that

$$\sum_{j=1}^{L} \mathrm{softmax}(\boldsymbol{X}^\top(\boldsymbol{P}_Q^0 + \sigma_2(\boldsymbol{B}_{h,j})\sigma_2(\boldsymbol{A}_j))\boldsymbol{X} + c_j)(\boldsymbol{P}_{V,h}^0 + \sigma_2(\boldsymbol{B}_{h,j}^*)\sigma_1(\boldsymbol{A}_j^*))\boldsymbol{X}$$

$$= \sum_{j=1}^{L'} \mathrm{softmax}(\boldsymbol{X}^\top(\boldsymbol{P}_Q^0 + \sigma_2(\bar{\boldsymbol{B}}_{h,j})\sigma_2(\bar{\boldsymbol{A}}_j))\boldsymbol{X} + c_j^*)(\boldsymbol{P}_{V,h}^0 + \sigma_2(\bar{\boldsymbol{B}}_{h,j}^*)\sigma_1(\bar{\boldsymbol{A}}_j^*))\boldsymbol{X}. \quad (24)$$

This equation implies that $L = L'$, and
$\{\mathrm{softmax}(\boldsymbol{X}^\top(\boldsymbol{P}_Q^0 + \sigma_2(\boldsymbol{B}_{h,j})\sigma_2(\boldsymbol{A}_j))\boldsymbol{X} + c_j) : j \in [L]\} = \{\mathrm{softmax}(\boldsymbol{X}^\top(\boldsymbol{P}_Q^0 + \sigma_2(\bar{\boldsymbol{B}}_{h,j})\sigma_2(\bar{\boldsymbol{A}}_j))\boldsymbol{X} + c_j^*) : j \in [L]\}$
for almost surely $\boldsymbol{X}$. Up to a permutation, we can assume without loss of generality that for any $j \in [L]$ that
$$\mathrm{softmax}(\boldsymbol{X}^\top(\boldsymbol{P}_Q^0 + \sigma_2(\boldsymbol{B}_{h,j})\sigma_2(\boldsymbol{A}_j))\boldsymbol{X} + c_j) = \mathrm{softmax}(\boldsymbol{X}^\top(\boldsymbol{P}_Q^0 + \sigma_2(\bar{\boldsymbol{B}}_{h,j})\sigma_2(\bar{\boldsymbol{A}}_j))\boldsymbol{X} + c_j^*).$$
Given the invariance to translation of the softmax function, Eq. (24) implies that

$$\sum_{j=1}^{L} \exp(c_j) \exp(\boldsymbol{X}^\top(\boldsymbol{P}_Q^0 + \sigma_2(\boldsymbol{B}_{h,j})\sigma_1(\boldsymbol{A}_j)\boldsymbol{X}))(\boldsymbol{P}_V^0 + \sigma_2(\boldsymbol{B}_{h,j})\sigma_1(\boldsymbol{A}_j))\boldsymbol{X}$$

$$= \sum_{j=1}^{L} \exp(c_j^*) \exp(\boldsymbol{X}^\top(\boldsymbol{P}_Q^0 + \sigma_2(\bar{\boldsymbol{B}}_{h,j})\sigma_1(\bar{\boldsymbol{A}}_j)\boldsymbol{X}))(\boldsymbol{P}_V^0 + \sigma_2(\bar{\boldsymbol{B}}_{h,j})\sigma_1(\bar{\boldsymbol{A}}_j))\boldsymbol{X}$$

for almost surely $\boldsymbol{X}$.

Noting that the index set $[L]$ can be partitioned into $\bar{m}$ subsets $\bar{K}_1, \ldots, \bar{K}_m$ where $m \leq L$ such that $\exp(c_j) = \exp(c_{j'}^*)$ for any indices $j, j' \in \bar{K}_i$ and $i \in [\bar{m}]$, we can write the equation above into

$$\sum_{i=1}^{\bar{m}} \sum_{j \in \bar{K}_i} \exp(c_j) \exp(\boldsymbol{X}^\top(\boldsymbol{P}_Q^0 + \sigma_2(\boldsymbol{B}_{h,j})\sigma_1(\boldsymbol{A}_j)))\boldsymbol{X}$$

$$= \sum_{i=1}^{\bar{m}} \sum_{j \in \bar{K}_i} \exp(c_j^*) \exp(\boldsymbol{X}^\top(\boldsymbol{P}_Q^0 + \sigma_2(\bar{\boldsymbol{B}}_{h,j})\sigma_1(\bar{\boldsymbol{A}}_j)))\boldsymbol{X}$$

for almost surely $\boldsymbol{X}$. The above equation implies that

$$\{(\boldsymbol{P}_V^0 + \sigma_2(\boldsymbol{B}_{h,j})\sigma_1(\boldsymbol{A}_j)) : j \in \bar{K}_i\} = \{(\boldsymbol{P}_V^0 + \sigma_2(\bar{\boldsymbol{B}}_{h,j})\sigma_1(\bar{\boldsymbol{A}}_j)) : j \in \bar{K}_i\}$$

Given that the activation $\sigma_1$ and $\sigma_2$ are algebraically independent, the above result demonstrates that

$$\sum_{i=1}^{\bar{m}} \sum_{j \in \bar{K}_i} \exp(c_j) \delta_{(\boldsymbol{B}_{h,j}, \boldsymbol{A}_j)} = \sum_{i=1}^{\bar{m}} \sum_{j \in \bar{K}_i} \exp(c_j^*) \delta_{(\boldsymbol{B}_{h,j}^*, \boldsymbol{A}_j^*)}.$$

As a consequence, we achieve that $\widetilde{G} \equiv \widetilde{G}_*$, which completes our proof. $\qquad \square$

***Proof of Proposition 1.*** The proof of Proposition 1 can be implemented using the following steps.

**Step 1: Equivalence between least square estimator and MLE.**

Bearing in mind that the sample $(\boldsymbol{X}_1, \boldsymbol{Y}_1), \dots, (\boldsymbol{X}_n, \boldsymbol{Y}_n) \in \mathbb{R}^{\bar{d}} \times \mathbb{R}^{\bar{d}}$ are i.i.d. from the regression model

$$\boldsymbol{Y}_i = g_{\widetilde{G}_*}(\boldsymbol{X}_i) + \epsilon_i, \quad i = 1, \dots, n,$$

such that the noises $\epsilon_1, \dots, \epsilon_n$ are independent and follow the Gaussian distribution: $\mathbb{E}[\epsilon_i|\boldsymbol{X}_i] = 0$ and $\mathrm{Var}[\epsilon_i|\boldsymbol{X}_i] = \sigma^2 I_{\bar{d}}$ for all $i \in [n]$. In addition, $g_{\bar{G}_*}$ follows the following form

$$g_{\widetilde{G}_*}(\boldsymbol{X}) = \sum_{h=1}^{H} \pi_h \sum_{j=1}^{L} \frac{\exp(\boldsymbol{X}^\top(\boldsymbol{P}_{Q,h}^0 + \sigma_2(\boldsymbol{W}_{2,j}^*\boldsymbol{B}_{h,j}^*)\sigma_1(\boldsymbol{W}_{1,j}^*\boldsymbol{A}_j^*)\boldsymbol{P}_{K,h}^0\boldsymbol{X} + c_j^*)}{D_h(\boldsymbol{X})}$$
$$\times (\boldsymbol{P}_{V,h}^0 + \sigma_2(\boldsymbol{W}_{2,j}^*\boldsymbol{B}_{h,j}^*)\sigma_1(\boldsymbol{W}_{1,j}^*\boldsymbol{A}_j^*))\boldsymbol{X}$$

where we denote $D_h(\boldsymbol{X}) = \sum_{j=1}^{L} \exp(\boldsymbol{X}^\top(\boldsymbol{P}_{Q,h}^0 + \sigma_2(\boldsymbol{W}_{2,j}^*\boldsymbol{B}_{h,j}^*)\sigma_1(\boldsymbol{W}_{1,j}^*\boldsymbol{A}_j^*)\boldsymbol{P}_{K,h}^0\boldsymbol{X} + c_j^*)$. Also, we consider the least-square estimator $\widetilde{G}_n$ of the form

$$\widetilde{G}_n := \arg \min_{\widetilde{G} \in \mathcal{G}_{H,L'}(\widetilde{\Theta})} \sum_{i=1}^{n} \|\boldsymbol{Y}_i - g_{\widetilde{G}}(\boldsymbol{X}_i)\|^2.$$

Using the Gaussianity assumption of $\epsilon_i|\boldsymbol{X}_i$ for all $i \in [n]$, we achieve that $\boldsymbol{Y}_i|\boldsymbol{X}_i \sim \mathcal{N}(g_{\widetilde{G}_*}(\boldsymbol{X}_i), \sigma^2 I_{\bar{d}})$ for all $i \in [n]$. As a result, the least square estimator $\bar{G}_n$ is actually a maximum likelihood estimator with respect to the data $\boldsymbol{Y}_1|\boldsymbol{X}_1, \dots, \boldsymbol{Y}_n|\boldsymbol{X}_n$:

$$\widetilde{G}_n \in \arg \max_{\widetilde{G} \in \mathcal{G}_{H,L'}(\widetilde{\Theta})} \frac{1}{n} \sum_{i=1}^{n} \log(p(\boldsymbol{Y}_i|g_{\widetilde{G}}(\boldsymbol{X}_i), \sigma^2 I_{\bar{d}}),$$

where $p(\boldsymbol{Y}_i|g_{\widetilde{G}}(\boldsymbol{X}_i), \sigma^2 I_{\bar{d}})$ denotes the multivariate Gaussian distribution with mean $g_{\widetilde{G}}(\boldsymbol{X}_i)$ and covariance matrix $\sigma^2 I_{\bar{d}}$.

**Step 2: Main ingredients for measuring regression function and their usefulness.**

Let $\mathcal{P}_{HL}$ denotes the set of conditional density of all mixing measures in $\widetilde{\mathcal{G}}_{H,L'}(\widetilde{\Theta})$, i.e. $\mathcal{P}_{H,L'}(\widetilde{\Theta}) := \{p_G(\boldsymbol{Y}|\boldsymbol{X}), \widetilde{G} \in \mathcal{G}_{H,L'}(\widetilde{\Theta})\}$. In addition, we denote

$$\tilde{\mathcal{P}}_{H,L'}(\widetilde{\Theta}) := \{p_{(\widetilde{G}+\widetilde{G}_*)/2}(\boldsymbol{Y}|\boldsymbol{X}) : \widetilde{G} \in \mathcal{G}_{H,L'}(\widetilde{\Theta})\}$$
$$\tilde{\mathcal{P}}_{H,L'}^{1/2}(\widetilde{\Theta}) := \{p_{(\widetilde{G}+\widetilde{G}_*)/2}^{1/2}(\boldsymbol{Y}|\boldsymbol{X}) : \widetilde{G} \in \mathcal{G}_{H,L'}(\widetilde{\Theta})\}$$

For each $\delta > 0$, we denote the Hellinger's ball in $\tilde{\mathcal{P}}_{H,L'}^{1/2}(\widetilde{\Theta})$ around the conditional density $p_{\widetilde{G}}(\boldsymbol{Y}|\boldsymbol{X})$:

$$\tilde{\mathcal{P}}_{H,L'}^{1/2}(\widetilde{\Theta}, \delta) := \{p^{1/2} \in \tilde{\mathcal{P}}_{H,L'}^{1/2}(\widetilde{\Theta}) : d_H(p, p_{\widetilde{G}_*}) \le \delta\}.$$

Lastly, as suggested in (van de Geer (2000)), we quantify the measure of the above set by

$$\mathcal{J}(\delta, \tilde{\mathcal{P}}_{H,L'}^{1/2}(\widetilde{\Theta}, \delta)) := \int_{\delta^2/2^{13}}^{\delta} H_B^{1/2}(t, \tilde{\mathcal{P}}_{H,L'}^{1/2}(\widetilde{\Theta}, t), \|\cdot\|_{\mathcal{L}^2(\mu)})dt \vee \delta,$$

where $H_B(t, \tilde{\mathcal{P}}_{H,L'}^{1/2}(\widetilde{\Theta}, t), \|\cdot\|_{\mathcal{L}^2(\mu)})$ denotes the bracketing entropy of $\tilde{\mathcal{P}}_{H,L}^{1/2}(\widetilde{\Theta}, t)$ under $\mathcal{L}^2$-norm, while $t \vee \delta = \max(t, \delta)$.

Employing similar argument of Theorem 7.4 and Theorem 9.2 in van de Geer (2000), it is tractable to achieve the following lemma.

**Lemma 1.** *Consider $\Psi(\delta) \geq \mathcal{J}(\delta, \mathcal{P}_{HL}^{1/2}(\Theta, \delta))$ such that $\Psi(\delta)/\delta^2$ is a non-increasing function of $\delta$. Then, there exist a universal constant $c$ and a sequence $(\delta_n)$ such that $\sqrt{n}\delta_n^2 \geq c\Psi(\delta_n)$ and*

$$\mathbb{P}\left(\mathbb{E}_{\boldsymbol{X}}[d_H(p_{\widetilde{G}_n}(\cdot|\boldsymbol{X}), p_{\widetilde{G}_*}(\cdot|\boldsymbol{X}))] > \delta\right) \leq c\exp\left(-\frac{n\delta^2}{\nu^2}\right)$$

*for all $\delta \geq \delta_n$.*

The main part of the proof consists of demonstrating the upper bound for the bracketing entropy for any $0 < \epsilon \leq 1/2$

**Lemma 2.** *We can bound the bracket entropy $H_B$ by*

$$H_B(\epsilon, \tilde{\mathcal{P}}_{H,L'}^{1/2}(\widetilde{\Theta}, t), \|\cdot\|_{\mathcal{L}^2(\mu)}) \lesssim \log(1/\epsilon). \tag{25}$$

If this estimation holds, since it is straightforward to check that

$$H_B(\epsilon, \tilde{\mathcal{P}}_{H,L'}^{1/2}(\widetilde{\Theta}, t), \|\cdot\|_{\mathcal{L}^2(\mu)}) \leq H_B(\epsilon, \tilde{\mathcal{P}}_{H,L'}^{1/2}(\widetilde{\Theta}, t), d_H)$$

where $d_H$ denotes the Hellinger's distance, we have

$$\mathcal{J}_B(\delta, \tilde{\mathcal{P}}_{H,L'}^{1/2}(\widetilde{\Theta}, \delta)) \leq \int_{\delta^2/2^{13}}^{\delta} H_B^{1/2}(\epsilon, \tilde{\mathcal{P}}_{H,L'}^{1/2}(\widetilde{\Theta}, t), d_H)dt \vee \delta \lesssim \int_{\delta^2/2^{13}}^{\delta} \log(1/t)dt \vee \delta. \tag{26}$$

Consider $\Psi(\delta) = \delta \cdot [\log(1/\delta)]^{1/2}$, the it is obvious that $\Psi(\delta)/\delta^2$ is non-increasing function of $\delta$. In addition, Eq. (26) implies that $\Psi(\delta) \geq \mathcal{J}_B(\delta, \tilde{\mathcal{P}}_{H,L'}^{1/2}(\widetilde{\Theta}, \delta))$. By choosing $\delta_n = \sqrt{\log(n)/n}$, we have $\sqrt{n}\delta_n^2 \geq c\Psi(\delta_n)$ for some universal constant $c$. An application of Lemma 1 leads us to the conclusion of Proposition 1:

$$d_H(p(\boldsymbol{Y}|g_{\widetilde{G}_n}(\boldsymbol{X}), \sigma^2 I_d), p(\boldsymbol{Y}|g_{\widetilde{G}_*}(\boldsymbol{X}), \sigma^2 I_d)) = \mathcal{O}(\sqrt{\log(n)/n}), \tag{27}$$

where $d_H$ denotes the Hellinger distance. The closed form of Hellinger distance between two multivariate normal distance gives us

$$d_H(p(\boldsymbol{Y}|g_{\widetilde{G}_n}(\boldsymbol{X}), \sigma^2 I_d), p(\boldsymbol{Y}|g_{\widetilde{G}_*}(\boldsymbol{X}), \sigma^2 I_d)) = 1 - \exp\left\{-\frac{1}{8\sigma^2}\|g_{\widetilde{G}_n}(\boldsymbol{X}) - g_{\widetilde{G}_*}(\boldsymbol{X})\|^2\right\}.$$

In consequence, for $n$ sufficiently large, there exists some universal constant $C$ such that the above inequality implies

$$\|g_{\widetilde{G}_n}(\boldsymbol{X}) - g_{\widetilde{G}_*}(\boldsymbol{X})\|^2 \leq 8\sigma^2 \log\left(\frac{1}{1 - C\log(n)/n}\right) \leq 16\sigma^2 C\log(n)/n.$$

From this inequality, we have

$$\|g_{\bar{G}_n}(\boldsymbol{X}) - g_{\bar{G}_*}(\boldsymbol{X})\| = \mathcal{O}(\sqrt{\log(n)/n}),$$

or $\|g_{\widetilde{G}_n} - g_{\widetilde{G}_*}\|_{L^2(\mu)} = \mathcal{O}_P(\sqrt{\log(n)/n})$. This concludes the proof of this proposition.

**Proof of the bound in Eq. (25).**

**Step 3: Relation between bracket entropy and covering number.**

The first step of this proof includes establishing the upper bound for the multivariate Gaussian density $p_{\widetilde{G}}(\cdot|\boldsymbol{X})$. Noting that the variance effect $\sigma^2$ is fixed, we have

$$p_{\widetilde{G}}(\boldsymbol{Y}|\boldsymbol{X}) = \frac{1}{(2\pi\sigma^2)^{d/2}} \exp\left(-\frac{\|\boldsymbol{Y} - g_{\widetilde{G}}(\boldsymbol{X})\|^2}{2\sigma^2}\right) \leq \frac{1}{(2\pi\sigma^2)^{d/2}}.$$

Since the input space $\mathcal{X}$ and parameter space $\widetilde{\Theta}$ are bounded, there exists a constant $M$ such that $\|g_{\widetilde{G}}(\boldsymbol{X})\| \leq M$ for $\widetilde{G} \in \mathcal{G}_{H,L'}$ and $\boldsymbol{X} \in \mathcal{X}$. Thus, for any $\|\boldsymbol{Y}\| \geq 2M$, we have $\frac{\|\boldsymbol{Y} - g_{\widetilde{G}}(\boldsymbol{X})\|^2}{2\sigma^2} \geq \frac{\|\boldsymbol{Y}\|^2}{8\sigma^2}$, which leads to

$$p(\boldsymbol{Y}|g_{\widetilde{G}}(\boldsymbol{X}), \sigma^2 I_{\bar{d}}) = \frac{1}{(2\pi\sigma^2)^{d/2}} \exp\left(-\frac{\|\boldsymbol{Y} - g_{\widetilde{G}}(\boldsymbol{X})\|^2}{2\sigma^2}\right) \leq \frac{1}{(2\pi\sigma^2)^{d/2}} \exp\left(-\frac{\|\boldsymbol{Y}\|^2}{8\sigma^2}\right).$$

Define the integrable function

$$K(\boldsymbol{Y}|\boldsymbol{X}) = \begin{cases} (2\pi\sigma^2)^{-d/2} & \text{for } \|\boldsymbol{Y}\| \leq 2M, \\ (2\pi\sigma^2)^{-d/2} \exp\left(-\frac{\|\boldsymbol{Y}\|^2}{8\sigma^2}\right) & \text{for } \|\boldsymbol{Y}\| > 2M, \end{cases}$$

then the above estimations give us $p(\boldsymbol{Y}|g_{\widetilde{G}}(\boldsymbol{X}), \sigma^2 I_{\bar{d}}) \leq K(\boldsymbol{Y}|\boldsymbol{X})$ for all $\boldsymbol{Y}$ and $\boldsymbol{X} \in \mathcal{X}$.

For $\eta < \epsilon$, consider an $\eta$-cover $\{\mu_1, \ldots, \mu_n\}$ of $\mathcal{P}_{H,L'}(\widetilde{\Theta})$ under $\ell_1$-norm such that $N := N(\eta, \mathcal{P}_{H,L'}(\widetilde{\Theta}), \|\cdot\|_1)$. Then, the brackets of the form $[L_i(\boldsymbol{Y}|\boldsymbol{X}), U_i(\boldsymbol{Y}|\boldsymbol{X})]$, for $1 \leq i \leq N$, can be constructed as

$$L_i(\boldsymbol{Y}|\boldsymbol{X}) := \max\{\mu_i(\boldsymbol{Y}|\boldsymbol{X}) - \eta, 0\},$$
$$U_i(\boldsymbol{Y}|\boldsymbol{X}) := \max\{\mu_i(\boldsymbol{Y}|\boldsymbol{X}) + \eta, K(\boldsymbol{Y}|\boldsymbol{X})\}.$$

It is straightforward to check that $\mathcal{P}_{H,L'}(\widetilde{\Theta}) \subset \bigcup_{i=1}^N [L_i(\boldsymbol{Y}|\boldsymbol{X}), U_i(\boldsymbol{Y}|\boldsymbol{X})]$ and $L_i(\boldsymbol{Y}|\boldsymbol{X}) - U_i(\boldsymbol{Y}|\boldsymbol{X}) \leq \min\{\eta, K(\boldsymbol{Y}|\boldsymbol{X})\}$. From this, we can achieve the following upper bound

$$\|U_i - L_i\|_1 = \int_{\|\boldsymbol{Y}\| \leq 2M} |U_i(\boldsymbol{Y}|\boldsymbol{X}) - L_i(\boldsymbol{Y}|\boldsymbol{X})| d(\boldsymbol{X}, \boldsymbol{Y}) + \int_{\|\boldsymbol{Y}\| > 2M} |U_i(\boldsymbol{Y}|\boldsymbol{X}) - L_i(\boldsymbol{Y}|\boldsymbol{X})| d(\boldsymbol{X}, \boldsymbol{Y})$$

$$\leq K\eta + \exp\left(-\frac{K^2}{2\sigma^2}\right) \leq K'\eta,$$

where $K := \max\{2M, \sqrt{8\sigma^2}\} \log(1/\eta)$, $K'$ be a positive constant. From the definition of bracket entropy, given that $H_B(K'\eta, \mathcal{P}_{H,L'}(\widetilde{\Theta}), \|\cdot\|_1)$ is the logarithm of the smallest number of bracket of size $K'\eta$ necessary to cover $\mathcal{P}_{H,L'}(\widetilde{\Theta})$, we have

$$H_B(K'\eta, \mathcal{P}_{H,L'}(\widetilde{\Theta}), \|\cdot\|_1) \leq \log(N) = \log N(\eta, \mathcal{P}_{H,L'}(\widetilde{\Theta}), \|\cdot\|_1).$$

If we can achieve the upper bound for the covering number $\log N(\eta, \mathcal{P}_{H,L'}(\widetilde{\Theta}), \|\cdot\|_1) \lesssim \log(1/\eta)$, then we achieve

$$H_B(K'\eta, \mathcal{P}_{H,L'}(\widetilde{\Theta}), \|\cdot\|_1) \lesssim \log(1/\eta).$$

By choosing $\epsilon = \epsilon/K'$, noting that Hellinger distance is upper bounded by $\ell_1$ norm, we have

$$H_B(\epsilon, \mathcal{P}_{H,L'}(\widetilde{\Theta}), d_H) \leq H_B(\epsilon, \mathcal{P}_{H,L'}(\widetilde{\Theta}), \|\cdot\|_1) \lesssim \log(1/\epsilon).$$

**Step 4: Bound covering number.**

Now, it is our turn to bound the covering number $N$. To do this, let $\Gamma := \{(\pi_1, \ldots, \pi_h) : \sum_{i=1}^h \pi_i = 1, \text{ and } \pi_i \geq 0\}$ and $\Delta = \{(\boldsymbol{B}_{h,j}, \boldsymbol{A}_j) : (\boldsymbol{B}_{h,j}, \boldsymbol{A}_j) \in \Omega\}$. Given that the parameter space $\Omega$ is compact, as well as $\Gamma$ is also a compact space, there exists $\xi$-cover for $\Gamma$ and $\Delta$, which can be denoted as $\Gamma_\xi$ and $\Delta_\xi$, respectively. In addition, it is straightforward to verify that

$$|\Gamma_\xi| \leq \mathcal{O}(\xi^{-(H-1)}), \quad |\Delta_\xi| \leq \mathcal{O}(\xi^{-(2rdHL^*)}).$$

For a mixing measure $G = \sum_{h=1}^H \pi_h \sum_{j=1}^L \exp(c_j) \delta_{(\boldsymbol{B}_{h,j}, \boldsymbol{A}_j)} \in \mathcal{G}_{H,L'}$, let $(\bar{c}_j, \bar{\boldsymbol{B}}_{h,j}, \bar{\boldsymbol{A}}_j) \in \Delta_\xi$ such that $(\bar{c}_j, \bar{\boldsymbol{B}}_{h,j}, \bar{\boldsymbol{A}}_j)$ is the closet point to $(c_j, \boldsymbol{B}_{h,j}, \boldsymbol{A}_j)$ in this set w.r.t. $\|\cdot\|_2$ norm, and $(\bar{\pi}_1, \ldots, \bar{\pi}_H) \in \Gamma_\xi$ such that $(\bar{\pi}_1, \ldots, \bar{\pi}_H)$ is the closet point to $(\pi_1, \ldots, \pi_H)$ in this set (also w.r.t. $\|\cdot\|_2$ norm). We consider two mixing measures:

$$\tilde{G} := \sum_{h=1}^H \pi_h \sum_{j=1}^L \exp(\bar{c}_j) \delta_{(\bar{\boldsymbol{B}}_{h,j}, \bar{\boldsymbol{A}}_j)}, \quad \bar{G} = \sum_{h=1}^H \bar{\pi}_h \sum_{j=1}^L \exp(\bar{c}_j) \delta_{(\bar{\boldsymbol{B}}_{h,j}, \bar{\boldsymbol{A}}_j)}.$$

For the sake of presentation, we denote

$$g_h(\boldsymbol{X}) := \sum_{l=1}^{L} \text{Softmax}(\boldsymbol{X}^\top(\boldsymbol{P}_{Q,h}^0 + \sigma_2(\boldsymbol{B}_{h,j}^*)\sigma_1(\boldsymbol{A}_j^*))\boldsymbol{P}_{K,h}^0\boldsymbol{X} + c_j^*) \cdot (\boldsymbol{P}_{V,h}^0 + \sigma_2(\boldsymbol{B}_{h,j}^*)\sigma_1(\boldsymbol{A}_j^*))\boldsymbol{X},$$

$$\tilde{g}_h(\boldsymbol{X}) := \sum_{l=1}^{L} \text{Softmax}(\boldsymbol{X}^\top(\boldsymbol{P}_{Q,h}^0 + \sigma_2(\boldsymbol{B}_{h,j}^*)\sigma_1(\boldsymbol{A}_j^*))\boldsymbol{P}_{K,h}^0\boldsymbol{X} + c_j^*) \cdot (\boldsymbol{P}_{V,h}^0 + \sigma_2(\bar{\boldsymbol{B}}_{h,j}^*)\sigma_1(\bar{\boldsymbol{A}}_j^*))\boldsymbol{X}$$

$$\bar{g}_h(\boldsymbol{X}) := \sum_{l=1}^{L} \text{Softmax}(\boldsymbol{X}^\top(\boldsymbol{P}_{Q,h}^0 + \sigma_2(\bar{\boldsymbol{B}}_{h,j}^*)\sigma_1(\bar{\boldsymbol{A}}_j^*))\boldsymbol{P}_{K,h}^0\boldsymbol{X} + \bar{c}_j^*) \cdot (\boldsymbol{P}_{V,h}^0 + \sigma_2(\bar{\boldsymbol{B}}_{h,j}^*)\sigma_1(\bar{\boldsymbol{A}}_j^*))\boldsymbol{X},$$

for all $h \in [H]$. We provide an upper bound for the discrepancy $\|g_G - g_{\tilde{G}}\|_\infty$ as

$$\|g_G - g_{\tilde{G}}\|_\infty \le \sum_{h=1}^{H} \pi_h \|g_h - \bar{g}_h\|_\infty \le \sum_{h=1}^{H} \|g_h - \bar{g}_h\|_\infty$$

$$\le \sum_{h=1}^{H} (\|g_h - \tilde{g}_h\|_\infty + \|\tilde{g}_h - \bar{g}_h\|_\infty) \tag{28}$$

For simplicity, denote $\mathcal{K}(\boldsymbol{X}, \boldsymbol{B}_{h,j}^*, \boldsymbol{A}_j^*) := (\boldsymbol{P}_{V,h}^0 + \sigma_2(\boldsymbol{B}_{h,j}^*)\sigma_1(\boldsymbol{A}_j^*))\boldsymbol{X}$. The discrepancy $\|g_h - \tilde{g}_h\|_\infty$ can be estimated as

$$\|g_h - \tilde{g}_h\|_\infty \le \sum_{l=1}^{L} \sup_{\boldsymbol{X} \in \mathcal{X}} |\text{Softmax}(\boldsymbol{X}^\top(\boldsymbol{P}_{Q,h}^0 + \sigma_2(\boldsymbol{B}_{h,j}^*)\sigma_1(\boldsymbol{A}_j^*))\boldsymbol{P}_{K,h}^0\boldsymbol{X} + c_j^*)| \cdot |\mathcal{K}(\boldsymbol{B}_{h,j}^*, \boldsymbol{A}_j^*) - \mathcal{K}(\bar{\boldsymbol{B}}_{h,j}^*, \bar{\boldsymbol{A}}_j^*)|$$

$$\le \sum_{l=1}^{L} \sup_{\boldsymbol{X} \in \mathcal{X}} |\mathcal{K}(\boldsymbol{X}, \boldsymbol{B}_{h,j}^*, \boldsymbol{A}_j^*) - \mathcal{K}(\boldsymbol{X}, \bar{\boldsymbol{B}}_{h,j}^*, \bar{\boldsymbol{A}}_j^*)|$$

$$\le \sum_{l=1}^{L} \sup_{\boldsymbol{X} \in \mathcal{X}} |(\sigma_2(\boldsymbol{B}_{h,j}^*)\sigma_1(\boldsymbol{A}_j^*)\boldsymbol{X} - \sigma_2(\bar{\boldsymbol{B}}_{h,j}^*)\sigma_1(\bar{\boldsymbol{A}}_j^*))\boldsymbol{X}|$$

$$\lesssim \sum_{l=1}^{L} \sup_{\boldsymbol{X} \in \mathcal{X}} (\|(\boldsymbol{B}_{h,j}^*, \boldsymbol{A}_j^*) - (\bar{\boldsymbol{B}}_{h,j}^*, \bar{\boldsymbol{A}}_j^*)\| \cdot \|\boldsymbol{X}\|)$$

$$\lesssim \sum_{l=1}^{L} \xi \cdot B \lesssim \xi, \tag{29}$$

where the second last inequality holds due to the fact that the input space is bounded: $\|\boldsymbol{X}\| \le B$ for all $\boldsymbol{X} \in \mathcal{X}$.

For the second term $\|\tilde{g}_h - \bar{g}_h\|_\infty$, denote $\mathcal{M}(\boldsymbol{X}, \boldsymbol{B}_{h,j}^*, \boldsymbol{A}_j^*, c_j^*) := \boldsymbol{X}^\top(\boldsymbol{P}_{Q,h}^0 + \sigma_2(\boldsymbol{B}_{h,j}^*)\sigma_1(\boldsymbol{A}_j^*))\boldsymbol{P}_{K,h}^0\boldsymbol{X} + c_j^*$, we bound it using the following argument:

$$\|\tilde{g}_h - \bar{g}_h\|_\infty \le \sum_{l=1}^{L} \sup_{\boldsymbol{X} \in \mathcal{X}} |\text{Softmax}(\mathcal{M}(\boldsymbol{X}, \boldsymbol{B}_{h,j}^*, \boldsymbol{A}_j^*)) - \text{Softmax}(\mathcal{M}(\boldsymbol{X}, \bar{\boldsymbol{B}}_{h,j}^*, \bar{\boldsymbol{A}}_j^*))| \cdot |\mathcal{K}(\boldsymbol{X}, \boldsymbol{B}_{h,j}^*, \boldsymbol{A}_j^*)|$$

$$\lesssim \sum_{l=1}^{L} \sup |\text{Softmax}(\mathcal{M}(\boldsymbol{X}, \boldsymbol{B}_{h,j}^*, \boldsymbol{A}_j^*)) - \text{Softmax}(\mathcal{M}(\boldsymbol{X}, \bar{\boldsymbol{B}}_{h,j}^*, \bar{\boldsymbol{A}}_j^*))|$$

$$\lesssim \sum_{l=1}^{L} \|(\bar{\boldsymbol{B}}_{h,j}^*, \bar{\boldsymbol{A}}_j^*) - (\boldsymbol{B}_{h,j}^*, \boldsymbol{A}_j^*)\|$$

$$\lesssim \sum_{l=1}^{L} \xi \lesssim \xi, \tag{30}$$

given that the $\boldsymbol{X}$ belongs to a compact space $\mathcal{X}$. Thus, the Eq. (28), Eq. (30), and Eq. (29) implies that $\|g_G - g_{\bar{G}}\|_\infty \lesssim \xi$. In addition, we similarly can bound $\|g_{\tilde{G}} - g_{\bar{G}}\|_\infty$ using the following step:

$$\|g_{\tilde{G}} - g_{\bar{G}}\|_\infty \leq \sum_{h=1}^{H} |\pi_h - \bar{\pi}_h| \cdot \|g_h\|$$

$$\lesssim \sum_{h=1}^{H} M \cdot |\pi_h - \bar{\pi}_h| \lesssim \xi,$$

where the second last inequality follows from the fact that the input spaces are compact, which means $|g_h(\boldsymbol{X})| \leq M$ for $\boldsymbol{X} \in \mathcal{X}$.

As a result, from the triangle inequality, we have

$$\|g_G - g_{\bar{G}}\|_\infty \leq \|g_G - g_{\tilde{G}}\|_\infty + \|g_{\tilde{G}} - g_{\bar{G}}\|_\infty \lesssim \xi.$$

Thus, noting that the Gaussian density function $f(x) = (2\pi\sigma^2)^{-d/2} \exp\left(-\|x\|^2/2\sigma^2\right)$ is a global Lipschitz function, we have

$$\|p(\boldsymbol{Y}|g_G(\boldsymbol{X}), \sigma^2 Id) - p(\boldsymbol{Y}|g_{\bar{G}}(\boldsymbol{X}), \sigma^2 Id)\|_1 \lesssim \|g_G(\boldsymbol{X}) - g_{\bar{G}}(\boldsymbol{X})\|_\infty \lesssim \xi.$$

From the definition of covering number, we get

$$\log N(\eta, \mathcal{P}_{H,L'}(\widetilde{\Theta}), \|\cdot\|_1) \leq |\Gamma_\xi| \times |\Delta_\xi| \leq \mathcal{O}(\xi^{-(H-1)}) \times \mathcal{O}(\xi^{-2rdHL^*}). \tag{31}$$

From Eq. (30) and Eq. (31), we have

$$H_B(\xi, \mathcal{P}_{H,L'}(\widetilde{\Theta}), \|\cdot\|_1) \lesssim \log(1/\xi).$$

By choosing $\xi = \epsilon/2$, we achieve that

$$H_B(\epsilon, \mathcal{P}_{H,L'}(\widetilde{\Theta}), d_H) \lesssim \log(1/\epsilon).$$

This completes our proof. $\qquad\square$

# C  ADDITIONAL EXPERIMENTAL DETAILS

## C.1  IMPLEMENTATION DETAILS

For vision tasks, we conduct experiments on ViT-B/16 (Vaswani et al. (2017)) for 100 epochs. The training configuration includes 100 warmup steps, a total batch size of 64, a Low-Rank Matrix rank of 8, and an alpha value of 8. We optimize the model using the AdamW optimizer with a cosine learning rate scheduler. To select learning rate and weight decay hyperparameters, we perform a grid search over the learning rate in $\{0.001, 0.005, 0.01, 0.05, 0.1\}$ and the weight decay in $\{0.0001, 0.0005, 0.001, 0.01, 0.1\}$. For the hypernetwork used in low-rank matrix B, the input dimension is 64, the hidden dimension is 16, and the activation function is leaky-relu.

For the commonsense reasoning tasks, we conduct experiments on two LLaMA versions, LLaMA-7B with 32 Transformer layers and LLaMA-13B with 40 Transformer layers Touvron et al. (2023). The training configuration includes a warmup steps of 100, a total batch size of 32, a learning rate of 2e-4, and a dropout value of 0.05. The models are trained with 1 A100-GPUs for 3 epochs. The rank of Low-Rank Matrix is 32 and the alpha value is 64. We optimize the models using AdamW optimizer with a linear learning rate scheduler. In both LLaMA-7B and LLaMA-13B settings, the hypernetwork used in low-rank matrix B has input dimension of 64 and hidden dimension of 40, while the activation function is leaky-relu.

## C.2  DETAIL OF SAMPLE EFFICIENCY

We provide in Figure 3 and 4 the detail of the sample efficiency problem in each commonsense reasoning dataset with LLaMA-7B setting and LLaMA-13B setting. From these figures, we observe that HoRA yield clear sample-efficiency improvements over LoRA on both 7B and 13B scales.

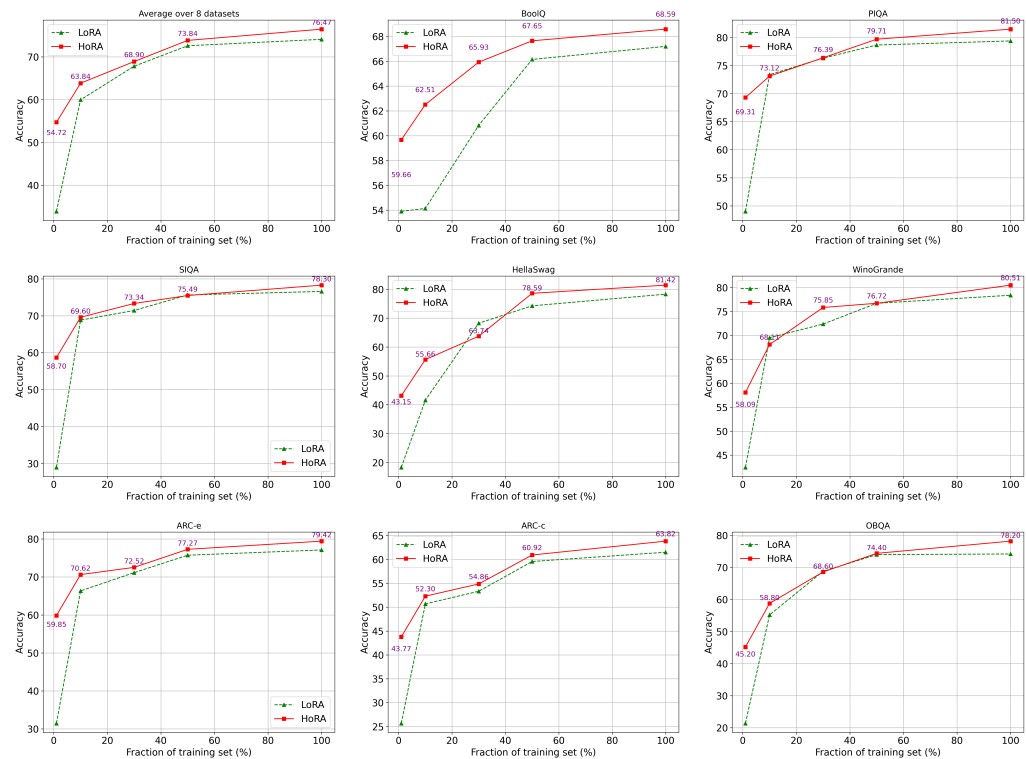

Figure 3: The detail of sample efficiency on each commonsense reasoning dataset with LLaMA-7B settings.

## C.3 DETAIL OF RESULTS ON VTAB-1K DATASETS

In Table 4, we provide the results of HoRA in detail for each dataset in the VTAB-1K domain. Compared to LoRA, HoRA consistently outperforms by 1-6% percents on almost all datasets except for sNORB-ele with only a modest increase in the number of parameters, therefore suggesting the effectiveness of having shared information among the attention heads.

Table 4: Classification accuracy on the VTAB-1K dataset

| | **Natural** | | | | | | | **Specialized** | | | | **Structured** | | | | | | | | |
| Method | CIFAR100 | Caltech101 | DTD | Flower102 | Pets | SVHN | Sun397 | Camelyon | EuroSAT | Resisc45 | Retinopathy | Clevr-Count | Clevr-Dist | DMLab | KITTI | dSpr-Loc | dSpr-ori | sNORB-Azim | sNORB-Ele | AVG |
|---|---|---|---|---|---|---|---|---|---|---|---|---|---|---|---|---|---|---|---|---|
| FFT | 68.9 | 87.7 | 64.3 | 97.2 | 86.9 | 87.4 | 38.8 | 79.7 | 95.7 | 84.2 | 73.9 | 56.3 | 58.6 | 41.7 | 65.5 | 57.5 | 46.7 | 25.7 | 29.1 | 65.6 |
| LoRA | 67.1 | 91.4 | 69.4 | 98.2 | 90.4 | 85.3 | 54 | 84.9 | 95.3 | 84.4 | 73.6 | 82.9 | 69.2 | 49.8 | 78.5 | 75.7 | 47.1 | 31 | 44 | 72.2 |
| DoRA | 67.9 | 90.4 | 70.6 | 99 | 90.2 | 89.6 | 54.6 | 83.9 | 95.5 | 85.3 | 75.9 | 80.8 | 69.8 | 50.5 | 80.9 | **79.1** | 47.7 | 32.5 | 39.6 | 72.8 |
| VeRA | 61.1 | 89.1 | 70.1 | 99.1 | 89.1 | 89.1 | 53.9 | 81.7 | 96.2 | 84.9 | 75.5 | 71.7 | 57.4 | 46.6 | 74.4 | 66.9 | 47.3 | 23.6 | 30.6 | 68.8 |
| Adapter | 69.2 | 90.1 | 68 | 98.8 | 89.9 | 82.8 | 54.3 | 84 | 94.9 | 81.9 | 75.5 | 80.9 | 65.3 | 48.6 | 78.3 | 74.8 | 48.5 | 29.9 | 41.6 | 71.4 |
| Prefix | 75.5 | 90.7 | 65.4 | 96.6 | 86 | 78.5 | 46.7 | 79.5 | 95.1 | 80.6 | 74 | 69.9 | 58.2 | 40.9 | 69.5 | 72.4 | 46.8 | 23.9 | 34.4 | 67.6 |
| **HoRA** | 70.7 | **92.9** | **72.2** | **99.2** | **91.8** | **89.8** | **55.1** | **86.4** | **96.2** | **87.7** | **76.4** | **83.5** | **70.5** | **55** | **82.6** | 78.2 | **48.5** | **35** | 41.9 | **74.4** |

## C.4 ABLATION ON LOW-RANK MATRICES IN QUERY, VALUE, UP, AND DOWN PROJECTIONS

In addition to applying low-rank matrices to the query and value matrices in each layer, we further investigate whether our design on LoRA can generalize to scenarios where low-rank matrices are also incorporated into additional modules. Specifically, we extend our method to the proj_up and proj_down matrices, where the query and value matrices still follow our proposed design, while the proj_up and proj_down use the original version of low-rank matrices. As shown in Table 5, HoRA consistently achieves the highest performance compared to LoRA and DoRA in the LLaMA-13B setting, improving over LoRA and DoRA by an average of 1.5% and 0.4%, respectively. This

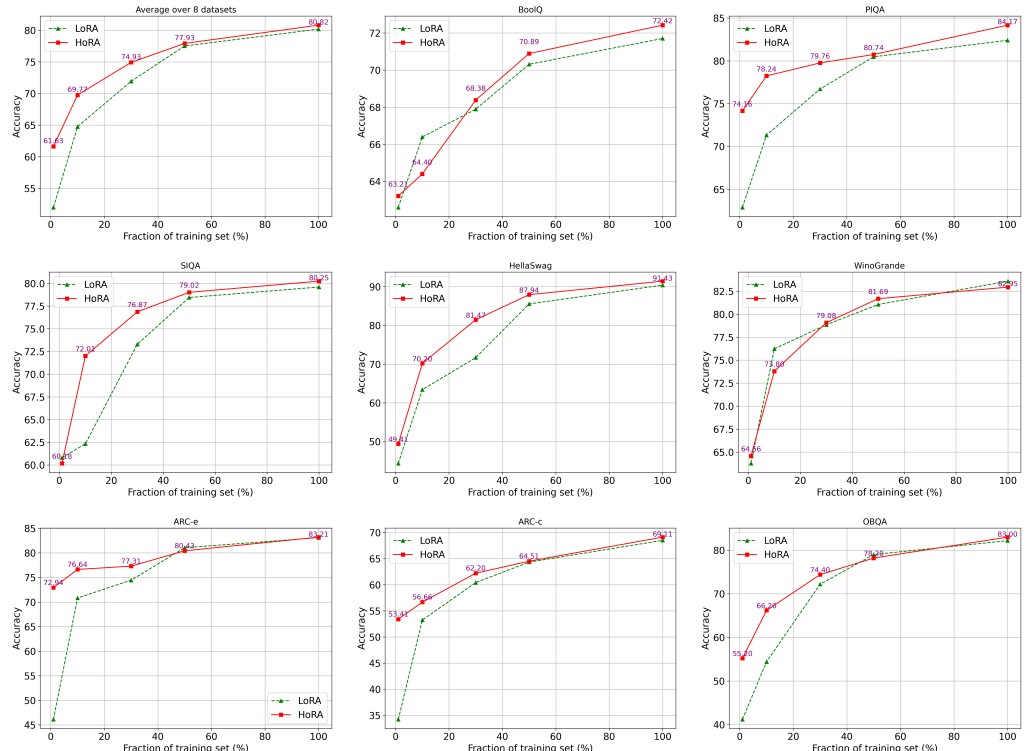

Figure 4: The detail of sample efficiency on each commonsense reasoning dataset with LLaMA-13B settings.

demonstrates that our proposed method, when applied to the query and value matrices in multi-head attention layers, remains effective even when low-rank matrices are additionally applied to other modules in the model.

Table 5: Ablation Study on Low-Rank Matrices in Query, Value, Up, and Down Weights.

| Model | Method | #Params (%) | BoolQ | PIQA | SIQA | HellaSwag | WinoGrande | ARC-e | ARC-c | OBQA | Average |
|-------|--------|-------------|-------|------|------|-----------|------------|-------|-------|------|---------|
| LLaMA-13B | LoRA | 0.57 | 72.11 | 83.73 | 80.5 | 90.5 | 83.74 | 82.11 | 68.09 | 82.4 | 80.4 |
| | DoRA | 0.58 | **72.42** | 84.98 | **81.17** | 91.81 | **84.61** | 84.22 | 69.88 | 82.8 | 81.49 |
| | HoRA | 0.57 | 72.23 | **85.8** | 80.25 | **92.47** | 84.37 | **84.47** | **70.99** | **84.4** | **81.87** |

## C.5    CONVERGENCE CURVES, RATIO OF LEARNABLE PARAMETERS, RUNTIME, AND FLOPS COMPARISONS

In this section, We also include a comparison of ratio of learnable parameters, FLOPs, and runtimes for low-rank–based approaches, include LoRA, DoRA, and HoRA, in the Table 6. During training, although HoRA incorporates additional hypernetwork modules that increase FLOPs, its runtimes and parameter ratios remain largely similar to those of LoRA. Meanwhile, HoRA achieves notable accuracy gains (up to +2.5% on the LLaMA-7B commonsense reasoning task) compared to LoRA. Moreover, at inference, since the low-rank matrices are merged into the pre-trained model, LoRA and HoRA have identical FLOPs and runtimes, making HoRA a practical and efficient approach for PEFT.

Table 6: Comparison among LoRA, DoRA, and HoRA on LLaMA-7B and LLaMA-13B settings

| Model | Method | #Params. (%) | GFLOPs | Runtimes | Performance |
|-------|--------|--------------|--------|----------|-------------|
| LLaMA-7B | LoRA | 0.25 | 6.63 | 4.17 | 74.09 |
|  | DoRA | 0.25 | 25.42 | 5.22 | 74.94 |
|  | HoRA | 0.28 | 41.32 | 4.21 | 76.64 |
| LLaMA-13B | LoRA | 0.2 | 12.88 | 7.25 | 80.18 |
|  | DoRA | 0.2 | 49.58 | 7.37 | 80.05 |
|  | HoRA | 0.21 | 80.51 | 7.33 | 80.82 |

We also compare the validation loss curves of LoRA and HoRA on the LLaMA-13B setting, as shown in Figure 5. Across most of the training period, HoRA exhibits consistently lower validation loss than LoRA, indicating improved convergence behavior.

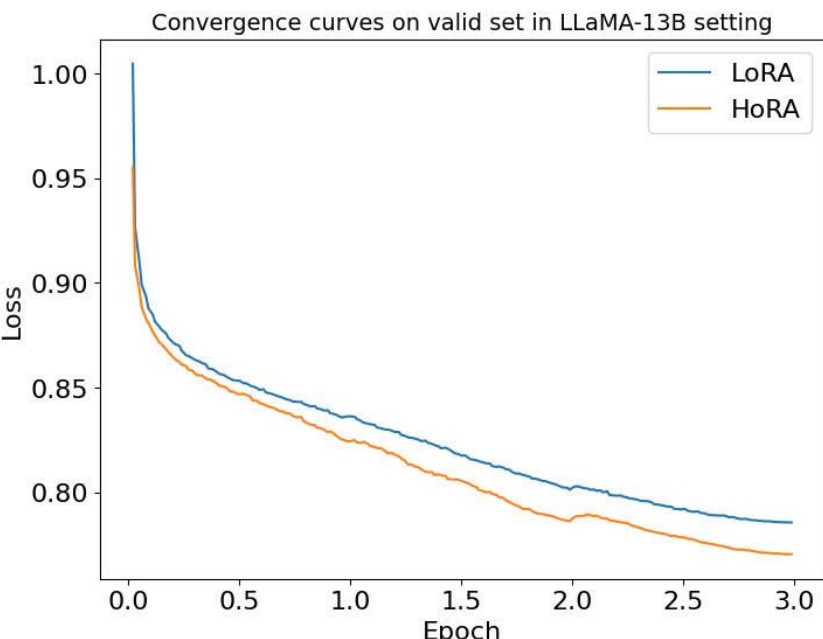

Figure 5: Validation loss curves comparing LoRA and HoRA on LLaMA-13B.

## D    USE OF LARGE LANGUAGE MODELS

In this paper, we use large language models (LLMs) solely for editorial support, including grammar refinement and spelling enhancements. We do not use LLMS for content generation, data analysis, or experimental design.

