# OpenReview forum: "HoRA: Cross-Head Low-Rank Adaptation with Joint Hypernetworks"
_ICLR.cc/2026/Conference — ICLR 2026 Conference Desk Rejected Submission_

### Official Review · Reviewer_zvwU · 2025-10-28

**Soundness:** 3
**Presentation:** 3
**Contribution:** 4
**Rating:** 6
**Confidence:** 3

**Summary:**

This paper introduces HoRA, a parameter-efficient fine-tuning method designed to overcome LoRA’s limitation of independently adapting each attention head. The authors reinterpret multi-head LoRA through the lens of Hierarchical Mixture of Experts and prove that this shared structure improves sample efficiency from exponential to polynomial rates. Empirical evaluations show consistent improvements over LoRA, DoRA, and adapters with ~0.1% additional parameters.

**Strengths:**

The paper is well written and provides a conceptually sound theoretical grounding for parameter sharing mechanisms. The insight that attention heads often exhibit redundancy is well known, and formalising this through a shared hypernetwork is an intuitive and promising approach. Cross-domain validation demonstrates flexibility across modalities.

**Weaknesses:**

W1. Some assumptions used for theoretical proofs may not be realistic in practical situations. For example, independent per-head subspaces are assumed to simplify the HMoE formulation, where each head corresponds to a separate expert with no shared latent factors. This is necessary for deriving Theorem 1, which provides the lower bound on sample efficiency under the non-shared setting. While this is not an inconsistency in the theory, attention heads in practice are coupled through shared normalisation layers, residuals, and joint gradients. Does this meaningfully impact the realism of the theory?

W2. Recent works like ReLoRA, AdaLoRA, Compacter, and HyperPEFT are either omitted or cited only tangentially. Since HoRA is a hypernetwork-based PEFT variant, stronger baselines with shared generators would improve the paper. Runtime and FLOPS analyses could also be included to strengthen efficiency claims.

W3. The formal analysis is mathematically rigorous but slightly disconnected from the empirical evaluation: the theorems rely on simplified mixture-model and Gaussian-noise assumptions, whereas experiments involve large-scale transformer fine-tuning. The paper could more clearly explain how the theoretical insights translate into architectural choices (e.g., activation functions, normalisation strategy, or hypernetwork structure).

**Questions:**

See weaknesses.

---

> ### Author Response · Authors · 2025-11-18
> **Response to Reviewer zvwU (Part 1)**
>
> Dear Reviewer zvwU,
>
> Thanks for your insightful reviews, and for giving **excellent grade (4)** to the contribution and **good grade (3)** to the soundness and presentation of our paper. In the sequel, we will provide our response to your questions regarding the paper and hope that it will completely address your concerns.
>
> **W1. Concerns regarding the assumptions for the theoretical analysis.**
>
> Thanks for your comments. We think that there might be confusion about the assumptions for our theoretical analysis, so let us clarify it. In particular, we did not impose any independence condition on per-head subspaces in Theorem 1. The hierarchical MoE in Eq.(6) followed directly from the connection between standard multi-head LoRA and hierarchical MoE presented in Section 3.1. Due to the lack of shared latent factors among experts, we showed in Theorem 1 that estimating low-rank matrices in the standard multi-head LoRA was not sample efficient.
>
> To address this issue, we proposed a shared structure across attention heads. Thus, it can be seen from Eq.(9) that experts in different heads shared latent factors of the form $\sigma(W_{1,j}A_{j})$. As a result, in Theorem 2, we demonstrated that our proposed shared structure helped significantly reduce the sample complexity of estimating low-rank matrices in multi-head LoRA.
>
> **W2. Concerns regarding hypernetwork-based PEFT variants.**
>
> We thank the reviewers for your suggestions on references for hypernetwork-based PEFT variants. We have added them to the revised manuscript.
>
> Regarding the comparison with stronger baselines with shared generators, we additionally compare our method with OP-LoRA [3], which leverages a hypernetwork to generate the low-rank matrices. The result are given below:
>
> | Method  | #Param (\%) | CUB-200-2011 | NABirds | Oxford Flowers | Stanford Dogs | Stanford Cars | AVG    |
> |---------|------------|--------------|---------|----------------|----------------|----------------|--------|
> | LoRA    | 0.55       | 84.6         | 78.2    | 98.9           | 85.1           | 77.1           | 84.78  |
> | OP-LoRA | 0.70       | 85.2         | 81.3    | 98.4           | 89.7           | 80.1           | 86.94  |
> | **HoRA** | 0.64  | **88.6**     | **85.9**| **99.2**       | **91.0**       | **85.0**       | **89.96** |
>
> A key difference between OP-LoRA and HoRA is that OP-LoRA use an independent hypernetworks for each attention head, leading to a notable increase in the number of trainable parameters compared to HoRA. Moreover, motivated by our theoretical analysis, we design shared hypernetwork between the keys and values instead of between the $A$ and $B$ matrices. As shown in the results, while OP-LoRA improves upon LoRA, which demonstrates benefits of using hypernetworks, our design shows an improvement in the performance with the theoretically motivated designs.
>
> Regarding the analysis of runtime and FLOPs, we reported the ratio of learnable parameters to enable a fair comparison across PEFT methods in the original manuscript. For completeness, in the revised manuscript, we have provided a comparison of FLOPs and runtimes for low-rank–based approaches, including LoRA, DoRA, and HoRA, which is shown in the table below. During training, although HoRA introduces additional hypernetwork modules that increase FLOPs, its runtimes and the ratio of learnable parameters remain nearly identical to those of LoRA. At the same time, HoRA achieves significant accuracy improvements (up to +2.5\% on the LLaMA-7B commonsense reasoning task) compared to LoRA. During inference, because the low-rank matrices are merged into the pre-trained model, the FLOPs and runtimes of LoRA and HoRA are identical, making HoRA a practical and efficient solution for PEFT.
>
> | **Model**      | **Method** | **#Params(\%)** | **GFLOPs** | **Runtime** | **Performance** |
> |:---------------|:-----------|----------------:|-----------:|------------:|----------------:|
> | **LLaMA-7B**   | LoRA       | 0.25            | 6.63       | 4.17        | 74.09           |
> |                | DoRA       | 0.25            | 25.42      | 5.22        | 74.94           |
> |                | **HoRA**   | 0.28            | 41.32      | 4.21        | **76.64**       |
> |----------------|------------|-----------------|------------|-------------|----------------|
> | **LLaMA-13B**  | LoRA       | 0.20            | 12.88      | 7.25        | 80.18           |
> |                | DoRA       | 0.20            | 49.58      | 7.37        | 80.05           |
> |                | **HoRA**   | 0.21            | 80.51      | 7.33        | **80.82**       |

---

> > ### Author Response · Authors · 2025-11-18
> > **Response to Reviewer zvwU (Part 2)**
> >
> > **W3. How do the theoretical insights translate into architectural choices?**
> >
> > Thanks for your feedback. Firstly, we would like to clarify that in our theoretical analysis, we consider a regression framework where the regression function admits the form of a hierarchical MoE, which is way more complex than traditional mixture models. Furthermore, the regression framework with a Gaussian noise assumption in our analysis has been widely used in several works of other parameter-efficient fine-tuning methods, namely prefix tuning [1] and LLaMA-Adapter [2].
> >
> > Secondly, let us explain how theoretical insights translate into architectural choices. In Theorem 1, we show that under the non-shared structure setting in Eq.(6), the sample complexity of estimating low-rank matrices in multi-head LoRA is suboptimal. Our theories indicate that the main reasons come from (i) a simple linear form of low-rank adapters in Eq.(6) and (ii) the lack of structure sharing of low-rank matrices across heads. Technically, these two factors yield an intrinsic interaction among low-rank matrices expressed in the following partial differential equation:
> >
> > $$\frac{\partial^2 F}{\partial B_{V}^{(u_1v_1)}\partial B_{V}^{(u_2v_2)}}=\frac{\partial^2 F}{\partial A_{V}^{(u_1v_1)}\partial A_{V}^{(u_2v_2)}},$$
> >
> > where $F(X,A,B):=\exp(X^{\top}(P_{Q}+B_{Q}A_{Q})P_KX)(P_{V}+B_{V}A_{V})X$.
> >
> > The above theoretical insights motivate us to employ a shared structure across attention heads where we use a joint hypernetwork to generate low-rank matrices, making low-rank adapters more expressive than the simple linear form. Furthermore, we also encourage low-rank matrices $\boldsymbol{A}$ in different heads to share information through the shared structure setting. As a result, in Theorem 2, we show that the sample complexity of estimating low-rank matrices decreased substantially.
> >
> > **References**
> >
> > [1] Le et al. Revisiting prefix-tuning: Statistical benefits of reparameterization among prompts. In ICLR, 2025.
> >
> > [2] Diep et al. On zero-initialized attention: Optimal prompt and gating factor estimation. In ICML, 2025.
> >
> > [3] Piotr Teterwak et al. OP-LoRA: The Blessing of Dimensionality. In Arxiv, 2025.

---

### Official Review · Reviewer_f5c6 · 2025-10-30

**Soundness:** 3
**Presentation:** 3
**Contribution:** 3
**Rating:** 6
**Confidence:** 4

**Summary:**

This paper proposes HoRA (Cross-Head Low-Rank Adaptation), a new PEFT method designed to enhance the adaptability of large pre-trained models by introducing cross-head low-rank interactions. Unlike conventional LoRA-based approaches that apply low-rank updates independently to each attention head, HoRA allows cross-head sharing and interactions to better capture inter-head dependencies while maintaining efficiency. The authors provide a detailed theoretical analysis that establishes the expressiveness and rank properties of HoRA, supported by formal proofs and ablation-based empirical validation. Extensive experiments across both natural language processing and vision benchmarks demonstrate consistent performance improvements over baselines such as LoRA, DoRA, and HiRA under similar parameter budgets.

**Strengths:**

- The paper offers rigorous and clearly structured mathematical analysis, including rank-related theorems and proofs that ground the proposed cross-head low-rank formulation in solid linear algebraic reasoning. This theoretical depth enhances the credibility of the method and distinguishes it from many empirical-only PEFT studies.

- The authors conduct extensive experiments across NLP and vision tasks, showing HoRA‘s performance surpasses or matches strong baselines while maintaining comparable parameter efficiency. The evaluation includes detailed ablation and convergence studies that support the claimed benefits.


- The paper is well-organized and easy to follow. Figures and tables effectively complement the text. The proofs are self-contained and pedagogically written.

**Weaknesses:**

(1) **Limited experimental diversity and overemphasis on sample efficiency.** While the paper provides compelling evidence that HoRA improves sample efficiency in few-shot and low-data settings, the experimental validation remains narrowly focused on this aspect. It is unclear whether the cross-head hypernetwork design continues to yield benefits under large-scale or high-resource regimes, or whether the gains vanish when data is abundant. Broader experiments on full-data fine-tuning, convergence behavior, and computational trade-offs would significantly strengthen the empirical foundation.

(2) **Limited Analysis of Rank and Correlation Dynamics.**
The theoretical section claims improved expressiveness due to cross-head mixing, yet the experiments do not empirically examine the effective rank or correlation between head updates after fine-tuning. Providing SVD-based or cosine-similarity analysis would substantiate the claim that HoRA genuinely induces more diverse and high-rank adaptations rather than introducing redundant cross-talk.

**Questions:**

- **Experimental scope appears narrow, focusing mainly on few-shot sample efficiency.**
Specifically, I suggest including convergence curves, runtime or FLOPs comparisons, and GPU memory usage statistics versus LoRA/HiRA. These would clarify whether HoRA maintains its efficiency and performance benefits when data and computational resources are not constrained.

- **Scaling experiments to larger foundation models.** Have the authors tested HoRA on larger foundation models (e.g., LLaMA-13B or ViT-L)? It would be valuable to see whether the cross-head benefits persist as model size grows, or if the method’s advantage diminishes with increased intrinsic capacity.

- **Practical Applicability.** Since HoRA introduces inter-head coupling, does it affect the modularity or mergeability of adapters (e.g., combining task-specific adapters as in LoRA-Merge)? Discussing compatibility with existing PEFT deployment frameworks would clarify its real-world usability.

---

> ### Author Response · Authors · 2025-11-18
> **Response to Reviewer f5c6 (Part 1)**
>
> Dear Reviewer f5c6,
>
> Thanks for the wealth of your comments, and for giving **good grade (3)** to the contribution, soundness and presentation of our paper. In the sequel, we will provide our response to your questions regarding the paper and hope that it will completely address your concerns.
>
> **W1: Limited experimental diversity and overemphasis on sample efficiency**
>
> Thanks for your comments. We are sorry for confusion about the experimental settings and we will clarify it here. In fact, only the experiments in Section 5.1 and Appendix C.2 focus on sample efficiency, which are conducted on commonsense reasoning tasks using the LLaMA-7B setting. All other experiments - including the two VTAB-1k and FGVC tasks in image classification, as well as the commonsense reasoning task on large-scale dataset - are trained on the full datasets. In these settings, our method consistently achieves significant improvements compared to LoRA, demonstrating that the benefits of our approach do not vanish when the data is abundant.
>
> **W2: Limited Analysis of Rank and Correlation Dynamics in claim "HoRA improves expressiveness and induces more diverse and high-rank adaptations".**
>
> Thanks for your feedback. We think that there might be confusion about our claims in the section ``Theoretical Developments", so let us clarify it here. In that section, we demonstrate that HoRA helps reduce the sample complexity of estimating low-rank matrices from exponential order (vanilla LoRA) to polynomial order. In addition, our main contribution is to propose a LoRA variant that helps reduce the number of trainable parameters and impose some prior via the shared structure without sacrificing expressiveness as observed in the experiments. For that purpose, we introduce HoRA in which the hypernetwork allows the model to reuse common patterns across heads (the first layer is shared across all heads) while preserving head-specific variations (the second layer is distinct for each head). While the analysis of rank and correlation dynamics is important, it lies beyond the scope of our work. Therefore, we leave this problem for future development.
>
>
>  **Q1: Convergence curves, runtime and FLOPs comparisons.**
>
> We thank the reviewer for your valuable feedback. In the original manuscript, we reported the ratio of learnable parameters to allow a fair comparison across various PEFT methods. For completeness, in the revised manuscript, we have included a comparison of FLOPs and runtimes for low-rank–based approaches, such as LoRA, DoRA, and HoRA, which is shown in the table below. The validation loss curves comparing LoRA and HoRA on the LLaMA-13B setting are also presented in Figure 5 in the revised manuscript. During training, although HoRA incorporates additional hypernetwork modules that increase FLOPs, its runtimes and parameter ratios remain largely similar to those of LoRA. Meanwhile, HoRA achieves notable accuracy gains (up to +2.5\% on the LLaMA-7B commonsense reasoning task) compared to LoRA. At inference, since the low-rank matrices are merged into the pre-trained model, LoRA and HoRA have identical FLOPs and runtimes, making HoRA a practical and efficient approach for PEFT.
>
> | **Model**      | **Method** | **#Params(\%)** | **GFLOPs** | **Runtime** | **Performance** |
> |:---------------|:-----------|----------------:|-----------:|------------:|----------------:|
> | **LLaMA-7B**   | LoRA       | 0.25            | 6.63       | 4.17        | 74.09           |
> |                | DoRA       | 0.25            | 25.42      | 5.22        | 74.94           |
> |                | **HoRA**   | 0.28            | 41.32      | 4.21        | **76.64**       |
> |----------------|------------|-----------------|------------|-------------|----------------|
> | **LLaMA-13B**  | LoRA       | 0.20            | 12.88      | 7.25        | 80.18           |
> |                | DoRA       | 0.20            | 49.58      | 7.37        | 80.05           |
> |                | **HoRA**   | 0.21            | 80.51      | 7.33        | **80.82**       |
>
> **Q2: Scaling experiments to larger foundation models.**
>
>  Thanks for your questions. In the original manuscript, we already compared the performance of HoRA and LoRA on commonsense reasoning tasks under the LLaMA-13B setting using full-scale training data, as shown in Table 3. For completeness, we provided results on sample-efficiency experiments with LLaMA-13B in Figure 4 in the revised manuscript. From that figure, we observe that HoRA continues to yield clear sample-efficiency improvements over LoRA, even at the scale of 13B parameters.

---

> > ### Author Response · Authors · 2025-11-18
> > **Response to Reviewer f5c6 (Part 2)**
> >
> > **Q3: Since HoRA introduces inter-head coupling, does it affect the modularity or mergeability of adapters (e.g., combining task-specific adapters as in LoRA-Merge)? Discussing compatibility with existing PEFT deployment frameworks would clarify its real-world usability.**
> >
> > Thanks for your questions. Recall that the main difference between vanilla LoRA and HoRA is the formulation of low-rank matrices $B$. In particular, while $\boldsymbol{B}_{Q,i}$ and $\boldsymbol{B}_{V,i}$ are separated across attention heads in vanilla LoRA, we propose to generate these matrices with the hypernetworks as
> >
> > $$B_{Q,i}=W_{Q,B,2}\sigma_2(W_{B,1}\mathrm{LN}(B_{i})),$$
> >
> > $$B_{V,i}=W_{V,B,2}\sigma_2(W_{B,1}\mathrm{LN}(B_{i})),$$
> >
> > where $i$ denotes the head index. It can be seen that $B_{Q,i}$ only shares the matrix $B_{Q,B,2}$ across heads to capture general knowledge, while it keeps the embedding vector $B_{i}$ unshared to learn head-specific knowledge. Similar arguments apply for $B_{V,i}$. Thus, HoRA allows to reuse common patterns across heads (the first layer is shared across all heads) while preserving head-specific variations (the second layer is distinct for each head). Hence, HoRA in general does not affect modularity or mergeability of adapters.
> >
> > Next, we confirm that HoRA remains fully compatible with standard PEFT deployment frameworks: the hypernetwork-generated low-rank matrices can still be applied to the base model weights as LoRA at inference, and task-specific adapters generated by hypernetworks can be stored and loaded modularly in inference time for merging to pre-trained model. Therefore, the modularity and mergeability of adapters can still be preserved at the matrix level, even though the hypernetwork introduces inter-head coupling during training.

---

### Official Review · Reviewer_SKLj · 2025-11-01

**Soundness:** 3
**Presentation:** 3
**Contribution:** 4
**Rating:** 8
**Confidence:** 2

**Summary:**

This paper proposes to work with hypernetworks, which generate low-rank matrices across attention heads and layers.
To achieve this goal the paper establishes a theoretical connection between low rank adaptation methods, multi-head self attention and hierarchical mixture of experts.
The paper's theoretical considerations suggest that the mechanism improves sample efficiency from an exponential to polynomial rate.

**Strengths:**

- Very well written background section
- Innovative idea
- Theoretically and experimentally sound.

**Weaknesses:**

- The experimental section does not compare to tensor decomposition based finetuning work.
- The paper does not tell us much about its computational cost.

**Questions:**

- The paper approaches the multi-head tuning problem from a hyper-network perspective, which is refreshing.
- Related work employs tensor decompositions i.e. Tucker
or Tensor-Train to capture correlation along the head dimension (see https://arxiv.org/pdf/2212.03145 for example). Since this is similar in spirit, perhaps it's worth comparing to this and similar approaches?
- Similarly, this paper's results for VTAB-1k are better than what the NAS-Based NOAH paper reported (https://arxiv.org/pdf/2206.04673). Using the numbers from the NOAH paper
``` python
np.mean([69.6, 92.7, 70.2, 99.1, 90.4, 86.1, 53.7, 84.4, 95.4, 83.9, 75.8, 82.8, 68.9, 49.9, 81.7, 81.8, 48.3, 32.8, 44.2])
```
yields $73.24$, HoRA is better. Unfortunately, NOAH reports a group wise mean which is misleading. This paper could be an opportunity to set the record straight. Perhaps, the authors are willing to do it?

---

> ### Author Response · Authors · 2025-11-18
> **Response to Reviewer SKLj**
>
> Dear Reviewer SKLj,
>
> Thanks for your constructive and valuable feedback, and for giving **excellent grade (4)** to the contribution and **good grade (3)** to the soundness and presentation of our paper. In the sequel, we will provide our response to your questions regarding the paper and hope that it will completely address your concerns.
>
> **W1, Q1: The experimental section does not compare to tensor decomposition based finetuning work.**
>
> Thanks for your comments. We would like to clarify that we do not compare HoRA to tensor decomposition based PEFT methods since they have different benefits. In particular, Tensor decompositions based PEFT methods (i.e., Tucker or Tensor-Train) aim to capture correlations across attention heads. For that purpose, they impose a fixed factorization on a reshaped attention tensor, which constrains cross-head interaction through core tensors. Meanwhile, by coupling their adaptation through a shared generator, HoRA encourages cross-head information sharing, and thus reducing the number of trainable parameters without sacrificing expressiveness, rather than capturing correlation between heads. Furthermore, the shared hypernetwork parameterization in HoRA yields provably improved sample efficiency, while existing tensor decomposition-based methods do not provide such theoretical guarantee.
>
> **W2: The paper does not tell us much about its computational cost.**
>
> Thanks for your feedback. In the original manuscript, we reported the ratio of learnable parameters to enable a fair comparison across PEFT methods. For completeness, we further provide a comparison of FLOPs and runtimes for low-rank–based approaches, including LoRA, DoRA, and HoRA, in the table below. During training, although HoRA introduces additional hypernetwork modules that increase FLOPs, its runtimes and the ratio of learnable parameters remain nearly identical to those of LoRA. At the same time, HoRA achieves significant accuracy improvements (up to +2.5\% on the LLaMA-7B commonsense reasoning task) compared to LoRA. During inference, because the low-rank matrices are merged into the pre-trained model, the FLOPs and runtimes of LoRA and HoRA are identical, making HoRA a practical and efficient solution for PEFT.
>
> | **Model**      | **Method** | **#Params(\%)** | **GFLOPs** | **Runtime** | **Performance** |
> |:---------------|:-----------|----------------:|-----------:|------------:|----------------:|
> | **LLaMA-7B**   | LoRA       | 0.25            | 6.63       | 4.17        | 74.09           |
> |                | DoRA       | 0.25            | 25.42      | 5.22        | 74.94           |
> |                | **HoRA**   | 0.28            | 41.32      | 4.21        | **76.64**       |
> |----------------|------------|-----------------|------------|-------------|----------------|
> | **LLaMA-13B**  | LoRA       | 0.20            | 12.88      | 7.25        | 80.18           |
> |                | DoRA       | 0.20            | 49.58      | 7.37        | 80.05           |
> |                | **HoRA**   | 0.21            | 80.51      | 7.33        | **80.82**       |
>
> **Q2: Similarly, this paper's results for VTAB-1k are better than what the NAS-Based NOAH paper reported. Unfortunately, NOAH reports a group wise mean which is misleading. This paper could be an opportunity to set the record straight. Perhaps, the authors are willing to do it?**
>
> We thank the reviewer for their feedback and for recognizing our work. For further clarification regarding our experiments, our proposed method is built upon the prior work in PEFT [1], and we retain most of the original experimental settings, with only the learning rate and weight decay tuned to ensure a fair comparison with the methods reported in that paper. In our study, beside accuracy for each dataset, we report two types of mean results: first, the mean accuracy for each group (Natural, Specialized, and Structured), and second, the overall mean accuracy across all datasets in VTAB-1k. This performance report is the same as previous work [1] for fair comparison. Using group-wise mean, HoRA achieves a score of 76.77, which remains state-of-the-art compared to the methods reported in [2].
>
> **References**
>
> [1] Xin, Yi, et al. V-petl bench: A unified visual parameter-efficient transfer learning benchmark. In NeurIPS, 2024.
>
> [2] Zhang, Yuanhan, et al. Neural prompt search. IEEE Transactions on Pattern Analysis and Machine Intelligence, 2024.

---

> > ### Comment · Reviewer_SKLj · 2025-11-23
> > **I continue to recommend acceptance**
> >
> > Having read the author's response and the other reviews I continue to recommend including this paper in the conference proceedings.

---

> ### Author Response · Authors · 2025-11-23
> **Thank You**
>
> Dear Reviewer SKLj,
>
> We would like to thank you for maintaining your strongly positive rating of 8 for our paper, we really appreciate it. Please feel free to let us know if you have any further concerns.
>
> Best,
>
> The Authors

---

### Author Response · Authors · 2025-11-18
**General Response**

Dear Area Chairs and Reviewers,

We would like to thank you for your value feedback and constructive comments, which have helped us improve the paper substantially. We are encouraged by the endorsement that:

**Contribution:** Innovative idea *(Reviewer SKLj)*. The paper offers rigorous and clearly structured mathematical analysis, enhancing the credibility of the method and distinguishes it from many empirical-only PEFT studies *(Reviewer f5c6)*. The insight that attention heads often exhibit redundancy is well known, and formalising this through a shared hypernetwork is an intuitive and promising approach *(Reviewer zvwU)*.

**Soundness:** Theoretically and experimentally sound *(Reviewer SKLj)*. The evaluation includes detailed ablation and convergence studies that support the claimed benefits *(Reviewer f5c6)*. The paper provides a conceptually sound theoretical grounding for parameter sharing mechanisms *(Reviewer zvwU)*.

**Presentation:** Very well written background section *(Reviewer SKLj)*. The paper is well-organized and easy to follow. Figures and tables effectively complement the text. The proofs are self-contained and pedagogically written *(Reviewer f5c6)*. The paper is well written *(Reviewer zvwU)*.

We appreciate the reviewers' positive feedback and thoughtful reviews. In the sequel, we will address reviewers' concerns regarding our manuscript and describe the corresponding changes made based on their valuable recommendations.

Best regards,

The Authors

---

### Note · Program_Chairs · 2026-01-17
**Submission Desk Rejected by Program Chairs**

The following references in this submission do not refer to real documents and/or have major errors in bibliographic information:

 Yifan Gao, Shiyue Chen, Shujie Wang, and Furu Wei. Prefix-tuning for parameter-efficient speech recognition. In IEEE Spoken Language Technology Workshop (SLT), 2022.